# TM5-FASST: a global atmospheric source-receptor model for rapid impact analysis of emission changes on air quality and short-lived climate pollutants

Rita Van Dingenen[1], Frank Dentener[1], Monica Crippa[1], Joana Leitao[1,2], Elina Marmer[1,3], Shilpa Rao[4], Efisio Solazzo[1], Luana Valentini[5]

[1] European Commission, Joint Research Centre (JRC), Ispra (VA), Italy
[2] Now at IASS – Institute for Advanced Sustainability Studies, Potsdam, Germany
[3] Now at University of Hamburg, Germany
[4] Norwegian Institute for Public Health, PO Box 4404 Nydalen, N-0403 Oslo, Norway,
[5] GFT Italia S.r.l., Via Sile, 18 20139 Milan, Italy

*Correspondence to*: Rita Van Dingenen (rita.van-dingenen@ec.europa.eu)

**Abstract** This paper describes, documents and validates the TM5-Fast Scenario Screening Tool (TM5-FASST), a global reduced-form air quality source-receptor model that has been designed to compute ambient pollutant concentrations as well as a broad range of pollutant-related impacts on human health, agricultural crop production, and short-lived pollutant climate metrics, taking as input annual pollutant emission data aggregated at the national or regional level. The TM5-FASST tool, providing a trade-off between accuracy and applicability, is based on linearized emission-concentration sensitivities derived with the full chemistry-transport model TM5. The tool has been extensively applied in various recent critical studies. Although informal and fragmented validation has already been performed in various publications, this paper provides a comprehensive documentation of all components of the model and a validation against the full TM5 model. We find that the simplifications introduced in order to generate immediate results from emission scenarios do not compromise the validity of the output and as such TM5-FASST is proven to be a useful tool in science-policy analysis. Furthermore, it constitutes a suitable architecture for implementing the ensemble of source-receptor relations obtained in the frame of the HTAP modelling exercises, thus creating a link between the scientific community and policy-oriented users.

## 1    Introduction

A host of policies influence the emissions to air. In principle any policy that influences the economy and use of resources will also impact emissions into the atmosphere. Specific air pollution policies aim to mitigate the negative environmental impacts of anthropogenic activities, some of which may be affected by other policies, like climate mitigation actions, transport modal shifts or agricultural policies. Further, air quality policies may impact outside their typical environmental target domains (human and ecosystem health, vegetation and building damage,…) for instance through the role played by short-lived pollutants in the Earth's radiation balance (Myhre et al., 2011; Shindell et al., 2009). Insight into the impacts of policies in a multi-disciplinary framework through a holistic approach could contribute to a more efficient and cost-effective implementation of control measures (e.g. Amann et al., 2011; Maione et al., 2016; Shindell et al., 2012).

Several global chemical transport models are available for the evaluation of air pollutants levels from emissions, sometimes in combination with off-line computed climate relevant metrics such as optical depth or instantaneous radiative forcing (e.g. Lamarque et al., 2013; Stevenson et al., 2013). These models provide detailed output, but are demanding in terms of computational and human resources for preparing input, running the model, and analyzing output. Further they often lack flexibility to evaluate ad-hoc a series of scenarios, or perform swift what-if analysis of policy options. Therefore there is a need for computationally-efficient methods and tools that provide an integrated environmental assessment of air quality and climate policies, which have a global dimension with sufficient regional detail, and evaluate different impact categories in an internally consistent way. Reduced-form source-receptor models are a useful concept in this context. They are typically constructed from pre-computed emission-concentration transfer matrices between pollutant source regions and receptor regions. These matrices emulate underlying meteorological and chemical atmospheric processes for a pre-defined set of

meteorological and emission data, and have the advantage that concentration responses to emission changes are obtained by a simple matrix multiplication, avoiding expensive numerical computations. Reduced-form source-receptor models (SRM) are increasingly being used, not only to compute atmospheric concentrations (and related impacts) from changes in emissions but they have also proven to be very useful in cost optimization and cost-benefit analysis because of their low computational cost (Amann et al., 2011). Further, because of the detailed budget information embedded in the source-receptor matrices, they are applied for apportionment studies, as a complementary approach to other techniques such as adjoint models (e.g. Zhang et al., 2015) and chemical tagging (e.g. Grewe et al., 2012).

Although the computational efficiency of SRMs comes at a cost of accuracy, regional detail and flexibility in spatial arrangement of emissions, they have been successfully applied in regional studies (Foley et al., 2014; Li et al., 2014; Liu et al., 2017; Porter et al., 2017) and have demonstrated their key role in policy development (Amann et al., 2011).

An extensive collaborative global chemistry modelling effort evaluated local and long-range pollutant responses to emission reductions in 4 world regions in the first phase of HTAP (Dentener et al., 2010; Fiore et al., 2009), hereafter referred to as HTAP1. The resulting ensemble source-receptor relations between those regions have been used to evaluate the driving factors behind regional ozone changes in 5 world regions (Wild et al., 2012). Similarly, Yu et al. (2013) evaluated aerosol radiative forcing (RF) from HTAP1 results, whereas Fry et al. (2012) assessed the RF effects by ozone. Several papers in this special issue (e.g. Stjern et al., 2016) are devoted to advance the HTAP analysis with new models and methodologies.

One of the participating global models in the HTAP1 assessment was the 2-way nested global chemical transport model TM5, applied with 1°x1° resolution over the continents (Krol et al., 2005)..In order to address the need for swift scenario analysis, we used TM5 to develop a reduced-form global source-receptor (SR) model, with the capability to assess in a single framework a broad portfolio of short-lived pollutants environmental impacts at the global scale, including their interaction with climate, impact on human health, on vegetation and on ecosystems. The reduced-form version was named "TM5-Fast Scenario Screening Tool" (TM5-FASST). The TM5-FASST approach refines and extends the one developed in the HTAP1 assessment by defining source-receptor regions at a finer resolution and by implementing a direct emission-based calculation of pollutant concentrations and their impacts. To our knowledge such a comprehensive global source-receptor model for a variety of components and impacts (primary and secondary particulate matter, trace gases, wet and dry deposition, climate and health metrics) is at this moment not available for fast impact assessments. The need for models like TM5-FASST is demonstrated by its extensive application in various critical studies (OECD, 2016; Rao et al., 2016; The World Bank, The International Cryosphere Climate Initiative, 2013; UNEP, 2011). An overview of earlier studies with TM5-FASST, in which fragmented and informal validation has already been performed, is given in section S1 of the Supplemental Information (SI).

The tool is undergoing continuous developments and updates regarding metrics and impact evaluations. Hereafter we will refer to the native chemical transport model and the derived SR model as TM5 and TM5-FASST_v0 (or its shortcuts TM5-

FASST and FASST) respectively, with version number v0 referring to the features and methodologies described in this paper and as applied in the earlier assessments. The present paper is a comprehensive documentation of the model and its validation against TM5, to ensure credibility and future applications.

In section 2, we describe the methods implemented in TM5-FASST to evaluate in a single framework a broad portfolio of short-lived air pollutants (including $CH_4$) and their environmental impacts, such as interaction with climate, impact on human health, on natural vegetation and crops. Section 3 focuses on how the derived reduced-form TM5-FASST replicates the full native TM5 model in terms of linearity, additivity and application to a realistic set of future scenarios. We also evaluate the performance of TM5-FASST against some case studies from the literature. We finish with a discussion (section 4) of the limitations of the methodology, future development paths and possible ways forward for the best-use of such modelling systems for future policy assessments.

## 2    Methods

### 2.1    The native TM5 model.

The Tracer Model version 5 (TM5) is a 3-dimensional global atmospheric chemical transport model that simulates the transport, chemical processes, as well as wet and dry deposition of chemically active atmospheric trace gases (e.g. ozone ($O_3$), $SO_2$ $NO_x$, VOCs, $NH_3$), and particulate matter components, including $SO_4^{2-}, NO_3^-$, $NH_4^+$, primary $PM_{2.5}$ and its components black carbon, organic carbon, sea salt, and mineral dust. Biogenic secondary organic aerosol (BSOA) was included following the AEROCOM recommendation (Dentener et al., 2006a; Kanakidou et al., 2005) which parameterized BSOA formation from natural VOC emissions as a fixed fraction of the primary emissions. The relative fraction compared to the anthropogenic POM emissions varies spatially, with a higher contribution in regions were the emissions of terpene emissions are higher. SOA from anthropogenic emission was not explicitly included in the current simulations.

Model version TM5-JRC-cy2-ipcc (abbreviated TM5) was used to compute the source receptor relationships as first described by Krol et al. (2005). This model version was used in the PhotoComp scenario studies (e.g. Dentener et al., 2006b; Stevenson et al., 2006) and in the HTAP1 multi-model source receptor assessment (e.g. Anenberg et al., 2009; Fiore et al., 2009; Wild et al., 2012). TM5 results used in the present study allow comparison with a range of other global model results in HTAP1, but ignore subsequent updates and improvements in TM5 as for instance described in Huijnen et al. (2010), which we consider not critical for this study. The most recent TM5 model does no longer consider zoom regions, but recoded the model into a Massive Parallel framework, enabling efficient execution on modern computers. While global horizontal resolution (1°x1°) is similar to the resolution of the most refined zoom region in TM5, vertical resolution was increased. Further, the model also uses vertical mass fluxes from the parent ECMWF meteorological model, not available at

the time of development of TM5-cy2-ipcc, which could lead to somewhat different mixing characteristics. The gas phase chemical module has been updated to a modified version of CMB5.

The TM5 model operates with offline meteorology from the European Centre for Medium range Weather Forecasts (ECMWF; 6 hours IFS forecast). These data are stored at a 6-hourly horizontal resolution of 1°x1° for large-scale 3D fields, and 3-hourly resolution for parameters describing exchange processes at the surface. Of the 60 vertical layers in the operational (OD) ECMWF model (status ca. 2008), a subset of 25 layers is used within TM5, including 5 in the boundary layer, 10 in the free troposphere, and 10 stratospheric layers. Although for most health and ecosystem impacts only the surface level fields are required, climate metrics (e.g. radiative forcing) require the full vertical column and profile information. Therefore base simulation and perturbed pollutant concentrations were calculated and stored for the 25 vertical levels of the model as monthly means, and some air quality-relevant parameters as hourly or daily fields. Meteorological fields are obtained from the ECMWF operational forecast representative for the year 2001. The implications of using a single meteorological year will be discussed in section 4.2.

TM5 utilizes a so-called two-way nested approach, which introduces refinements in both space and time in predefined regions. The nesting comprises a regional high resolution 'zoom' (1°x1°) within relatively coarse global resolution (6°x4°), and a transitional grid of 3°x2°, as illustrated in Fig. S2.1 of the SI. A pre-processing software aggregated the 3D 1°x1° meteorological fields into the abovementioned coarser resolutions in a fully mass-conserving way. TM5 has a flexible choice of regional extent and amount of zoom regions. For instance, the HTAP1 simulation setup utilized a set of 4 simultaneous 1°x1° zooms nested over Europe, North America, South and East Asia. Since hundreds of simulations are needed to drive the TM5-FASST Source-Receptor model, due to computational constraints, it was decided to use single zoom regions, covering the countries and regions for which emission perturbation studies were carried out. For example, the European zoom would contain all European countries, the East Asian zoom region countries like China and Korea, etc. An overview of zoom regions and their regional extent is given in SI section S2. Post-processing software merged the outputs of base and sensitivity simulations into uniform 1°x1° fields.

We note that at the time of development of the 'zoom' model, the TM5 specific model set-up allowed to perform photochemistry and aerosol calculations with a relatively high 1°x1° resolution in the source regions, whereas other global models were operating at much coarser resolutions (typically 2.8° x 2.8°). With the introduction of massive parallel computing, however, this comparative advantage is now slowly disappearing, and global model resolutions of 1°x1° or finer are now becoming more common (see the model descriptions in this special issue, e.g. Liang et al., 2018). The model grid resolution influences the predicted pollutant concentrations as well as the estimated population exposure, especially near urban areas where strong gradients occur in population density and pollutant levels, which cannot be resolved by the 1°x1° resolution. In section 2.4 we describe a methodology to improve population $PM_{2.5}$ exposure estimates by applying sub-grid concentration adjustments based on high-resolution ancillary data. The bias introduced by model resolution affects as well

computed SR matrices, e.g. off-setting the share of 'local' versus 'imported' pollution in a given receptor region. We will discuss this aspect more in detail in section 4.3.

More details on the TM5 model, together with an overview of earlier validation efforts is provided in Section S2 of the SI.

## 2.2 Base emissions

As base simulation emissions we use the community generated representative concentration pathways (RCP) pollutant emissions for the year 2000 at 1°x1° resolution, prepared for IPCC 5[th] Assessment (Lamarque et al., 2010). Relevant emitted anthropogenic pollutants include $SO_2$, $NO_x$, $NH_3$, black carbon (BC), organic carbon (OC), NMVOC, CO and $CH_4$. (Semi-) natural emissions (sea-salt, mineral dust, volcanoes, lightning, vegetation, biomass burning, and terrestrial and oceanic DMS) for the base simulations were included following the recommendations for the AEROCOM study (Dentener et al.,

2006a) but they are not affected in the perturbation simulations where we consider only perturbations of anthropogenic emissions.

## 2.3 Air pollutants source-receptor relations

In general, air quality source-receptor models (AQ-SRM) link emissions of pollutants in a given source region with downwind concentrations and related impacts, implicitly including the underlying effects of meteorology and atmospheric

chemical and physical processes. The source region is any point or area from which emissions are considered; the receptor is any point or area at which the pollutant concentration and impact is to be evaluated. Primary pollutants concentrations are primarily affected by dry and wet removal from the atmosphere (e.g. elemental carbon, seasalt and mineral dust) after being emitted. Secondary pollutants are formed from reactions of primary emissions, e.g. $NO_2$ forms nitrate aerosol but also leads to the formation of $O_3$; emitted $SO_2$ is transformed into sulfate aerosols.

A change of pollutant emissions has the potential to change the chemical formation of other secondary species, e.g. $NO_2$ affects the oxidative capacity of the atmosphere and therefore influences the lifetime of methane. In summary, a specific secondary component and related impact can be influenced from one or more emitted precursors, and an emitted precursor can change the impact from one or more pollutants. An AQ-SRM will need to include a functional relationship between each precursor and each relevant pollutant or pollutant metric, for each source region and each receptor region.

TM5-FASST_v0 has been designed as a reduced-form SRM: the relation between the emissions of compound $i$ from source $x$ and resulting concentration (or burden) of pollutant $j$ at receptor $y$ is expressed by a simple functional relation that mimics the underlying meteorological and chemical processes. In the current version v0 of TM5-FASST the emission-concentration relationship is locally approximated by a linear function expressing the change in pollutant concentration in the receptor region upon a change in precursor emissions in the source region with the generic form $dC_y = SRC \times dE_x$ where $dC_y$ equals

the change in the pollutant concentration compared to a reference concentration in receptor region $y$, $dE_x$ is the change in precursor emission compared to a reference emission in source region $x$, and SRC the source-receptor coefficient for the specific compound and source-receptor pair – in this case emulating atmospheric processes linked to the meteorology in

2001. The source-receptor coefficients are implemented as matrices with dimension [$n_x$,$n_y$] with $n_x$ and $n_y$ the number of source and receptor regions respectively. A single SR matrix is available for each precursor and for each resulting component from that precursor. Table 1 gives an overview of all precursor – pollutant links that have been included.

For TM5-FASST_v0 we defined 56 source regions, as shown in Fig. 1. A detailed break-down of regions by country is given in Section S2 of the SI. The choice of regions has been made to obtain an optimal match with integrated assessment models such as IMAGE (Eickhout et al., 2004; van Vuuren et al., 2007), MESSAGE (Riahi et al., 2007), GAINS (Höglund-Isaksson and Mechler, 2005) as well as the POLES model (Russ et al., 2007; Van Aardenne et al., 2007). Most European countries are defined as individual source regions, except for the smallest countries, which have been aggregated. In the current version v0, the USA, China and India are treated as a single emission regions each, i.e. without break-down in states or provinces. Although most integrated assessment models cover Africa, South America, Russia and South-East Asia as a single socio-economic entity, it was decided to sub-divide these regions, to account for climatological difference in these vast continents. Apart from the 56 regions, source-receptor coefficients were calculated between global international shipping and aviation as sources, and the global grid as receptor, resulting in $n_x = 58$ source functions.

The SR matrices, describing the concentration response in each receptor upon a change in emissions in each source region, have been derived from a set of simulations with the full chemical transport model TM5 by applying -20% emission perturbations for each of the 56 defined source regions (plus shipping and aviation), for all relevant anthropogenic precursor components, in comparison to a set of unperturbed simulations, hereafter denoted as 'base simulations'. Emissions from biogenic organic components were included as a spatial/temporally varying component, but did not vary in the model sensitivity simulations. Consequently, absolute concentrations of BSOA were identical across base and perturbation simulations and no SR coefficients are available.

A 15 to 20% emission perturbation is commonly used to establish source-receptor emission-concentration sensitivities (Alcamo et al., 1990; Amann et al., 2011; Dentener et al., 2010). The applicability of the established SRs for larger emission perturbations - e.g. in future emission scenario studies - depends on the linearity of the emission-concentration responses, and will be evaluated in detail in section 3.

As elucidated in the previous section, base and perturbed simulations are available on a 1°x1° global resolution. Figures S3.1 and S3.2 in the SI shows some examples of emission perturbation - concentration response grid maps for PM$_{2.5}$, O$_3$ metrics, deposition and column burden for source regions China, India and USA, illustrating clearly the difference in long-range transport characteristics between different species.

For each receptor point $y$ (i.e. each model vertical level 1°x1° grid cell), the change in concentration of component $j$ in receptor $y$ resulting from a -20% perturbation of emitted precursor $i$ in source region $x$, is expressed by a unique SR coefficient $A_{ij}[x,y]$:

$$A_{ij}[x,y] = \frac{\Delta c_j(y)}{\Delta E_i(x)} \text{ with } \Delta E_i(x) = 0.2 E_{i,base}(x) \tag{1}$$

In the present version TM5-FASST_v0, the SR coefficients for pollutant concentrations are stored as annual mean responses to annual emission changes. Individual $PM_{2.5}$ components SRs are stored as dry mass ($\mu g\ m^{-3}$). $PM_{2.5}$ residual water at 35% is optionally calculated a posteriori for sensitivity studies, assuming mass growth factors for ammonium salts of 1.27 (Tang, 1996) and for sea-salt of 1.15 (Ming and Russell, 2001). The presence of residual water in $PM_{2.5}$ is not irrelevant:

epidemiological studies establishing $PM_{2.5}$ exposure-response functions are commonly based on monitoring data of gravimetrically determined $PM_{2.5}$, for which measurement protocols foresee filter conditioning at 30 – 50% RH. As many health impact modelling studies consider dry $PM_{2.5}$ mass or do not provide information on the inclusion of residual water we use dry PM2.5 for health impact assessment in this study for consistency, unless mentioned differently.

We also established SR matrices linking annual emissions to specific $O_3$ exposure metrics that are based on seasonal or

hourly $O_3$ concentrations (e.g. crop exposure metrics based on daytime ozone during crop growing season, human exposure to $O_3$ during highest 6 monthly mean of hourly maximum values). The total concentration of component (or metric) $j$ in receptor region $y$, resulting from arbitrary emissions of *all $n_i$* precursors $i$ at *all $n_x$* source regions $x$, is obtained as a perturbation on the base-simulation concentration, by summing up all the respective SR coefficients scaled with the actual emission perturbation:

$$C_j(y) = C_{j,base}(y) + \sum_{k=1}^{n_x} \sum_{i=1}^{n_i} A_{ij}[x_k, y] \cdot \left[ E_i(x_k) - E_{i,base}(x_k) \right] \tag{2}$$

Pollutants $C_j$ include particulate matter components ($SO_4$, $NO_3$, $NH_4$, BC, particulate organic matter – POM), trace gases ($SO_2$, NO, $NO_2$, $NH_3$, $O_3$), and deposition fluxes of BC, N and S species. In the case of ozone, the $n_i$ precursors in equation (2) would comprise [$NO_x$, NMVOC, CO, $CH_4$]. The set of linear equations (2) with associated source-receptor matrices (1) for all components and all source and receptor regions thus emulates the 'full' TM5-CTM, and constitutes the 'kernel' of

TM5-FASST_v0. When OC emissions are provided in mass units C, the OC mass is multiplied with a factor 1.3 to obtain Particulate Organic Matter (POM) (Kanakidou et al., 2005).

BC and POM are assumed not to interact with other pollutants and their atmospheric lifetimes are prescribed and assumed neither to be affected by mixing with other soluble species like sulfate, nitrate or ammonium salts, nor to undergo oxidation by $O_3$. Recent work (e.g. Huang et al., 2012) indicates that a parameterized approach, as applied in TM5, tends to

underestimate BC and POM atmospheric lifetimes, leading to a low concentration bias. When explicitly modelled, including the combined impact of both mechanisms, Huang et al., 2012 find that the global atmospheric residence times of BC and POM are lengthened by 9% and 3% respectively.

We note that, unlike many other inventories, the RCP emission scenarios do not include a separate inventory for total primary $PM_{2.5}$ which includes besides BC and POM other non-specified primary particulates (e.g. primary sulfate, fly-ash).

When specific scenario studies require so, TM5-FASST_v0 treats this 'other' primary PM.5 (OPP = Primary $PM_{2.5}$ – BC – POM) as BC in Eq. (2), where both $C_{OPP,base}$ and $E_{OPP,base}$ are zero.

$$C_{OPP}(y) = \sum_{n_x} \sum_{n_i} A_{BC}[x, y] \cdot E_{OPP}(x) \tag{3}$$

TM5 Surface ozone (and $NO_2$) fields from base and perturbation experiments were stored at hourly intervals allowing for the calculation of specific vegetation and health related $O_3$ metrics, often based on thresholds of hourly $O_3$ concentrations, or concentrations during daytime. The hourly $O_3$ surface fields were converted into specific $O_3$ metrics responses to annual emissions, including accumulated hourly ozone above a threshold of 40 ppb during a 3 months crop growing season (AOT40), 3-monthly mean of 7 hr or 12 hr daytime ozone during crop growing season (M7, M12), maximum 6-monthly running average of daily maximum hourly $O_3$ (6mDMA1), the sum of daily maximal 8hr ozone mean concentrations above 35ppbV (SOMO35).

The -20% perturbation simulations were performed for the combination of precursors given in Table 2, with P0 the unperturbed reference simulation, and P1 through P5 -20% perturbations for combined or single precursors. Due to limited CPU availability, precursors that are expected not to interact chemically are perturbed simultaneously, with P1 combining $SO_2$, $NO_x$, BC, and POM and P4 combining $NH_3$ and NMVOC. P1 and P4 were computed for each of the 56 continental source regions plus shipping (P1 and P4) and aviation (P1). Additionally, a $SO_2$-only perturbation was computed for all individual source regions and shipping (P2) and $NO_x$-only for a selection of key source regions (P3). Finally a set of combined $NO_x$ + NMVOC perturbation simulations (P5) was performed for a set of key regions.

For a limited set of representative source regions, an additional wider range of emission perturbations $P_i'$ [-80% to +100%] has been applied to evaluate possible non-linearities in the emission-concentration relationships. The list of these additional perturbation simulations is given in Table S3 of the SI. In section 3.1 we explain how this set of perturbation runs is combined into FASST to obtain a complete set of source-receptor matrices for each precursor and source region.

We did not perform dedicated perturbation simulations on $CH_4$ as $O_3$ precursor, but implemented TM5 results obtained in the frame of the first phase of the Hemispheric Transport of Air Pollutants (HTAP1) assessment (Dentener et al., 2010; Fiore et al., 2008). In one of the prescribed experiment set-up, models evaluated how surface ozone levels are responding when the global steady-state $CH_4$ concentration decreases with 20% from 1760 ppbv (the global mean $CH_4$ concentration in the year 2000) to 1408 ppbv. The outcome of this experiment is a set of global grid maps with hourly $O_3$ concentration responses from which all relevant $O_3$ metrics can be obtained. As an example, the annual mean $O_3$ concentration response to the $CH_4$ concentration perturbation is shown in Fig. S3.3 in the SI. Annex S3 in the SI provides more details on the methodology applied to convert the $CH_4$ concentration perturbation into a $CH_4$ emission-based perturbation.

Because of its long life time compared to short-lived ozone precursors, $CH_4$ source-receptor coefficients are considered independent on the location of emission and are therefore provided as global emission-to-regional (or gridded) concentration responses.

Because of the mismatch between the HTAP1 source - receptor regions and the FASST ones, the current version of TM5-FASST does not include source-receptor relations between CO and $O_3$ concentration (or $O_3$ exposure metrics), only impacts

of CO emissions on global methane and $O_3$ global radiative forcing, also in this case retrieved from HTAP1 dedicated CO perturbation experiments with TM5.

Deposition source-receptor matrices of nitrogen and sulfur compounds are obtained in the same way as for the pollutant ambient concentration fields, making the difference between the base and perturbation simulations. Nitrogen depositions are calculated from accumulation of the instantaneous surface budgets of all relevant nitrogen components (NO, $NO_2$, $NO_3$, $2 \times N_2O_5$, $HNO_4$, organic nitrates, $NH_3$, $NH_4$) and similar for sulfur from $SO_2$ and $SO_4$, into monthly time steps. Column amounts of ozone and particulate matter are also computed using 3D monthly output of concentrations and meteorological parameters.

### 2.4    PM2.5 adjustments in urban regions for health impact evaluation

TM5-FASST is specifically aiming at providing pollutant exposure fields for further impact evaluation. For the evaluation of health impacts from outdoor air pollution, a 1°x1° horizontal resolution may not adequately represent sub-grid gradients of pollutants. Indeed, higher pollutant levels are expected to concur with high population density in urban areas, hence an area-averaged concentration for a nominally 100x100km² sized grid cell will underestimate the exposure of population located in pollution hotspots within a single grid cell. We provide a simple parameterization, generating a correction factor on the gridbox area-mean $PM_{2.5}$ concentration, to better represent the actual mean population exposure within that grid cell. In the current approach we only consider $PM_{2.5,}$ although also ozone and $NO_2$ are likely subject to sub-grid gradients. The parameterization is based on the underlying assumption that the spatial distribution of *primary emitted* $PM_{2.5}$ correlates with population density. Our parameterization builds upon high-resolution population grid maps, allowing a sub-grid readjustment of the $PM_{2.5}$ concentration within each 1°x1° grid cell. Further, it needs additional information to flag the population sub-grids as 'urban' or 'rural', e.g. population density for which an urban threshold can be defined, or more sophisticated schemes defining urban areas. We further assume that only primary $PM_{2.5}$ from the residential and the surface transport sectors is contributing to the local (urban) increment, whereas other aerosol precursor components and other sectors are assumed to be homogenously distributed over the 1°x1° grid cell. Indeed, secondary $PM_{2.5}$ is formed over longer time scales and therefore deemed to be more homogeneously distributed at the regional scale, while primary $PM_{2.5}$ emissions from other sources than the residential and transport sector are assumed to occur more remotely from urban areas. The adjusted population-weighted mean concentration within each 1°x1° grid cell (conserving the area-based grid cell mean) is then calculated as follows:

$$PM_{2.5,inc} = DU + SS + SO_4^{2-} + NO_3^- + NH_4^+ + (1-k_{BC})\, BC + (1-k_{POM})\, POM + INCR(k_{BC}\, BC + k_{POM}\, POM) \qquad (4)$$

with DU and SS the fixed natural mineral dust and sea-salt contributions respectively; $SO_4^{2-}$, $NO_3^-$, $NH_4^+$, BC and POM the 1°x1° grid cell average values resulting from TM5 or TM5-FASST; $k_{BC}$ ($k_{POM}$) the fraction of (residential + transport) BC (POM) emissions in the total BC (POM) emissions within the 1°x1° grid cell and *INCR* the urban increment factor. This sub-grid parameterization has been applied as a part of the methodology to estimate population exposure in the Global

Burden of Disease assessments (Brauer et al., 2012). Supplemental Information section S4 provides details on the calculation of *INCR*.

The required gridded sectorial emission data may not be readily available for any assessment. A "default" set of regional population-weighted averaged increment factors for BC and POM is given in Table S4.2, based on the RCP year 2000 baseline simulations performed with TM5 for the year 2000, i.e. using year 2000 population (CIESIN GWPv3) and the RCP year 2000 gridded emissions by sector.

## 2.5     Health impacts

TM5-FASST provides output of annual mean $PM_{2.5}$ and $O_3$ health metrics (3-monthly and 6-monthly mean of daily maximum hourly $O_3$ (3mDMA1, 6mDMA1), and the sum of the maximal 8-hourly mean above a threshold of 35 ppbV (SOMO35) or without threshold (SOMO0), as well as annual mean $NO_x$ and $SO_2$ concentrations at grid resolution of 1°x1°. These are the metrics consistent with underlying epidemiological studies (Jerrett et al., 2009; Krewski et al., 2009; Pope et al., 2002). The population-weighted pollutant exposure metrics grid maps, in combination with any consistent population grid map, are thus available for human health impact assessment. The TM5-FASST_v0 tool provides a set of standard methodologies, including default population and health statistics, to quantify the number of air quality-related premature deaths from $PM_{2.5}$ and $O_3$.

Health impacts from $PM_{2.5}$ are calculated as the number of annual premature mortalities from 5 causes of death, following the Global Burden of Disease methodology (Lim et al., 2012): ischemic heart disease (IHD), chronic obstructive pulmonary disease (COPD), stroke, lung cancer (LC) and acute lower respiratory airways infections (ALRI) whereas mortalities from exposure to $O_3$ are related to respiratory disease.

Cause-specific excess mortalities are calculated at grid cell level using a population-attributable fraction approach as described in Murray et al. (2003) from $\Delta Mort = m_0 \times AF \times Pop$, where $m_0$ is the baseline mortality rate for the exposed population, $AF = (RR-1)/RR$ is the fraction of total mortalities attributed to the risk factor (exposure to air pollution), $RR =$ relative risk of death attributable to a change in population-weighted mean pollutant concentration, and *Pop* is the exposed population (adults $\geq$ 30 years old, except for ALRI for which infant population <5 years old was considered). *RR* for $PM_{2.5}$ exposure is calculated from the Integrated Exposure-Response functions (IER) developed by Burnett et al. (2014), and first applied in e.g. the  Global Burden of Disease study (Lim et al., 2012).

In order to facilitate comparison with earlier studies, TM5-FASST provides as well mortality estimates based on a log-linear exposure response function $RR = \exp^{\beta \Delta PM2.5}$ where *β* is the concentration–response factor (CRF; i.e., the estimated slope of the log-linear relation between concentration and mortality) and $\Delta PM_{2.5}$ is the change in concentration. More details on the health impact methodologies, as well as sources for currently implemented population and baseline mortality statistics and their projections in TM5-FASST_v0 are given in section S5 of the SI.

For O$_3$ exposure, $RR = e^{\beta(\Delta 6mDMA1)}$ , $\beta$ is the concentration–response factor, and $RR$ = 1.040 [95% confidence interval (CI): 1.013, 1.067] for a 10 ppb increase in 6mDMA1 according to Jerrett et al. (2009). We apply a default counterfactual concentration of 33.3 ppbV, the minimum 6mDMA1 exposure level in the Jerrett et al. (2009) epidemiological study.

We note that the coefficients in the IER functions used in the GBD assessments have been recently updated due to methodological improvements in the curve fitting, leading to generally higher RR and mortality estimates (Cohen et al.,2017; Forouzanfar et al., 2016). In particular, the theoretical minimum risk exposure level was assigned a uniform distribution of 2.4–5.9 µg/m$^3$ for PM$_{2.5}$, bounded by the minimum and fifth percentiles of exposure distributions from outdoor air pollution cohort studies, compared to the presently used range of 5.8 - 8.8 µg m$^{-3}$ which would increase the health impact from PM$_{2.5}$ in relatively clean areas. Further, a recent health impact assessment (Malley et al., 2017), using updated RR estimate and exposure parameters from the epidemiological study by Turner et al. (2016), estimates 1.04–1.23 million respiratory deaths in adults attributable to O$_3$ exposure, compared with 0.40–0.55 million respiratory deaths attributable to O$_3$ exposure based on the earlier (Jerrett et al., 2009) risk estimate and parameters. These recent updates have not been included in the current version of TM5-FASST. Health impacts from exposure to other pollutants (NO$_2$, SO$_2$ for example) are currently not being evaluated in TM5-FASST-v0.

## 2.6    Crop impacts

The methodology applied in TM5-FASST to calculate the impacts on four crop types (wheat, maize, rice, and soy bean) is based on Van Dingenen et al. (2009). In brief, TM5 base and -20% perturbation simulations of gridded crop O$_3$ exposure metrics (averaged or accumulated over the crop growing season) are overlaid with crop suitability grid maps to evaluate receptor region-averaged exposure metrics SR coefficients. Gridded crop data (length and centre of growing period, as well as a gridded crop-specific suitability index, based on average climate 1961 – 1990) have been updated compared to Van Dingenen et al. (2009) using the more recent and more detailed Global Agro-Ecological Zones (GAEZ) data set (IIASA and FAO, 2012, available at http://www.gaez.iiasa.ac.at/).

Available crop ozone exposure metrics are 3-monthly accumulated ozone above 40 ppbV (AOT40) and seasonal mean 7 hr or 12 hr day-time ozone concentration (M7, M12) for which exposure-response functions are available from the literature (Mills et al., 2007; Wang and Mauzerall, 2004). Both metrics (M$_i$) are calculated as the 3-monthly mean daytime (09:00 – 15:59 for M7, 08:00 – 19:59 for M12) ozone concentration. AOT40 and M$_i$ are evaluated over the 3 months centred on the midpoint of the location-dependent crop-growing season provided by the GAEZ data set. Note that in the GAEZ methodology, the theoretical growing season is determined based on prevailing temperatures and water balance calculations for a reference crop, and can range between 0 and 365 days, however our approach always considers 3 months as the standard metric accumulation or averaging period.

The crop relative yield loss (RYL) is calculated as linear function from AOT40 and from a Weibull-type exposure-response as a function of Mi:

$$RYL[AOT40] = a \times AOT40 \qquad\qquad\qquad\qquad (5)$$

$$\left.\begin{array}{ll} RYL(M_i) = 1 - \dfrac{exp\left[-\left(\frac{M_i}{a}\right)^b\right]}{exp\left[-\left(\frac{c}{a}\right)^b\right]} & M_i \geq c \\[4mm] \quad RYL(M_i) = 0 & M_i < c \end{array}\right\} \qquad (6)$$

The parameter values in the exposure response functions are given in Table 3. Coefficients $a$ and $b$ are shape factors of the Weibull function, while $c$ represents the lower $M_i$ threshold for visible crop damage. Also here, the non-linear shape of the

RYL(Mi) function requires the ΔRYL for 2 scenarios (S1, S2) being evaluated as $RYL(M_{i,S2}) - RYL(M_{i,1})$, and not as RYL $(M_{i,S2} - M_{i,S1})$.

Finally, it is important to note that TM5-FASST modelled $O_3$ surface concentrations refer to the middle of the TM5's lower layer gridbox, i.e. 30m above surface, whereas monitoring of $O_3$ (from which exposure metrics are derived) actually happens at a standard altitude of 3 to 5m above the surface where, due to deposition and meteorological processes, the concentration

may differ. However comparing TM5 simulated gridbox-centre ozone metrics with observations from 99 monitoring stations worldwide, Van Dingenen et al. (2009) find that, when averaged at the regional scale, TM5 simulated crop metrics obtained from the grid box centre are reproducing the observations within their standard deviations, and that the monthly 10m TM5 metric values do not significantly improve the bias between model and observations. Therefore we use the standard model output at 30m.

**2.7     Climate metrics**

We make use of the available 3D aerosol and $O_3$ fields in the -20% emission perturbation simulations with TM5 to derive the change in global forcing for each of the perturbed emitted precursors. The region-to-global radiative forcing SR for precursor $j$, emitted from region $k$ $\left(SR\_RF_k^j\right)$ is calculated as the emission-normalized change in global radiative forcing between the TM5 base and the corresponding -20% emission perturbation experiment:

$$SR\_RF_k^j = \frac{RF\_PERT[j,k] - RF\_BASE}{0.2E_k^j} \text{ [W/m}^2\text{]/[kg/yr]} \qquad\qquad (7)$$

where RF_PERT and RF_BASE are the TM5 global radiative forcings for the perturbation and base simulations respectively, and $E_k^j$ is the annual base emission of precursor $j$ from region $k$.

For each emitted pollutant (primary and secondary) the resulting normalized global forcing responses are then further used to

calculate the global warming potential (GWP) and global temperature potential (GTP) for a series of time horizons H. In this way, a set of climate metrics is calculated with a consistent methodology as the air quality metrics, health and ecosystem impacts calculated from the concentration and deposition fields. In this section we describe in more detail the applied

methodologies in TM5 to obtain the radiative forcing from aerosols, clouds and gases, as well as the derivation of the GWP and GTP metrics.

### 2.7.1    Instantaneous radiative forcing by aerosols

The base simulation and -20% perturbation response of the column-integrated aerosol mass over all 25 vertical layers of TM5 for all relevant species was calculated and stored. The calculation of the top-of-atmosphere (TOA) instantaneous forcing by aerosol is based on the radiative transfer model described by Marmer et al. (2007) using monthly average meteorological fields and surface characteristics using ECMWF monthly average meteorological fields (temperature, clouds, relative humidity, surface albedo) for the year 2001. We assume externally mixed aerosols and calculate the forcing separately for each component. The total aerosol forcing is obtained by summing up these contributions. We refer to section S6 of the SI for a more detailed description of the forcing calculations. To avoid further extensive radiative transfer calculation, monthly-mean radiative forcing efficiencies, expressed as [W/m²]/[mg], were calculated once using the 1°x1° gridded TM5 base simulation outputs and off-line radiative code using monthly fields of aerosol, ECMWF meteorology and surface characteristics, and stored for further use (Marmer et al., 2007). The annual TOA global forcing for each scenario is then obtained by multiplying the monthly column-integrated aerosol mass with this grid-cell specific monthly mass forcing efficiency and subsequently averaged over one year. Neglecting the aerosol mixing state and using column-integrated mass rather than vertical profiles introduces additional uncertainties in the resulting forcing efficiencies. Accounting for internal mixing may increase the BC absorption by 50 to 200% (Bond et al., 2013), whereas including the vertical profile would weaken BC forcing and increase $SO_4$ forcing (Stjern et al. 2016). Further, the BC forcing contribution through the impact on snow and ice is not included, nor are semi- and indirect effects of BC on clouds. Our evaluation of pre-industrial to present radiative forcing in the validation section demonstrates that, in the context of the reduced-form FASST approach, the applied method however provides useful results. Figure S6.1 (a, b, c) in the SI shows the resulting global radiative forcing fields for sulfate, POM and BC. The regional emission-normalized forcing SRs for aerosol precursors (in W m$^{-2}$ Tg$^{-1}$) are given in Table S6.2 of the SI.

### 2.7.2    Indirect aerosol forcing

Aerosols modify the microphysical and radiative properties and lifetime of clouds, commonly denoted as the aerosol indirect effect (Haywood and Boucher, 2000). This forcing results from the ability of the hydroscopic particles to act as (warm) cloud condensation nuclei thus altering the size, the number and the optical properties of cloud droplets (Twomey, 1974). More and smaller cloud droplets increase the cloud albedo, which leads to cooling. Using TM5 output, indirect forcing is evaluated considering only the so far best studied first indirect effect, and using the method described by Boucher and Lohmann (1995). Fast feedbacks on cloud lifetimes and precipitation were not included in this off-line approach. This

simplified method uses TM5 3D time-varying fields of $SO_4$ concentrations, cloud liquid water content, and cloud cover (the latter from the parent ECMWF meteorological data). The parameterization uses the cloud information (liquid water content and cloud cover) from the driving ECMWF operational forecast data (year 2001). Fast feedbacks on cloud lifetimes and precipitation were not included in this off-line approach. The cloud droplet number concentrations and cloud droplet effective radius were calculated following Boucher and Lohmann (1995) separating continental and maritime clouds. The equations are given in section S6 of the SI. The global indirect forcing field associated with sulfate aerosols is shown in Fig. S6.1(d) of the SI and regional forcing SRs are listed in Table S6.2. Indirect forcing by clouds remains however highly uncertain, and although FASST evaluates its magnitude, it is often not included in our analyses.

### 2.7.3 Radiative forcing by $O_3$ and $CH_4$

Using TM5 output, radiative forcing (RF) by ozone is approximated using the forcing efficiencies obtained by the STOCHEM model as described in Dentener et al. (2005), normalized by the ozone columns obtained in that study. Here we use annual averaged forcing based on the RF computations provided as monthly averages by D. Stevenson (personal communication, 2004). The radiative transfer model was based on Edwards and Slingo (1996). These forcings account for stratospheric adjustment, assuming the fixed dynamical heating approximation, which reduces instantaneous forcings by ~22%.

For $CH_4$ the RF associated with the base simulation was taken from the equations in the IPCC-Third Assessment Report (TAR) (Table 6.2 of Ramaswamy et al., 2001). Using the HTAP1 calculated relationship between $CH_4$ emission and concentration (see section S3.1 in the SI), we evaluated a globally uniform value of 2.5 mW/m² per Tg $CH_4$ emitted. It includes both the direct $CH_4$ greenhouse gas (GHG) forcing (1.8 mW/m²) as well as the long-term feedback of $CH_4$ on hemispheric $O_3$ (0.7 mW/m²). From the TM5 perturbation experiments we derive as well region-to-global radiative forcing SRs (expressed as [W m$^{-2}$]/[Tg yr$^{-1}$]) for precursors $NO_x$, NMVOC , $SO_2$ and CO (the latter taken from HTAP1 experiments) through their feedback on the $CH_4$ lifetime and subsequently on long-term hemispheric $O_3$ levels. Hence, the greenhouse gas radiative forcing contribution of each ozone precursor consists of 3 components: a direct effect through the production of $O_3$, a contribution by a change in $CH_4$ through modified OH levels (including a self-feedback factor accounting for the modified $CH_4$ lifetime), and a long-term contribution via the feedback of $CH_4$ on hemispheric ozone. The details of the applied methodology for direct and indirect $CH_4$ forcing SRs are given in section S6.2 of the SI, including tables with the regional emission-based forcing efficiencies for all precursors (Tables S6.3 to S6.5).

In its current version, TM5-FASST_v0 provides the steady-state concentrations and forcing response of the long-term $O_3$ and $CH_4$ feedback of sustained precursor emissions, i.e. it does not include transient computations that take into account the time lag between emission and establishment of the steady-state concentration of the long-term $O_3$ and $CH_4$ responses.

### 2.7.4   Calculation of GWP, GTP, delta T and $CO_{2eq}$ emissions

The obtained emission-based forcing efficiencies (Tables S6.2 to S6.5 in the SI) are immediately useful for evaluating a set of short-lived climate pollutant climate metrics. Applying the methodology described by Fuglestvedt et al. (2010) briefly outlined below, the resulting emission-normalized specific forcing responses $A_x$ [W/m²]/[kg/year] are used to calculate the absolute global warming potential (AGWP) and absolute global temperature potential (AGTP) for various time horizons H (20, 50, 100, 500 yr), as a basis to obtain the corresponding $CO_{2eq}$ for the actually emitted amounts.

The AGWP for emitted short-lived (exponentially decaying) species $x$ with lifetime $a_x$ is calculated by integrating the specific forcing over a time span H of an emission pulse at t=0:

$$AGWP(H) = \int_0^H A_x \exp\left(\frac{-t}{a_x}\right) dt = A_x a_x \left[1 - \exp\left(\frac{-H}{a_x}\right)\right] \tag{8}$$

AGTP of a short-lived (exponentially decaying) component is calculated as an endpoint change in temperature after H years from a one-year emission pulse at time 0.

$$AGTP(H) = \int_0^H A_x \exp\left(\frac{-t}{a_x}\right) R(H - t) dt \tag{9}$$

where R(t) represents the response in global-mean surface temperature to a unit pulse in forcing. Following Fuglestvedt et al. (2010) we adopt the functional form for R(t) from Boucher and Reddy (2008), derived from a GCM :

$$R(t) = \sum_{j=1}^2 \frac{c_j}{d_j} \exp\left(-\frac{t}{d_j}\right) \tag{10}$$

The first term in the summation can crudely be associated with the response of the ocean mixed-layer to a forcing, the second term as the response of the deep ocean with $c_j$ [ K(Wm$^{-2}$)$^{-1}$] and $d_j$ [years] represent temperature sensitivity and response time of both compartments respectively. This leads to:

$$AGTP(H) = \sum_{j=1}^2 \frac{A_x a_x c_j}{(a_x - d_j)} \left( \exp\left(\frac{-H}{a_x}\right) - \exp\left(\frac{-H}{d_j}\right) \right) \tag{11}$$

As discussed earlier, we take into account that species such as $NO_x$, NMVOC and CO lead to changes in $O_3$ and $CH_4$ and consequently have a short-lived component ($O_3$) as well as long-lived components ($CH_4$ and $CH_4$-induced $O_3$) contributing to AGWP and AGTP. We refer to Appendix 2 in Fuglestvedt et al., 2010 for a detailed description of the methodology and numerical values for $c_j$ and $d_j$. As aerosols and directly produced $O_3$ from ozone precursors have a lifetime of the order of days (aerosols) to several months ($O_3$), the resulting integrated specific forcing is insensitive to the actual lifetime for the range of time horizons considered (decades to centuries), and in practice we use a default value of 0.02yr for aerosols and 0.27yr for short term $O_3$. This does however not apply to the long-term forcing contribution of $CH_4$ and the associated $O_3$ feedback from $O_3$ precursors for which we use a perturbation adjustment time of 14.2 years (Wild et al., 2001). Note that this adjustment time scale is larger than the total atmospheric time scale for $CH_4$ oxidation by OH combined with losses to soils and the stratosphere (HTAP1 model ensemble mean: 8.8 years (Fiore et al., 2009)) due to the feedback of $CH_4$ on atmospheric OH concentrations and thereby its own lifetime (Forster et al., 2007). Fuglestvedt et al. (2010) report $CH_4$

adjustment times from various modelling studies between 10.2 and 16.1 years. Dimensionless metrics GWP (GTP) are obtained dividing AGWP (AGTP) by the AGWP (AGTP) of $CO_2$ as a reference gas for which we use values from Joos et al. (2013).

Finally, still following Fuglestvedt et al. (2010), we also include a calculation of the global temperature change $\Delta T_x(H)$ between year 0 and year H for a sustained emission change $\Delta E_x(t) = E_x(t) – E_x(0)$ of component $x$ as the sum of the delta T from one-year emission 'pulses' approaching the time horizon.

$$\Delta T_x(H) = \sum_{t=0}^{H} \Delta E_x(t) AGTP(H - t) \qquad (12)$$

In this way, a set of climate metrics is obtained which is consistent with the air quality metrics, health and ecosystem impacts calculated from the concentration and deposition fields.

## 3    Results: validation of the reduced-form TM5-FASST

In this section we focus on the validation of regionally aggregated TM5-FASST_v0 outcomes (pollutant concentrations, exposure metrics, impacts), addressing specifically:

(1)  The additivity of individual pollutant responses as an approximation to obtain the response to combined precursor perturbations,

(2)  The linearity of the emission responses over perturbation ranges extending beyond the -20% perturbation

(3)  The FASST outcome versus TM5 for a set of global future emission scenarios that differ significantly from the reference scenario

(4)  FASST key-impact outcomes versus results from the literature for some selected case studies, with a focus on climate metrics, health impacts and crops.

## 3.1    Validation against the full TM5 model: additivity and linearity

We recall that the TM5-FASST computes concentrations and metrics based on a perturbation approach, i.e. the linearization applies only on the difference between scenario and reference emission. Therefore we focus on evaluating the perturbation response, i.e. the second term in the right hand side of Eq. 2.

The standard set of -20% emission perturbation simulations, available for all 56 continental source regions and constituting the kernel of TM5-FASST_v0 are simulations P1 (perturbation of $SO_2$, $NO_x$, BC and POM), P2 ($SO_2$ only), and P4 ($NH_3$ and NMVOC) shown in Table 2. Additional standard -20% perturbation experiments P3 ($NO_x$ only) and P5 ($NO_x$ and NMVOC), as well as an additional set of perturbation simulations P1' to P5' over the range [-80%, +100%], listed in Table S3 of the SI, have been performed for a limited selection of representative source regions (Europe, USA, China, India, Japan) due to limited CPU resources. For the same reason, no combined perturbation studies are available for ($SO_2$ + $NH_3$) and ($NO_x$ + $NH_3$) for a systematic evaluation of additivity and linearity. The available [-80%, +100%] perturbations are used to validate

the linearized reduced-form approach against the full TM5 model, exploring chemical feedback mechanisms (additivity) and extrapolation of the -20% response sensitivity towards larger emission perturbation magnitudes (linearity). This is in particular relevant for the $NO_x$ - NMVOC - $O_3$ chemistry and for the secondary $PM_{2.5}$ components $NO_3^-$ - $SO_4^{2-}$ - $NH_4^+$. These mechanisms could also be important for organic aerosol, but we remind that in this study organic aerosol formation was parameterized as pseudo-emissions.

### 3.1.1 Additivity and linearity of secondary inorganic $PM_{2.5}$ response:

Experiment P1, where BC, POM, $SO_2$ and $NO_x$ emissions are simultaneously perturbed by -20% relative to base simulation P0, delivers SR matrices for primary components BC and POM, and a first-order approximation for the precursors $SO_2$ and $NO_x$ whose emissions do not only affect $SO_2$ and $NO_x$ gas concentrations but also lead to several secondary products ($SO_2$ forms ammonium sulfate, $NO_x$ leads to $O_3$ and ammonium nitrate). Experiment P2 perturbs $SO_2$ only, whereas experiment P3 perturbs $NO_x$ only (in this latter case, to limit the computational cost, computed for a limited set of representative source regions only).

We first test the hypothesis that the $PM_{2.5}$ response to the combined ($NO_x$ + $SO_2$) -20% perturbation (P1) can be approximated by the sum of the single precursor perturbations responses (P2 + P3). Figure 2 summarizes the resulting change in $SO_4^{2-}$, $NO_3^-$, $NH_4^+$ and total inorganic $PM_{2.5}$ respectively for the selected source regions. For Europe, the emission perturbations were applied over all European countries simultaneously, hence the responses are partly due to inter-regional transport from other countries. Following findings result from the perturbation experiments P1, P2 and P3:

(1) Sulfate shows a minor response to $NO_x$ emissions, and likewise nitrate responds only slightly to $SO_2$ emissions and both perturbations are additive. In general the response is one order of magnitude lower than the direct formation of $SO_4^{2-}$ and $NO_3^-$ from $SO_2$ and $NO_x$ respectively (Fig. 2a, b);

(2) $NH_4$ responds to $NO_x$ and $SO_2$ emissions with comparable magnitudes and in an additive way (Fig. 2c);

(3) The response of total sulfate, nitrate and ammonium to a combined $NO_x$ and $SO_2$ -20% perturbation can be approximated by the sum of the responses to the individual perturbations, i.e. P1 ≈ P2+P3 (Fig. 2d). Scatterplots of P1 versus P2+P3 responses for the regional averaged individual secondary products and total inorganic $PM_{2.5}$ are shown in Fig. S7.1 of the SI.

From the combined [$SO_2$+$NO_x$] perturbation (P1), and the separate $SO_2$ perturbation simulations (P2), both available for all source regions, the missing $NO_x$ SR matrices have been gap-filled using (P1 – P2). By lack of simulations for combined ($SO_2$ + $NH_3$) or ($NO_x$ + $NH_3$) perturbations we assume additivity for simultaneous $NH_3$, $SO_2$ and $NO_x$ perturbations, i.e. the response is computed from a linear combination of P2, P3 and P4.

Next we evaluate the hypothesis that the -20% perturbation responses can be extrapolated towards any perturbation range, as an approximation of a full TM5 simulation. Figure 3 shows, for the selected regions listed in Table S3 of the SI, the TM5

computed relative change in secondary $PM_{2.5}$ concentration versus the relative change in precursor emission in the range [-80%, +100%]. The figure illustrates the general near-linear behaviour of regionally aggregated responses to single precursor emission perturbations for all regions, except for India where the linearity of the response to $NO_x$ emissions breaks down for emission reductions beyond -50%. For India we further observe a relative strong nitrate response to $NO_x$ emissions, with $NO_3^-$ increasing by a factor of 3 for a doubling of $NO_x$ emissions, although the responses shown in Fig. 2 indicate that absolute changes (in µg m$^{-3}$) in $NO_3$ are relatively low and that secondary $PM_{2.5}$ in this region is dominated by $SO_4$. We are not aware of reliable observations or other published $NO_x$-aerosol sensitivity studies from that region that could corroborate this calculated sensitivity. Because such a feature may strongly affect projected future $PM_{2.5}$ levels and associated impacts, we recommend regional multi-model studies devote attention this feature.

Because the TM5-FASST linearization is based on the extrapolation of the -20% perturbation slope, concave-shaped trends in Fig. 3 indicate a tendency of TM5-FASST to over-predict secondary $PM_{2.5}$ at large negative or positive emission perturbations, and opposite for convex-shaped trends. Figure 4 illustrates the error introduced in regional secondary $PM_{2.5}$ concentrations responses when linearly extrapolating the regional -20% perturbation sensitivities to -80% (blue dots) and +100% (red dots) perturbations respectively. While the scatter plots for the single perturbations (Fig. 4 a,b,c) evaluate the linearity of the single responses, the panel showing the combined ($SO_2+NO_x$) perturbation (Fig. 4d) is a test for the linearity combined with additivity of $SO_2$ and $NO_x$ perturbations over the considered range. In general, the linear approximation leads to a slight over-prediction of the resulting secondary $PM_{2.5}$ (i.e. the sum of sulfate, nitrate and ammonium) for all regions considered, in either perturbation direction. Table 4 shows regional statistical validation metrics (normalized mean bias NMB [%], mean bias MB [µg m$^{-3}$], and correlation coefficient, definitions are given in the Table Notes) for the grid-to-grid comparison between TM5-FASST and TM5-CTM of the response to the [-80%, 100%] perturbation simulations (with Europe presented as a single region). In terms of NMB, the FASST linearisation performs worst for the $NO_x$ perturbations, with almost a factor 2 overestimate in Japan for an emission doubling. However, because of the already low $NO_x$ emissions in this region, the absolute error (MB) remains below 0.2µg m$^{-3}$. In all considered perturbation cases, FASST shows a positive MB, except for the $NO_x$ perturbation in India. In general, the highest NMB are observed for the regions where secondary $PM_{2.5}$ shows low response sensitivity to the applied perturbations and where the impact on the total $PM_{2.5}$ is therefore relatively low. Indeed, when considering the total resulting secondary $PM_{2.5}$ (i.e. the full right-hand side of Eq. 2, including the $PM_{2.5}$ base-concentration term containing primary and secondary components), regional averaged FASST secondary $PM_{2.5}$ values stay within 15% of TM5 (see Table S7.1of the SI). A break-down for the individual receptor regions within the European zoom region of the linearisation error on the resulting total secondary $PM_{2.5}$ from individual and combined precursor perturbations is shown in Fig. S7.3 of the SI.

### 3.1.2     Additivity and linearity of $O_3$ responses to combined precursor emissions

$O_3$ atmospheric chemistry is in general highly non-linear, displaying a response magnitude and sign depending on the concentration ratio of its two main ozone precursors $NO_x$ and NMVOC, with high VOC/$NO_x$ ratios corresponding to $NO_x$-sensitive chemistry and low VOC/$NO_x$ ratios corresponding to VOC-sensitive chemistry (Seinfeld and Pandis, 1998; Sillman, 1999). Because the $NO_x$/NMVOC ratio determines the $O_3$ response to emission changes, a perturbation with simultaneous $NO_x$ and NMVOC emission changes of the same relative size is expected to behave more linearly than single perturbations since the chemical regime remains similar. The FASST reduced-form approach builds on the assumption that the $O_3$ response to combined precursor perturbation can be approximated by the sum of the single component emission perturbations (additivity hypothesis). This is in particular relevant for combined and individual $NO_x$ and NMVOC perturbations, and to a less extend for the ($SO_2$, $NO_x$) combination.

Although the impact of $SO_2$ chemistry on $O_3$ is low, for gap-filling purposes we first evaluate the additivity hypothesis for the combined ($SO_2$ + $NO_x$) perturbation. Comparing experiments P1 ($SO_2$ + $NO_x$ perturbation), P2 ($SO_2$ perturbation) and P3 ($NO_x$ perturbation) confirms that the ozone response to $SO_2$ emissions is marginal and additive to the response to $NO_x$ (P1 ≈ P2+P3) over the full range of perturbations, as shown in Fig. S7.2 in the SI, and hence we can gap-fill the missing $NO_x$ perturbation SR matrix for all source and receptor regions from P3 ≈ P1 - P2

Next, we evaluate whether the $O_3$ response to the combined $NO_x$ + NMVOC perturbation (P5) can be approximated by the sum of $O_3$ responses to individual $NO_x$ (P3) and NMVOC (P4) perturbations, i.e. assuming P5 = P4 + P3. P5 was obtained for a limited set of representative source regions: Europe (by perturbing precursor emissions from all FASST source regions inside the EUR master zoom region simultaneously), China, India and USA. As shown in Fig. 5, for the -20% perturbations we find good agreement between the combined ($NO_x$ + NMVOC) perturbation (open circles) with the sum of the individual precursor perturbation (black dots). This occurs even in situations where titration by $NO_2$ causes a reverse response in $O_3$ concentration as is the case in most of Europe and the USA, indicating that a -20% perturbation in individual precursors appears not to change the prevailing $O_3$ regime. However extending the $O_3$ (and metrics) linearized responses as a sum of scaled individual -20% precursor responses towards more extreme perturbation ranges could be a challenge, as the individual perturbation of one of the ($NO_x$, NMVOC) precursor may change the ozone formation regime. In particular during winter time, titration of $O_3$ under high $NO_x$ conditions may reverse the slope of the $NO_x$ emission – $O_3$ concentration response. On the other hand, the impact-relevant $O_3$ metrics, both health and crop related, are based on summertime and daytime values and are expected to be less affected by titration and consequently to maintain a positive emission-response slope (Wu et al., 2009).

Figure 6 shows that, while the response to NMVOC (with constant $NO_x$) is near-linear and monotonically increasing over the full range for all regions, the $NO_x$ response (with constant NMVOC) is showing a more complex behaviour, exhibiting a negative slope for annual mean $O_3$ over nearly all European regions and the USA, whereas the slope is positive for India and

China. For the health-relevant exposure metric 6mDMA1 and the crop metric M12 the slope is positive in most regions, due to their implicit constraint to the summer season when titration plays a minor role, except in strongly $NO_x$-polluted North-Western European countries (Great Britain, Germany, Belgium and The Netherlands, as well as Finland) where titration in large urbanized areas remains important even during summer. The concave shapes of the response curves indicate significant non-linearities, in particular for responses of crop and health exposure metrics to strong $NO_x$ emission perturbations.

Figure 7 illustrates the performance of the TM5-FASST approach versus TM5 for regional-mean annual mean ozone, health exposure metric 6mDMA1 (both evaluated as population-weighted mean), and for the crop-relevant exposure metrics AOT40 and M12 (both evaluated as area-weighted mean) over the extended emission perturbation range. In most cases the response (i.e. the *change* between base and perturbed case) to emission perturbations lies above the 1:1 line across the 4 metrics, indicating that FASST tends to over-predict the resulting metric (as a sum of base concentration and perturbation). Of the four presented metrics, AOT40 is clearly the least robust one, which can be expected for a threshold-based metric that has been linearized. Tables 5 to 7 give the statistical metrics for the grid-to-grid comparison of the perturbation term between FASST and TM5 for the health exposure metric 6mDMA1, and crop exposure metrics AOT40 and M12 respectively. Statistical metrics for the total absolute concentrations (base concentration + perturbation term) are given in Tables S7.2 to S7.4 in the SI. As anticipated, the $NO_x$-only perturbation terms are showing the highest deviation, in particular for a doubling of emissions, however combined $NO_x$-NMVOC perturbations are reproduced fairly well for all regions, staying within 33% for a -80% perturbation for all 3 exposure metrics, and within 38% for an emission doubling for 6mDMA1 and M12, while the AOT40 metric is overestimated by 76 to 126% for emission doubling. The total resulting concentration over the entire perturbation range for single and combined $NO_x$ and NMVOC perturbation agrees within 5% for 6mDMA1 and M12, and within 64% for AOT40. The mean bias is positive for both perturbations, for all metrics and over all analysed regions, except for crop metric M12 under a doubling of NMVOC emissions over Europe showing a small negative bias. The deviations for individual European receptor regions under single and combined NMVOC and $NO_x$ perturbations for health and crop exposure metrics are shown in Figs. S7.4 to S7.6 of the SI.

### 3.2 TM5-FASST_v0 versus TM5 for future emission scenarios

In this section we evaluate different combinations of precursor emission changes relative to the base scenario in a global framework. We take advantage of available TM5 simulations for a set of global emission scenarios which differ significantly in magnitude from the FASST base simulation, and as such provide a challenging test case to the application of the linear source-receptor relationships used in TM5-FASST. We assume that the full TM5 model provides valid evaluations of emission scenarios, and we test to what extent these simulations can be reproduced by the linear combinations of SRs implemented in the TM5-FASST_v0 model.

We use a set of selected policy scenarios prepared with the MESSAGE integrated assessment model in the frame of the Global Energy Assessment GEA (Rao et al., 2012, 2013; Riahi et al., 2012). These scenarios are the so called "frozen

legislation" and "mitigation" emission variants for the year 2030 (named FLE2030, MIT2030 respectively), policy variants that describe two different policy assumptions on air pollution until 2030. These scenarios and there outcomes are described in detail in Rao et al. (2013), the scope of the present study is the inter-comparison between FASST and TM5 resulting pollutant concentration and exposure levels, as well as associated health impacts.

Major scenario features and emission characteristics are provided in section S8 of the SI. Table S8.1 shows the change in global emission strengths for the major precursors for both test scenarios, relative to the RCP2000 base, aggregated to the FASST 'master zoom' regions listed in Table S2.2. Emission changes for the selected scenarios mostly exceed the 20% emission perturbation amplitude from which the SRs were derived. Under the MIT2030 low emission scenario, all precursors and primary pollutants (except primary $PM_{2.5}$ in East-Asia and $NH_3$ in all regions) are showing a strong decrease

compared to the RCP2000 reference scenario. The strongest decrease is seen in Europe ($NO_x$: -83%, $SO_2$: -93%, BC: -89%, primary $PM_{2.5}$ − 56%) while $NH_3$ is increasing by 14 to 46% across all regions. The FLE2030 scenario displays a global increase for all precursors, however with heterogeneous trends across regions. In Europe, North-America and Australia, the legislation in place, combined with use of less and cleaner fuels by 2030, leads to a decrease in pollutant emissions except for $NH_3$ and primary $PM_{2.5}$. On the other hand, very substantial emission increases are projected in East and South-East for

BC, $NO_x$ and primary $PM_{2.5}$. Anticipating possible linearity issues, we note that for both scenarios, in all regions, $SO_2$ and $NO_x$ emissions are evolving in the same direction, although not always with similar relative changes, whereas $NH_3$ is always increasing, which may induce linearity issues in the ammonium-sulfate-nitrate system. Regarding $O_3$ metrics, NMVOC and $NO_x$ are evolving in the same direction, but also here we observe possible issues due to a changing emission ratio (in particular in Russia and Asia). We further note that not only the emission levels of these scenarios are different from the

FASST base scenario (RCP year 2000), but also the spatial distribution of the emissions, at the resolution of grid cells, may differ from the reference set.

We use FASST to compute $PM_{2.5}$ and ozone concentrations applying Eq. (2), i.e. considering the FLE2030 and MIT2030 emission scenarios as a perturbation on the FASST reference emission set (RCP year 2000).

The scope of TM5-FASST is to evaluate on a regional basis the impacts of policies that affect emissions of short-lived air

pollutants and their precursors. Hence we average the resulting $O_3$ and $PM_{2.5}$ concentration and $O_3$ exposure metric 6mDMA1 over the each of the 56 FASST regions and compare them with the averaged TM5 results for the same regions.

Further, in a policy impact analysis framework, the *change* in pollutant concentrations between two scenarios (e.g. between a reference and policy case) is often more relevant than the absolute concentrations. We therefore present absolute concentrations as well as the change (delta) between the two GEA scenarios, evaluating the benefit of a mitigation scenario

versus the frozen legislation scenario.

Figure 8 shows the FASST versus TM5 regional scatter plots for absolute and delta population-weighted mean anthropogenic $PM_{2.5}$ for all 56 FASST receptor regions while the population-weighted means over the 9 larger zoom areas

are shown in Figure 9. Similarly annual mean population-weighted $O_3$ and 6mDMA1 scatter plots are shown in Fig. 10, and the regional distribution in Fig. 11. The grid-cell statistics (mean, NMB, MB and $R^2$) over larger zoom areas are given in Tables 8 and 9 for $PM_{2.5}$ and 6mDMA1 respectively.

Figure 8 and Table 8 show that on a regional basis, the low emission scenario generally overestimates population-weighted $PM_{2.5}$ concentrations, with the highest negative bias in Europe and Asia, while the lowest deviation is found in Latin America and Africa. The agreement between FASST and TM5 is significantly better for the high emission scenario, in line with the findings in the previous section. As shown in Table 8, averaged over the larger zoom regions, we find that the relative deviation for $PM_{2.5}$ is within 11% for FLE2030, and within 28% for MIT2030, except for Europe where the (low) $PM_{2.5}$ concentration is overestimated by almost a factor of 2. The policy-relevant delta between the scenarios however is for all regions reproduced within 23%.

The ozone health metric 6mDMA1 is more scattered than annual mean ozone, and also here, as expected, the low emission scenario performs worse than the high emission one. Over larger zoom areas however the agreement is acceptable for both scenarios (FASST within 22% of TM5). Contrary to $PM_{2.5}$, the NMB for the delta 6mDMA1 between two scenarios is higher than the NMB on absolute concentrations, with a low bias for the delta metric of -38% and -45% for Europe and North-America respectively, and a high bias of 35 to 46% in Asia. However, the MB on the delta is of the same order or lower than the absolute concentrations (Table 9). This is a consequence of the fixed background ozone in the absolute concentration reducing the weight of the anthropogenic fraction in the relative error.

Figures 9 and 11 provide a general picture of the performance of FASST: despite the obvious uncertainties and errors introduced with the FASST linear approximation for large emission changes compared to the RCP base run, at the level of regionally aggregated concentrations, a consistent result emerges both for absolute concentrations from the individual scenarios as for the policy-relevant delta.

A major issue in air pollution or policy intervention impact assessments is the impact on human health; therefore we also evaluate the TM5-FASST outcome on air pollution premature mortalities with the TM5-based outcome, applying the same methodology on both TM5 and FASST outcomes. We evaluate mortalities from $PM_{2.5}$ using the IER functions (Burnett et al., 2014) and $O_3$ mortalities using the log-linear ER functions and RR's from Jerrett et al. (2009) respectively. Figure 12 ($PM_{2.5}$) and Fig. 13 ($O_3$) illustrate how FASST-computed mortalities compare to TM5, both as absolute numbers for each scenario, as well as the delta (i.e. the health benefit for MIT2030 relative to FLE2030). Regional differences in premature mortality numbers are mainly driven by population numbers. In line with the findings for the exposure metrics ($PM_{2.5}$ and 6mDMA1) FASST in general over-predicts the absolute mortality numbers, in particular in the low-emission case. For MIT2030, global $PM_{2.5}$ mortalities are overestimated by 19%, in Europe and North-America FASST even by 43%. In the FLE2030 case, we find a better agreement, with a global mortality over-prediction of 3% (for Europe and North-America 5% and 11% respectively). For the latter scenario, the highest deviation is found in Latin America (10 – 20%). $O_3$ mortalities are

overestimated globally by 11% (7%) with regional agreement within 20% (14%) for MIT2030 (FLE2030). However, as shown by the error bars, the difference between FASST and TM5 is smaller than the uncertainty on the mortalities resulting from the uncertainty on RR's only. The potential health benefit of the mitigation versus the non-mitigation scenario (calculated as FLE2030 minus MIT2030 mortalities) is shown in Figs. 12c and 13c. Globally, FASST underestimates the reduction in global $PM_{2.5}$ mortalities by 17% with regional deviations ranging between -30% for Europe and North-America, and -12% for India. The global health benefit for ozone is underestimate by 2% for $O_3$, however as a net result of 11% overestimation in India and 12 to 59% underestimation in the other regions. The numbers corresponding to Figs. 12 and 13 are provided in Table S8.4 and S8.5 of the SI.

The error ranges presented here are obviously linked to the choice of the test scenarios and will for any particular scenario depend on the magnitude and the relative sign of the emission changes relative to RCP2000, but given the amplitude of the emission change for the currently two selected scenarios relative to RCP2000, these results support the usefulness of TM5-FASST as a tool for quick scenario screening.

### 3.3 Comparison of TM5-FASST_v0 impact estimates with published studies

In this section we evaluate TM5-FASST_v0 outcomes for a number of key impacts (climate metrics, human health and O3 damage to crops) with results from earlier studies in the literature.

### 3.3.1 Year 2000 total global anthropogenic forcing by component

The most widely published radiative forcing estimates compare the present-day with the pre-industrial time. To simulate pre-industrial, for simplicity in our TM5-FASST_v0 evaluation we set all anthropogenic in the base simulation (RCP year 2000) to zero and calculate the change in forcing compared to the base case. We include forcing from all aerosol components, as well as $CH_4$ (including its feedback on $O_3$) and the short and long term forcing impacts of $NO_x$, NMVOC and CO on ozone and the methane lifetime. Figure 14 shows the anthropogenic forcings derived from TM5-FASST by emitted component, together with results from AR5 (year 1750-2011). We find that, except for BC, TM5-FASST_v0 reproduces, within the uncertainties reported by IPCC AR5, the global forcing values by emitted component. Only our estimated BC forcing (0.15 W/m²) falls just outside the AR5 90% confidence interval (0.23, 1.02) W/m², which can be partly explained by the different emission years used in the inter-comparison (also explaining the relatively low estimate for $CH_4$). However, comparing to another widely used literature source (Bond et al., 2013), the TM5-FASST_v0 BC forcing estimate still falls within the 90% CI (0.08, 1.27) W/m² direct radiative forcing given for the year 2005, with a comparable global BC emission rate. Our low-end BC forcing estimate can be partly explained by the simplified treatment as externally mixed aerosol, without accounting for the enhancement of the mass absorption cross-section when BC particles become mixed or coated with scattering components. Not-included snow albedo and indirect cloud effects would contribute with +0.13 (+0.04 to +0.33) W/m² and +0.23 (-0.47 to +1.0) W/m² respectively (Bond et al., 2013).

A break-down of the forcing contributions of each emitted pollutant to aerosol, ozone (including immediate and long-term response modes) and methane (when applicable) forcing is given in Table S6.6 of the SI, together with the respective AR5 central values. Although there are very large uncertainties associated with the estimates of the indirect aerosol effect due to the strong approximations made in this work, the calculated magnitude (-0.81 W/m²) is in agreement with the published literature range -0.55 W/m² 90% CI (-1.33, -0.06) W/m².

Table 10 compares the contribution of anthropogenic $O_3$ precursors $CH_4$, $NO_x$, NMVOC and CO to the $O_3$ and $CH_4$ radiative forcing with earlier work (Shindell et al., 2005, 2009; Stevenson et al., 2013). Except for $NO_x$ which shows a large scatter across the studies, the FASST computed contributions to global $O_3$ and $CH_4$ forcing  - using the same year 1850 to 2000 emission changes as in Stevenson et al. (2013) - are in good agreement with the model ensemble range in the latter study. FASST $NO_x$ forcing contributions are a factor 3 lower than in the Stevenson et al. study and more in line with Shindell et al. (2005, 2009) values (based on the period 1750 – 2000), however the latter obtain a NMVOC contribution to $O_3$ forcing which is a factor of 5 to 6 lower than the other estimates. Differences across the studies are likely due to differences in oxidation chemistry and lifetimes across models.

### 3.3.2    Regional forcing efficiencies by emitted component

Earlier work in the frame of HTAP1 (Fry et al., 2012; Yu et al., 2013) and HTAP2 (Stjern et al., 2016) evaluated regional forcing efficiencies for larger regions than the ones defined for FASST. For a comparison we aggregate the FASST forcing efficiencies (as listed in section  S6.3 of the SI) by making an emission-weighted averages over Europe (EUR), North-America (NAM), South-Asia (SAS), East-Asia (EAS), Mediterranean and Middle East (MEA) and Russia, Belarus and Ukraine (RBU). Tables 11 (PM precursors) and 12 ($NO_x$, NMVOC and CO) show the earlier studies along with the FASST results. The FASST forcing efficiencies for PM precursors confirm our earlier observation that FASST is particularly biased low for BC, in particular compared to Stjern et al. (2016), but further compares relatively well with the earlier work, in particular with Yu et al. (2013) which was based on a year 2001 baseline, similar to conditions of our base scenario. A similar observation is made for the regional $O_3$ precursors for which FASST forcing efficiencies correspond within 1 standard deviation to the study by Fry et al. (2012) except for South- and East-Asia where FASST falls within 2 standard deviations.

### 3.3.3    Direct radiative forcing of short-lived climate pollutants by sector

The segregation of the RCP reference emission inventory by sector enables the evaluation of the contribution of individual sectors to the global instantaneous forcing. This is achieved by 'switching off' the respective sectorial emissions in the base emission scenario one by one, and comparing the resulting ΔForcing with the reference case. In Fig. 15 we compare the total

and sector-attributed direct radiative forcing with Unger et al. (2010) who made a similar evaluation for the year 2000 based on the EDGAR Fast Track 2000 emission inventory (Olivier et al., 2005). Figure 15b shows the break-down by forcing component, including the direct contributions by aerosols, by short-lived precursors to $O_3$ (SLS S-$O_3$), their indirect effect on $CH_4$ (SLS I-$CH_4$) and associated long-term $O_3$ (SLS M-$O_3$), as well as $CH_4$ forcing from direct $CH_4$ emissions and its associated feedback on background ozone (CH3 $O_3$). Fig. 15a separates the contributions by emission sector. Since different inventories are used, we do not expect a perfect match between the two analyses, however the emerging picture, in terms of over-all contribution by emitted component, as well as the contribution by sector is very similar, underlining the applicability of the TM5-FASST tool for this type of analysis in a consistent framework with other types of impacts. In general, BC forcing as well as the short-term $O_3$ forcing by $NO_x$ and NMVOC (SLS-$O_3$) are consistently lower for FASST, while the indirect feedbacks on $CH_4$ and long-term $O_3$ are corresponding well. This is also the case for the direct forcing by inorganic aerosols and POM. The higher direct $CH_4$ forcing and its feedback on $O_3$ by Unger et al. (2010) can be attributed to higher emissions in particular in the agricultural and waste – landfills sectors.

### 3.3.4    GWP and GTP

We use the methodology described in section 2.7.4 to evaluate global GTP and GWP for different time horizons H (20y and 100y) and compare with the range of values given in IPCC AR5 (Myhre et al., 2013). We recall that the forcings used to compute the FASST metrics, based on the meteorological year 2001 ad RCP year 2000 emissions, are region-specific and take into account differences in atmospheric life time and surface albedo. As shown in Table 13 we find an over-all good agreement with AR5 values. TM5-FASST BC metrics are at the low end of the IPCC range, in line with the previously made observation regarding the low FASST BC forcing. For the $NO_x$ metrics we have separately reported the strongly different ranges from Fuglestvedt et al. (2010) and Shindell et al. (2009). Our values for $NO_x$ appear to be more in line with the former study, except for GWP20 where FASST gives a negative value (-31) whereas AR5 reports a range (12, 26) from Fuglestvedt et al. (2010) and (-440, -220) from Shindell et al. (2009).

### 3.3.5    Health impacts

**Present-day health impacts**

Table 14 gives an overview of recent global $PM_{2.5}$ health impact studies, together with FASST estimates for the year 2000 (RCP) and year 2010 (HTAP2 scenario). The studies differ in emission inventories and year evaluated, in applied methodologies to estimate $PM_{2.5}$ exposure, in model resolution, as well as in the choice of the exposure response functions, the value of the minimum exposure threshold, and mortality statistics. Studies excluding natural dust from the exposure are mostly applying the log-lin exposure response function and RR from Krewski et al. (2009), and estimate between 1.6 and 2.7 million annual premature mortalities from $PM_{2.5}$ in scenario years 2000 to 2004. FASST returns 2.1 and 2.5 million deaths

using the GBD and log-lin exposure functions respectively. Studies including mineral dust are mostly applying the GBD integrated exposure-response functions and a non-zero threshold to avoid unrealistically high relative risk rates at high $PM_{2.5}$ levels in regions frequently exposed to dust. Depending on the choice of the exposure-response function and scenario year, FASST obtains 2.6 to 4.1 million global deaths, comparable with the range 1.7 to 4.2 million from previous studies.

Global ozone mortalities reported in Table 15 have been commonly based on the Jerrett et al. (2009) methodology, implemented in FASST. FASST obtains 197 thousand and 340 thousand deaths for RCP 2000 and HTAP2 2010 scenarios respectively, while the earlier studies find 380 to 470 thousand deaths in 2000, and 140 to 250 thousand in 2010 – 2015. Differences can be attributed to model chemical and meteorological processes, emission inventories, and the use of different sources for respiratory base mortality statistics.

Both for $PM_{2.5}$ and $O_3$, the difference between the different studies falls within the combined RR uncertainty and model variability range.

**Health impacts in future scenarios: intercomparison with ACCMIP model ensemble**

The health impact analysis of the RCP scenarios performed with the Atmospheric Chemistry and Climate Model
Intercomparison Project (ACCMIP) model-ensemble (Silva et al., 2016), provides a useful test case for the ability of TM5-FASST to reproduce trends derived from emission scenarios. The ACCMIP ensemble consisted of 14 state-of-the-art global chemistry climate models with spatial resolution from 1.9°x1.2° to 5°x5°. The ACCMIP models simulated future air quality for specific periods through 2100, for four global greenhouse gas and air pollutant emission scenarios projected in the Representative Concentration Pathways (RCPs). The analysis by Silva et al. (2016) used the same methodology
implemented in FASST for estimating premature mortalities from $PM_{2.5}$ and $O_3$ (i.e. Burnett et al., 2014 as in the Global Burden of Disease study and Jerrett et al., 2009 respectively), with the small difference that it does not include Acute Lower Respiratory Infections (ALRI) as a cause of death (in FASST applicable to age group below 5 years only) and the evaluated age group is >25 years old while in TM5-FASST this was done for population older than 30 years. Further, the ACCMIP health impact analysis uses scenario-specific projections for population and cause-specific base mortalities while FASST
uses the same population projections and mortality rates, as described in the methods section, across all scenarios.

Following the approach of Silva et al. (2016), we compare the global population-weighted annual mean $PM_{2.5}$ concentration change and ozone exposure metric 6mDMA1 relative to year 2000 concentrations for RCP scenarios 2.6, 4.5 and 8.5 for the years 2030 and 2050, with year 2000 exposure evaluated over the population of the respective scenario years (Tables S2 and S3 in Silva et al., 2016). Figure 16 shows the results from the ACCMIP model ensemble as well as individual model results
along with TM5-FASST outcome. We make the evaluation with and without the urban increment parameterization included (using the generic increment factors from Table S4.2). We find that TM5-FASST qualitatively reproduces $PM_{2.5}$ trends between 2030 and 2050 for the selected RCP scenarios although in only 2 of the 6 considered scenarios the TM5-FASST

concentration relative to year 2000 falls within the ACCMIP ensemble range. Even without urban increment correction, TM5-FASST consistently gives higher $PM_{2.5}$ exposure levels than ACCMIP (higher by 0.9, 1.5 and 1.0 µg m$^{-3}$ in 2030 and 0.7, 1.3 and 0.9 µg m$^{-3}$ in 2050 for RCP 2.6, 4.5 and 8.5 respectively). Apart from our previous finding that FASST tends to overestimate $PM_{2.5}$ levels compared to a full chemistry-transport model, an additional plausible explanation is the underlying

higher spatial resolution in FASST (1°x1°) than any of the ACCMIP models. Including the urban increment increases the global mean change in exposure relative to year 2000 with an additional 0.1 to 0.6 µg m$^{-3}$.

The ozone exposure metric 6mDMA1 falls within the range of the ACCMIP model ensemble for 2030 - 2050, but the slope between 2030 and 2050 is lower than for the ACCMIP ensemble mean, i.e. FASST shows a lower response sensitivity for $O_3$ to changing emissions between 2030 and 2050 than the ACCMIP models (-1ppb from 2030 to 2050 in FASST, versus -3ppb

for the ACCMIP mean). Given our previous observation that FASST reproduces TM5 relatively well, this indicates that inter-model variability is a stronger factor in the model uncertainty than the reduced-form approach.

The trends from 2000 to 2050 in global mortality burden from $PM_{2.5}$ and $O_3$ are shown in Figure 17. Assuming that the relative error for the year 2000 – the only uncertainty range given by Silva et al. (2016) – can be applied on the other cases, we find that TM5-FASST reproduces the ACCMIP health impacts from $PM_{2.5}$ within the ACCMIP range. Including the

urban increment correction increases the mortality by 26% in 2000, 24%, 22% and 17% in 2030, and 32%, 31% and 25% in 2050 for RCP2.6, 4.5 and 8.5 respectively.

While calculated $O_3$ mortalities for years 2000 and 2030 are within the ACCMIP range, TM5-FASST does not confirm the strongly increasing $O_3$ mortalities in the ACCMIP ensemble by 2050. However this difference can be attributed to the use of different baseline mortality statistics, in particular for the year 2050 where FASST, by lack of WHO projections for 2050,

assumes year 2030 WHO projected mortality rates whereas Silva et al. (2016) use International Futures (IFs) projections up till 2100. Indeed, the IFs projections (Fig. S7 in the SI of Silva et al., 2016) foresee relative constant global mortality rates (deaths per 1000 people) between 2030 and 2050 for all air pollution-related death causes, except for respiratory disease (on which $O_3$ mortality estimates are based) which increases with a factor 2.5 globally from 2030 to 2050. An acceptable agreement with the ACCMIP model ensemble outcome is achieved when this effect is included as a simple adjustment factor

on the FASST RCP year 2050 $O_3$ mortalities, as shown by the dot-symbols (year 2050) in Fig. 17. Respiratory mortality is not considered as a cause of death for $PM_{2.5}$, which explains why a similar disagreement is not observed in the $PM_{2.5}$ mortality trend in Fig. 17b.

A regional break-down of mortality burden from $PM_{2.5}$ in 2030 and 2050, relative to exposure to year 2000 concentrations, for major world regions and for the globe is shown in Figures S9.1 and S9.2 of the SI. Compared to Fig. 17 which shows the

global mortality trends as a combined effect of changing population, mortality rates and pollution level, here the effect of changing population and baseline mortality is eliminated by exposing the evaluated year's population to pollutant levels of the relevant year and to RCP year 2000 levels respectively, and calculating the change between the two resulting mortality

numbers. FASST reproduces the over-all observed trends across the regions: we see substantial reductions in North America and Europe in 2030, whereas in East Asia significant improvements in air quality impacts are realized after 2030. For the India region, all scenarios project a worsening of the situation. The global trend is dominated by the changes in East Asia. The observed differences between FASST and ACCMIP ensemble are not insignificant and partly due to different mortality and population statistics in particular for the year 2050, still they are consistent with the findings in the previous section: FASST tends to overestimate absolute $PM_{2.5}$ concentrations for emission scenarios different from RCP2000, and consequently tends to under-predict the benefit of emission reductions, while over-predicting the impact of increasing emissions.

### 3.3.6    Present day O3 – crop losses

Avnery et al. (2011) evaluate year 2000 global and regional $O_3$-induced crop losses for wheat, maize and soy bean, based on the same crop ozone exposure metrics as used in FASST, obtained with a global chemical transport model at 2.8°x2.8° resolution. Figure 18 compares their results (in terms of relative yield loss) with FASST (TM5) results based on RCP year 2000 for the globe and 3 selected key regions (Europe, North-America and East Asia). Despite the less-robust quantification of crop impacts from $O_3$ in a linearized reduced-form model set-up, we find that FASST reproduces the major features and trends across regions and crop varieties.   Differences may be attributed to a variety of factors, including model resolution, model O3 chemistry processes, emissions, definition of crop growing season and crop spatial distribution.

### 4    Discussion

Although the methodology of a reduced-form air quality model, based on linearized emission – concentration sensitivities is not new and has been successfully applied in earlier studies (Alcamo et al., 1990), the concept of  directly linking pollutant emission scenarios to a large set of impacts across various policy fields, in a global framework, have made TM5-FASST a highly requested tool in a broad field of applications. HTAP1 showed that TM5 source-receptor results (for the large HTAP1 regions) were in most cases similar to the median model results of more than 10 global models, lending additional trust to the model performance (e.g. Anenberg et al., 2014; Dentener et al., 2010; Fiore et al., 2009). The results in the previous sections have outlined its strengths and weaknesses. The major strength of the tool is its mathematical simplicity allowing for a quick processing of large sets of scenarios or scenario ensembles. An extreme example is the full family of SSP scenarios delivered by all participating Integrated Assessment Models, for decadal time slices up to 2050, constituting a batch of 594 scenarios of  which a selection of 124 scenarios was analysed with TM5-FASST in the study by Rao et al. (2017). Further, the tool is unique in having a broad portfolio of implemented impact modules which are evaluated consistently over the global domain from the same underlying pollutant field which creates a basis for a balanced evaluation of trade-offs and benefits attached to policy options.

On the other hand, the reduced-form approach inevitably encompasses a number of caveats and uncertainties that have to be considered with care and which are discussed in the following sections.

## 4.1     Issues related to the reduced-form approach

The reliability of the model output in terms of impacts depends critically on the validity of the linearity assumption for the relevant exposure metrics (in particular secondary components), which becomes an issue when evaluating emission scenarios that deviate strongly from the base and -20% perturbation on which the current FASST SRs are based. The evaluation exercise indicated that non-linearity effects in $PM_{2.5}$ and $O_3$ metrics in general lead to a higher bias for stringent emission reductions (towards -80% and beyond) than for strong emission increases compared to the RCP2000 base case, but over-all remain within acceptable limits when considering impacts. Indeed, because of the thresholds included in exposure-response functions, the higher uncertainty on low (below-threshold) pollutant levels from strong emission reductions has a low weight in the quantification of most impacts. In future developments the available extended-range (-80%, +100%) emission perturbation simulations could form the basis of a more sophisticated parameterization including a bias correction based on second order terms following the approach by Wild et al. (2012) both for $O_3$ and secondary $PM_{2.5}$. The break-down of the linearity at low emission strengths is relevant for $O_3$ and $O_3$ exposure metrics as the implementation of control measures in Europe and the US has already substantially lowered $NO_x$ levels over the past decade,  gradually modifying the prevailing $O_3$ formation regime from $NO_x$-saturated (titration regime) to $NO_x$-limited (Jin et al., 2017).

Ozone impact on agricultural crop production is deemed to be the least robustly quantified impact category included in FASST, in particular when evaluated from the threshold-based AOT40 metric, and has to be interpreted as indicative order-of-magnitude estimate. In an integrated assessment perspective of evaluating trade-offs and benefits of air pollutants scenarios, the dominant impact category however appears to be human health (Kitous et al., 2017; OECD, 2016; UNEP, 2011) where TM5-FASST provides reliable estimates.

Another issue for caution relates to the FASST analysis of emission scenarios with spatial distribution that differs from the FASST reference scenario (RCP year 2000). The definition of the source regions when establishing the SR matrices implicitly freezes the spatial distribution of pollutant emissions within each region, and therefore the reduced-form model cannot deal with intra-regional spatial shifts in emissions. In practice this is not expected to introduce large errors as anthropogenic emissions are closely linked to populated areas and road networks of which the extent may change, but much less so the spatial distribution.  It can be a problem when going far back in time, when large patterns of migration and land development occurred, while in RCP scenarios relatively simple expansions of emissions into the future did not assume huge shifts in regional emission patterns.

The implicitly fixed emission spatial distribution may also become relevant when making a sector apportionment of pollutant concentrations and impacts. Source-Receptor relations are indeed particularly useful to evaluate the apportionment of emission sources (in terms of economic sector as well as source regions) to pollutant levels in a given receptor. However, as

the TM5-FASST_v0 source-receptor matrices were not segregated according to economic sectors, an emission reduction of 20% for a given source region is implicitly considered as a 20% reduction in all sectors simultaneously. Although the atmospheric chemistry and transport of emissions is in principle independent of the specific source, a difference in the sector-specific SR matrices may occur due to differences in temporal and spatial (horizontal/vertical) distribution of the sources. Therefore apportionment studies on sectors which have a significantly different emission spatial distribution than other sectors in the same region should be interpreted with care. In particular impacts of off-shore flaring cannot be assessed with TM5-FASST because those emissions were not included in the RCP base emissions. This limitation however does not apply to international shipping and aviation for which specific SR matrices have been established.

Comparing to earlier studies and reference data, the performance of TM5-FASST with respect to climate metrics is satisfactory, with the exception of BC forcing which is at the low side of current best estimates. In fact, earlier TM5-FASST assessments where climate metrics were provided (UNEP, 2011; UNEP and CCAC, 2016) applied a uniform adjustment factor of 3.6 on BC forcing, in line with the observation by that many models underestimate atmospheric absorption attributable to BC with a factor of almost 3. In TM5-FASST, an adjustment factor of 3.6 leads to a global forcing by anthropogenic BC of 600 mW m$^{-2}$. This tuning factor implicitly accounts for not-considered BC forcing contributions and for a longer BC atmospheric lifetime than implemented in the TM5 model and the resulting FASST SR coefficients.

The current version of TM5-FASST is missing some source-receptor relations which may introduce a bias in estimated PM$_{2.5}$ and O$_3$ responses upon emission changes. The omission of secondary organic PM in TM5 is estimated to introduce a low bias in the base concentration of the order of 0.1 µg m$^{-3}$ as global mean however with regional levels in Central Europe and China up to 1 µg m$^{-3}$ in areas where levels of primary organic matter are reaching 20 µg m$^{-3}$ (Farina et al., 2010) indicating a relatively low contribution of SOA to total PM$_{2.5}$. O$_3$ formation from CO is included in the TM5 base simulations, but no SR matrices for the FASST source region definition are available. Based on the HTAP1 CO perturbation simulations with TM5, we estimate that a doubling of anthropogenic CO emissions contributes with 1 – 1.9 ppb in annual mean O$_3$ over Europe, 1.3 -1.9 ppb over North-America, 0.7-1.0 ppb over South Asia and 0.3 – 1.5 ppb over East-Asia. Development of CO-O$_3$ SRs is an important issue for the further development of the tool.

## 4.2 Inter-annual meteorological variability

A justified critique on the methodology applied to construct the FASST SRs relates to the use of a single and fixed meteorological year 2001, implying possible unspecified biases in pollutant concentrations and source-receptor matrices compared to using a 'typical meteorological/climatological year'. We followed the choice of the meteorological year 2001 made for the HTAP1 exercise. As the North-Atlantic Oscillation (NAO) is an important mode of the inter-annual variability in pollutant concentrations and long range transport (Christoudias et al., 2012; Li et al., 2002; Pausata et al., 2013; Pope et al., 2018), the HTAP1 expectation was that this year was not an exceptional year for long-rang pollutant transport - e.g. for the North-Atlantic region, as indicated by a North Atlantic Oscillation (NAO) index close to zero for that year

(https://www.ncdc.noaa.gov/teleconnections/nao/). The HTAP1 report (Dentener et al., 2010) also suggested that "Inter-annual differences in SR relationships for surface $O_3$ due to year-to-year meteorological variations are small when evaluated over continental-scale regions. However, these differences may be greater when considering smaller receptor regions or when variations in natural emissions are accounted for". The role of spatial and temporal meteorological variability can thus be reduced by aggregating resulting pollutant levels and impacts as regional and annual averages or aggregates, the approach taken in TM5-FASST.

The impact of the choice of this specific year on the TM5-FASST model uncertainty or possible biases in base concentrations and SR coefficients is not easily quantified. For what concerns the pollutant base concentrations, some insights in the possible relevance of meteorological variability can found in the literature. For example, Anderson et al., (2007) showed that in Europe, the meteorological component in regional inter-annual variability of pollutant concentrations ranges between 3% and 11% for airborne pollutants ($O_3$, $PM_{2.5}$), and up to 20% for wet deposition. On a global scale, Liu et al. (2007) demonstrated that the inter-annual variability in PM concentrations, related to inter-annual meteorological variability can even be up to a factor of 3 in the tropics (e.g. over Indonesia) and in the storm track regions. A sample analysis (documented in section S2.2 of the SI) of the RCP year 2000 emission scenario with TM5 at 6°x4° resolution of 5 consecutive meteorological years 2001 to 2005 indicates a year-to-year variability on regional $PM_{2.5}$ within 10% (relative standard deviation) and within 3% for annual mean $O_3$. We find a similar variability on the magnitudes of 20% emission perturbation responses within the source region for 6 selected regions (India, China, Europe, Germany, USA and Japan). The relative share of source regions to the pollutant levels within a given receptor region shows a lower inter-annual variability (typically between 2 and 6% for $PM_{2.5}$) than the absolute contributions.

**4.3     Impact of the native TM5 grid resolution on pollutant concentration and SRs**

FASST base concentrations and SRs have been derived at a 1°x1° resolution which is a relatively fine grid for a global model, but still not optimal for population exposure estimates and health impact assessments. Previous studies have documented the impact of grid resolution on pollutant concentrations. The effect of higher grid resolution in global models is in general to decrease ozone exposure in polluted regions and to reduce $O_3$ long-range transport, while $PM_{2.5}$ exposure – mainly to primary species - increases (Fenech et al., 2018; Li et al., 2016; Punger and West, 2013). Without attempting a detailed analysis, a comparison of TM5 available output for $PM_{2.5}$ and $O_3$ at 6°x4°, 3°x2° and 1°x1° resolution confirms these findings, as illustrated in Fig. S2.6 of the SI. Although FASST is expected to better represent population exposure to pollutants than coarser resolution models, a resolution of 1°x1° may not adequately capture urban scale pollutant levels and gradients when the urban area occupies only a fraction of the grid cell. The developed sub-grid parameterization for $PM_{2.5}$, providing an order-of-magnitude correction which is consistent with a high-resolution satellite product, is subject to improvement and to extension to other primary pollutants ($NO_2$, e.g. Kiesewetter et al., 2014, 2015) and $O_3$. To our

knowledge a workable parametrization to quantify the impact of sub-grid $O_3$ processes on population exposure – in particular titration due to local high $NO_x$ concentrations in urban areas - has not been addressed in global air quality models.

The impact of grid resolution on the within-region source-receptor coefficients can be significant, in particular for polluted regions where the coarse resolution includes ocean surface, like Japan. Table S2.3 in the SI shows as an example within-region and long-range SR coefficients for receptor regions Germany, USA and Japan. A higher grid resolution increases the within-region response and decreases the contribution of long–range transport (where the contribution of China to nearby Japan behaves as a within-region perturbation). In the case of Japan, the within-region $PM_{2.5}$ response magnitude increases with a factor of 3, and the sign of the within-region $O_3$ response is reversed when passing from 6°x4° to higher resolution. Also over the USA, the population-weighted within-region response sensitivity upon $NO_x$ perturbation increases with a factor of 5. Further, we find that in titration regimes, the magnitude of the $O_3$ response to $NO_x$ emissions increases with resolution (i.e. ozone increases more when $NO_x$ is reduced using a fine resolution) whereas the in-region ozone response is reduced in non-titration regimes (India and China, Fig. 2.7d). These indicative results are in line with more detailed studies (e.g. Wild and Prather, 2006).

## 5    Conclusions and way forward

The FASST_v0 version of TM5 is a trade-off between accuracy and applicability. TM5-FASST_v0 enables immediate "what-if" and sensitivity calculations, and, by means of the available source-receptor coefficients, the extraction of this information down to the level of individual regions, economic sectors and chemical compounds. In this paper we have extensively documented the embedded methodology and validated the tool against the full chemistry transport model as well as against selected case studies from the literature. In conclusion, provided that the TM5-FASST_v0 is considered as a screening tool, the simplifications introduced in order to generate immediate results from emission scenarios are not compromising the validity of the output and as such TM5-FASST_v0 has been proven to be a useful tool in science-policy analysis.

The native set of TM5-FASST region-to-grid source-receptor grid maps is sufficiently detailed, both in terms of spatial and temporal resolution as well as number of pollutant species and metrics, to include additional impact categories not included so far. Some examples are BC deposition to snow-covered surfaces, combined nitrogen fertilization and $O_3$ feedbacks on Carbon-sequestration by vegetation from $NO_x$ emission, both relevant as additional climate forcing, population exposure to $NO_2$ and $SO_2$ as additional health effects.

The regional 58x56 region-to-region source-receptor matrices aggregated from the high-resolution (region-to-gridmap) SRs are easily implemented in a spreadsheet-type environment. A user-friendly web-based interactive stable version based on the latter is available at http://tm5-fasst.jrc.ec.europa.eu/. This version offers the possibility to explore built-in as well as user-defined scenarios, using static default urban increment correction factors and crop production data. A more sophisticated in-

house research version with gridded output and flexibility in the choice of gridded ancillary data (population grid maps, scenario-specific urban increment factors, crop distribution) is under continuous development and has been applied for the assessments listed in table S1.

Some foreseen further developments of the TM5-FASST tool, making use of readily available SRs include:

- Using the available extended-range perturbation simulations to develop a correction algorithm on the current simple linear extrapolation procedure, in particular for the regions where the $O_3$ or secondary $PM_{2.5}$ regimes are non-linear, e.g. following the approach by Wild et al (2010) and Turnock et al. (2018)

- Update the health impact modules with recent findings in literature, specifically on the long-term $O_3$ impact (Turner et al., 2016), adjusted IER function parameters and age-specific exposure – response functions for $PM_{2.5}$ mortalities (Cohen et al., 2017), as well as including different health metrics (DALYS, life years lost) and improved projections for base mortalities and other health statistics.

- Including a transient $O_3$ response function to $CH_4$ emission changes

- Including cryosphere forcing via BC deposition

- Stomatal approach for crop ozone impacts and extension of vegetation types considered

- Higher temporal resolution exploiting the available native monthly source-receptor maps.

Even with these further developments, an important limitation of TM5-FASST_v0 remains that it is based on a single meteorological year (2001), on source-receptor relations computed by a single underlying Chemistry-Transport model, based on the reference year 2000, and using fixed fields for natural $PM_{2.5}$. The HTAP phase 2 modelling exercise addresses these issues: it has been designed in line with the FASST philosophy (albeit with a larger aggregation of source region definitions), with an *ensemble* of chemistry-transport or climate-chemistry models providing source-receptor simulations, based on an updated and harmonized common anthropogenic pollutant emission inventory for the years 2008 - 2010 (Janssens-Maenhout et al., 2015; Koffi et al., 2016). The FASST architecture allows for an implementation of new or additional SR matrices, for instance new HTAP2 model ensemble mean matrices, each one accompanied by an ensemble standard deviation matrix to include the model variability in the results. Efforts are now underway to create a new web-based and user-friendly HTAP-FASST version, operating under the same principles as TM5-FASST, but based on an up-to-date reference simulation and underlying meteorology, thus creating a link between the knowledge generated by the HTAP scientific community and interested policy-oriented users.

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

**Table 1: Relevant precursor-pollutant relationships included in TM5-FASST. ●: direct emission or immediate product; ○: effect via thermodynamic equilibration; ◊ effect via first order oxidation products (OH) affecting the lifetime of other precursors.**

| Pollutant / Precursor | $SO_{2\,(g)}$ | $NO_{x\,(g)}$ | $NH_{3\,(g)}$ | $O_3$ | $CH_4$ | $SO_4\,(PM_{2.5})$ | $NO_3\,(PM_{2.5})$ | $NH_4\,(PM_{2.5})$ | $EC\,(PM_{2.5})$ | $POM\,(PM_{2.5})$ | $SOx\,(dep)$ | $NOy\,(dep)$ | Rad. forcing |
|---|---|---|---|---|---|---|---|---|---|---|---|---|---|
| $SO_2$ (g) | ● | ◊ | ○ | ◊ | ◊ | ● | ○ | ○ | | | ● | | ● |
| $NO_x$ (g) | ◊ | ● | ○ | ● | ◊ | ○ | ● | ○ | | | | ◊ | ● |
| $NH_3$ (g) | ◊ | ◊ | ● | ◊ | ◊ | ○ | ○ | ● | | | | ◊ | ● |
| BC (pm) | | | | | | | | | ● | | | | ● |
| POM (pm) | | | | | | | | | | ● | | | ● |
| NMVOC (g) | ◊ | ◊ | ◊ | ● | ◊ | ◊ | ◊ | ◊ | | | | ◊ | ● |
| CO (g)* | | | | ● | ◊ | | | | | | | | ● |
| $CH_4$ (g)* | ◊ | ◊ | ◊ | ◊ | ● | ◊ | ◊ | ◊ | | | | ◊ | ● |

∗    From HTAP phase 1 (Dentener et al., 2010)

**Table 2: Overview of TM5-CTM perturbation simulations (20% emission reduction) for the calculation of the source-receptor (SR) matrices*comparing to the same zoom regions as in P0.**

| Simulation | Emission perturbations | Applied on source regions | Scope |
|---|---|---|---|
| P0 | No perturbations | Master zoom regions with 1°x1° resolution: AFR, AUS, EAS, EUR, MAM, MEA, NAM,RSA, RUS, SAM, SAS, SEA and  PAC (3°x2°) | Base simulation |
| P1 | $SO_2$, $NO_x$, BC, POM | All 56 continental regions* + international shipping + aviation | SR matrices for BC and POM and first order approximation for $SO_2$ and $NO_x$, assuming negligible chemical interaction |
| P2 | $SO_2$ | All 56 source regions* + shipping | Independent SR for $SO_2$, to be compared to P1 to quantify potential interference between $SO_2$ and $NO_x$ in the formation of sulfate and ozone |
| P3 | $NO_x$ | Representative source regions* (China, Europe, Japan, India, Germany, South-Africa, USA) | Independent SR for $NO_x$, to verify the additivity of P1 = P2 + P3 and justify the use of (P1 – P2) as a proxy for $NO_x$ perturbation for all other regions |
| P4 | $NH_3$, NMVOC | All 56 continental source* regions + international shipping | SR matrices for $NH_3$ and NMVOC emissions, assuming little chemical interaction between the selected precursors in the formation of $NH_4$ and $O_3$ |
| P5 | NMVOC, $NO_x$ | Representative source regions* (Europe, China, India, USA) | Quantify chemical feedbacks in $O_3$ formation between $NO_x$ and NMVOC (P5 = P3 + P4) additivity |

*See list of regions and their definition in Table S2.2 of the SI.

**Table 3.** Overview of air quality indices used to evaluate crop yield losses. The a, b and c coefficients refer to the exposure-response equations given in the equations 5 and 6. Source: Van Dingenen et al. (2009), Mills et al. (2007), Wang and Mauzerall (2004)

| Metric: | Wheat | | | Rice | | | Soy | | | Maize | | |
|---|---|---|---|---|---|---|---|---|---|---|---|---|
| | a | b | c | a | b | c | a | b | c | a | b | c |
| AOT40 (ppm.h) | 0.0163 | - | - | 0.00415 | - | - | 0.0113 | - | - | 0.00356 | - | - |
| Mi (ppbV) | 137 | 2.34 | 25 | 202 | 2.47 | 25 | 107 | 1.58 | 20 | 124 | 2.83 | 20 |

**Table 4: Statistical metrics describing the correspondence between the linearized FASST and TM5 computed change in secondary PM$_{2.5}$ upon -80% and 100% emission perturbation in its precursors (SO$_2$, NO$_x$, NH$_3$ and combined SO$_2$ + NO$_x$), relative to the RCP2000 base scenario. Statistics are calculated over all 1°x1° grid cells in each region. Statistics for total concentrations are given in annex S7 of the SI.**

| Region | FASST MEAN (µg m$^{-3}$) -80% | 100% | TM5 MEAN (µg m$^{-3}$) -80% | 100% | NMB[a] (%) -80% | 100% | MB[b] (µg m$^{-3}$) -80% | 100% | R$^{2}$[c] -80% | 100% |
|---|---|---|---|---|---|---|---|---|---|---|
| **Precursor: SO$_2$** | | | | | | | | | | |
| EUR | -1.0 | 1.2 | -1.1 | 1.1 | -11.8 | 12.8 | 0.13 | 0.14 | 0.99 | 1.00 |
| USA | -0.8 | 1.1 | -0.9 | 1.0 | -8.2 | 10.8 | 0.08 | 0.10 | 1.00 | 1.00 |
| JPN | -0.3 | 0.4 | -0.3 | 0.4 | -5.0 | 6.8 | 0.02 | 0.02 | 1.00 | 1.00 |
| CHN | -1.5 | 1.8 | -1.7 | 1.6 | -13.3 | 17.7 | 0.22 | 0.28 | 1.00 | 1.00 |
| IND | -2.1 | 2.7 | -2.2 | 2.5 | -4.6 | 8.3 | 0.10 | 0.20 | 1.00 | 1.00 |
| **Precursor: NO$_x$** | | | | | | | | | | |
| EUR | -0.9 | 1.2 | -1.1 | 0.8 | -13.7 | 44.4 | 0.15 | 0.36 | 0.96 | 0.95 |
| USA | -0.5 | 0.6 | -0.6 | 0.4 | -25.1 | 60.9 | 0.15 | 0.21 | 0.87 | 0.87 |
| JPN | -0.3 | 0.4 | -0.4 | 0.2 | -27.3 | 93.2 | 0.11 | 0.17 | 0.92 | 0.91 |
| CHN | -0.8 | 1.0 | -0.9 | 0.7 | -11.9 | 35.5 | 0.11 | 0.26 | 0.97 | 0.90 |
| IND | -0.6 | 0.7 | -0.6 | 0.8 | 6.8 | -9.3 | -0.04 | -0.08 | 0.95 | 0.94 |
| **Precursor: NH$_3$** | | | | | | | | | | |
| EUR | -1.1 | 1.4 | -1.6 | 1.2 | -29.0 | 12.8 | 0.45 | 0.16 | 0.97 | 0.92 |
| USA | -0.6 | 0.8 | -0.8 | 0.6 | -20.2 | 28.6 | 0.16 | 0.17 | 0.96 | 0.94 |
| JPN | -0.4 | 0.4 | -0.4 | 0.4 | -16.9 | 28.2 | 0.07 | 0.10 | 0.98 | 0.99 |
| CHN | -0.8 | 1.0 | -1.0 | 0.7 | -25.5 | 43.8 | 0.26 | 0.30 | 0.98 | 0.98 |
| IND | -0.2 | 0.3 | -0.4 | 0.2 | -47.6 | 48.4 | 0.18 | 0.08 | 0.88 | 0.94 |
| **Precursor: SO$_2$+NO$_x$** | | | | | | | | | | |
| EUR | -1.9 | 2.4 | -2.3 | 1.8 | -17.5 | 33.5 | 0.40 | 0.60 | 0.94 | 0.95 |
| USA | -1.3 | 1.6 | -1.6 | 1.2 | -16.1 | 31.2 | 0.25 | 0.39 | 0.96 | 0.97 |
| JPN | -0.6 | 0.7 | -0.7 | 0.5 | -16.5 | 44.9 | 0.11 | 0.22 | 0.96 | 0.96 |

[a] Normalized Mean Bias $= \left(\overline{FASST} - \overline{TM5}\right)/\overline{TM5}$

[b] Mean Bias $= \left(\overline{FASST} - \overline{TM5}\right)$

[c] Correlation coefficient

$\bar{Y}$ = average of all grid cells in region

**Table 5: Statistical metrics describing the correspondence between the linearized FASST and TM5 computed change in O$_3$ exposure metric 6mDMA1 upon -80% and 100% emission perturbation in its precursors (NMVOC, NO$_x$ and combined NO$_x$ + NMVOC), relative to the RCP2000 base scenario. Statistics are calculated over all 1°x1° grid cells in each region. Statistics for total concentrations are given in annex S7 of the SI.**

| Region | FASST MEAN (ppb) | | TM5 MEAN (ppb) | | NMB[a] (%) | | MB[b] (ppb) | | R$^{2}$[c] | |
|---|---|---|---|---|---|---|---|---|---|---|
| | -80% | 100% | -80% | 100% | -80% | 100% | -80% | 100% | -80% | 100% |
| **Precursor: NMVOC** | | | | | | | | | | |
| EUR | -1.5 | 1.8 | -1.7 | 1.3 | -11 | 36 | 0.2 | 0.5 | 0.55 | 0.41 |
| USA | -1.1 | 1.4 | -1.3 | 1.2 | -10 | 23 | 0.1 | 0.3 | 0.98 | 0.99 |
| JPN | -0.9 | 1.1 | -1.0 | 0.8 | -14 | 30 | 0.1 | 0.3 | 0.99 | 0.98 |
| CHN | -0.9 | 1.1 | -1.3 | 0.6 | -30 | 93 | 0.4 | 0.5 | 0.98 | 0.96 |
| IND | -0.9 | 1.1 | -1.2 | 0.7 | -25 | 59 | 0.3 | 0.4 | 0.99 | 0.99 |
| **Precursor: NO$_x$** | | | | | | | | | | |
| EUR | -2.7 | 3.3 | -4.5 | 1.2 | -41 | 169 | 1.9 | 2.1 | 0.87 | 0.77 |
| USA | -4.5 | 5.7 | -6.8 | 3.3 | -33 | 70 | 2.3 | 2.3 | 0.79 | 0.85 |
| JPN | -1.1 | 1.4 | -2.7 | -0.4 | -58 | -499 | 1.6 | 1.8 | 0.59 | 0.59 |
| CHN | -4.3 | 5.4 | -6.1 | 3.3 | -29 | 64 | 1.7 | 2.1 | 0.96 | 0.82 |
| IND | -7.3 | 9.1 | -9.6 | 6.4 | -25 | 41 | 2.4 | 2.7 | 0.98 | 0.96 |
| **Precursor: NO$_x$ + NMVOC** | | | | | | | | | | |
| EUR | -4.1 | 5.2 | -5.1 | 3.8 | -18 | 38 | 0.9 | 1.4 | 0.89 | 0.97 |
| USA | -5.7 | 7.1 | -7.1 | 5.2 | -20 | 36 | 1.4 | 1.9 | 0.97 | 0.95 |
| CHN | -5.2 | 6.5 | -6.0 | 5.2 | -13 | 26 | 0.8 | 1.3 | 0.99 | 0.99 |
| IND | -8.1 | 10.1 | -9.6 | 8.4 | -15 | 21 | 1.5 | 1.7 | 0.99 | 0.99 |

[a] Normalized Mean Bias $= \left(\overline{FASST} - \overline{TM5}\right)/\overline{TM5}$

[b] Mean Bias $= \left(\overline{FASST} - \overline{TM5}\right)$

[c] Correlation coefficient

$\overline{Y}$ = average of all grid cells in region

**Table 6: Statistical metrics describing the correspondence between the linearized FASST and TM5 computed change in $O_3$ crop exposure metric AOT40 upon -80% and 100% emission perturbation in its precursors (NMVOC, $NO_x$ and combined $NO_x$ + NMVOC), relative to the RCP2000 base scenario. Statistics are calculated over all 1°x1° grid cells in each region.**

| Region | FASST MEAN (ppm.h) -80% | 100% | TM5 MEAN ppm.h -80% | 100% | NMB[a] (%) -80% | 100% | MB[b] (ppm.h) -80% | 100% | $R^{2[c]}$ -80% | 100% |
|---|---|---|---|---|---|---|---|---|---|---|
| **Precursor: NMVOC** | | | | | | | | | | |
| EUR | -1.1 | 1.4 | -1.3 | 1.2 | -11 | 24 | 0.1 | 0.3 | 0.87 | 0.75 |
| USA | -1.0 | 1.3 | -1.1 | 1.0 | -10 | 26 | 0.1 | 0.3 | 0.98 | 0.99 |
| JPN | -0.7 | 0.8 | -0.8 | 0.6 | -13 | 38 | 0.1 | 0.2 | 0.98 | 0.98 |
| CHN | -0.7 | 0.8 | -0.9 | 0.4 | -29 | 95 | 0.3 | 0.4 | 0.98 | 0.96 |
| IND | -0.6 | 0.8 | -0.8 | 0.4 | -27 | 70 | 0.2 | 0.3 | 0.98 | 0.96 |
| **Precursor: $NO_x$** | | | | | | | | | | |
| EUR | -2.1 | 2.6 | -3.1 | 1.3 | -34 | 102 | 1.1 | 1.3 | 0.93 | 0.84 |
| USA | -4.6 | 5.7 | -6.3 | 3.7 | -27 | 57 | 1.7 | 2.1 | 0.82 | 0.86 |
| JPN | -0.7 | 0.9 | -1.7 | -0.2 | -56 | -498 | 0.9 | 1.1 | 0.83 | 0.63 |
| CHN | -3.0 | 3.7 | -3.5 | 2.5 | -14 | 50 | 0.5 | 1.3 | 0.92 | 0.87 |
| IND | -4.5 | 5.6 | -5.3 | 3.9 | -15 | 44 | 0.8 | 1.7 | 0.93 | 0.91 |
| **Precursor: $NO_x$ + NMVOC** | | | | | | | | | | |
| EUR | -3.2 | 4.0 | -4.2 | 1.8 | -23 | 126 | 1.0 | 2.2 | 0.94 | 0.91 |
| USA | -5.6 | 7.0 | -6.9 | 3.8 | -18 | 86 | 1.3 | 3.2 | 0.95 | 0.90 |
| CHN | -3.7 | 4.6 | -4.3 | 2.4 | -15 | 90 | 0.6 | 2.2 | 0.87 | 0.89 |
| IND | -5.1 | 6.3 | -5.8 | 3.6 | -12 | 76 | 0.7 | 2.7 | 0.89 | 0.90 |

[a] Normalized Mean Bias = $(\overline{FASST - TM5})/\overline{TM5}$

[b] Mean Bias = $(\overline{FASST - TM5})$

[c] Correlation coefficient

$\overline{Y}$ = average of all grid cells in region

**Table 7: Statistical metrics describing the correspondence between the linearized FASST and TM5 computed change in $O_3$ crop exposure metric M12 upon -80% and 100% emission perturbation in its precursors (NMVOC, $NO_x$ and combined $NO_x$ + NMVOC), relative to the RCP2000 base scenario. Statistics are calculated over all 1°x1° grid cells in each region**

| Region | FASST MEAN (ppb) | | TM5 MEAN (ppb) | | NMB[a] (%) | | MB[b] (ppb) | | $R^{2}$[c] | |
|---|---|---|---|---|---|---|---|---|---|---|
| | -80% | 100% | -80% | 100% | -80% | 100% | -80% | 100% | -80% | 100% |
| Precursor: NMVOC | | | | | | | | | | |
| EUR | -0.9 | 1.1 | -1.6 | 1.3 | -43 | -16 | 0.7 | -0.2 | 0.50 | 0.37 |
| USA | -1.0 | 1.3 | -1.2 | 1.0 | -11 | 27 | 0.1 | 0.3 | 0.98 | 0.99 |
| JPN | -0.7 | 0.9 | -0.8 | 0.6 | -16 | 38 | 0.1 | 0.2 | 0.98 | 0.97 |
| CHN | -0.8 | 0.9 | -1.1 | 0.5 | -33 | 102 | 0.4 | 0.5 | 0.98 | 0.95 |
| IND | -0.6 | 0.8 | -0.9 | 0.5 | -28 | 76 | 0.7 | 0.3 | 0.98 | 0.95 |
| Precursor: $NO_x$ | | | | | | | | | | |
| EUR | -1.6 | 2.0 | -3.2 | 0.4 | -49 | 392 | 1.6 | 1.6 | 0.87 | 0.78 |
| USA | -4.3 | 5.4 | -6.4 | 3.2 | -33 | 66 | 2.1 | 2.2 | 0.82 | 0.84 |
| JPN | 0.5 | -0.6 | -0.6 | -1.9 | -188 | -67 | 1.1 | 1.3 | 0.92 | 0.80 |
| CHN | -3.4 | 4.3 | -4.9 | 2.5 | -30 | 68 | 1.5 | 1.7 | 0.95 | 0.81 |
| IND | -4.8 | 6..0 | -6.8 | 3.9 | -29 | 54 | 2.0 | 2.1 | 0.94 | 0.98 |
| Precursor: $NO_x$ + NMVOC | | | | | | | | | | |
| EUR | -2.5 | 3.2 | -3.8 | 2.7 | -33 | 16 | 1.2 | 0.4 | 0.88 | 0.88 |
| USA | -5.3 | 6.7 | -6.6 | 5.0 | -19 | 34 | 1.3 | 1.7 | 0.96 | 0.94 |
| CHN | -4.2 | 5.2 | -4.8 | 4.2 | -13 | 25 | 0.6 | 1.1 | 0.98 | 0.96 |
| IND | -5.5 | 6.9 | -6.6 | 5.6 | -18 | 23 | 1.2 | 1.3 | 0.96 | 0.94 |

[a] Normalized Mean Bias $= \left(\overline{FASST - TM5}\right)/\overline{TM5}$

[b] Mean Bias $= \left(\overline{FASST - TM5}\right)$

[c] Correlation coefficient

$\bar{Y}$ = average of all grid cells in region

**Table 8: Regional grid cell mean anthropogenic PM$_{2.5}$ concentration (including primary and secondary components) and performance statistics for FASST vs. TM5, for the high (FLE2030) and low (MIT2030) emission scenarios and for the delta. See Table S2.2 in the SI for the region legend.**

| REG | PM$_{2.5}$ FASST (µg m$^{-3}$) | PM$_{2.5}$ TM5 (µg m$^{-3}$) | NMB | MB(µg m$^{-3}$) | R$^2$ |
|---|---|---|---|---|---|
| | | FLE2030 | | | |
| EUR | 9.2 | 8.7 | 6% | 0.56 | 0.94 |
| NAM | 4.7 | 4.2 | 11% | 0.47 | 0.95 |
| EAS | 30.2 | 27.5 | 10% | 2.75 | 0.93 |
| SAS+SEA | 26.4 | 26.8 | -2% | -0.42 | 0.84 |
| RUS | 5.8 | 5.7 | 1% | 0.07 | 0.91 |
| SAM | 5.0 | 4.9 | 1% | 0.07 | 0.77 |
| MEA | 8.9 | 9.2 | -3% | -0.23 | 0.88 |
| AFR | 8.5 | 9.4 | -10% | -0.90 | 0.77 |
| | | MIT2030 | | | |
| EUR | 4.0 | 2.1 | 86% | 1.84 | 0.83 |
| NAM | 2.8 | 2.2 | 28% | 0.63 | 0.78 |
| EAS | 10.1 | 8.5 | 19% | 1.58 | 0.94 |
| SAS+SEA | 8.8 | 7.1 | 24% | 1.72 | 0.73 |
| RUS | 2.6 | 2.1 | 24% | 0.51 | 0.85 |
| SAM | 4.4 | 4.3 | 1% | 0.04 | 0.74 |
| MEA | 3.6 | 3.2 | 11% | 0.36 | 0.74 |
| AFR | 4.9 | 4.7 | 5% | 0.21 | 0.93 |
| | | FLE2030 - MIT2030 | | | |
| EUR | 5.3 | 6.6 | -20% | -1.28 | 0.97 |
| NAM | 1.8 | 2.0 | -8% | -0.16 | 0.93 |
| EAS | 20.1 | 18.9 | 6% | 1.17 | 0.93 |
| SAS+SEA | 17.6 | 19.7 | -11% | -2.14 | 0.85 |
| RUS | 3.2 | 3.6 | -12% | -0.44 | 0.85 |
| SAM | 0.6 | 0.6 | 6% | 0.03 | 0.13 |
| MEA | 5.4 | 6.0 | -10% | -0.59 | 0.77 |
| AFR | 3.6 | 4.8 | -23% | -1.11 | 0.47 |

**Table 9: Regional grid cell mean anthropogenic ozone health exposure metric 6mDMA1 and performance statistics for FASST vs. TM5, for the high (FLE2030) and low (MIT2030) emission scenarios, and for the delta. See Table S2.2 in the SI for the region legend.**

| REG | 6mDMA1 FASST (ppb) | 6mDMA1 TM5 (ppb) | NMB | MB (ppb) | $R^2$ |
|---|---|---|---|---|---|
| | | FLE2030 | | | |
| EUR | 55 | 53 | 4% | 2 | 0.98 |
| NAM | 57 | 53 | 7% | 4 | 0.96 |
| EAS | 69 | 57 | 21% | 12 | 0.93 |
| SAS+SEA | 92 | 76 | 20% | 15 | 0.96 |
| RUS | 53 | 50 | 6% | 3 | 0.98 |
| SAM | 42 | 38 | 9% | 3 | 0.92 |
| MEA | 72 | 70 | 4% | 3 | 0.95 |
| AFR | 59 | 55 | 7% | 4 | 0.94 |
| | | MIT2030 | | | |
| EUR | 49 | 43 | 13% | 6 | 0.95 |
| NAM | 50 | 41 | 22% | 9 | 0.95 |
| EAS | 50 | 44 | 13% | 6 | 0.94 |
| SAS+SEA | 51 | 46 | 11% | 5 | 0.90 |
| RUS | 44 | 40 | 11% | 4 | 0.99 |
| SAM | 35 | 31 | 12% | 4 | 0.90 |
| MEA | 55 | 51 | 9% | 4 | 0.89 |
| AFR | 48 | 44 | 8% | 3 | 0.96 |
| | | FLE2030 - MIT2030 | | | |
| EUR | 6 | 9 | -38% | -4 | 0.89 |
| NAM | 7 | 12 | -45% | -5 | 0.67 |
| EAS | 19 | 13 | 46% | 6 | 0.89 |
| SAS+SEA | 40 | 30 | 35% | 10 | 0.94 |
| RUS | 8 | 10 | -15% | -1.4 | 0.89 |
| SAM | 6 | 7 | -5% | -0.3 | 0.47 |
| MEA | 17 | 19 | -9% | -1.8 | 0.89 |
| AFR | 11 | 10 | 4% | 0.4 | 0.72 |

**Table 10: Contributions of emissions of $CH_4$, $NO_x$, CO and NMVOC to $O_3$ and $CH_4$ radiative forcing. Stevenson et al. (2013): for the period 1850-2000; Shindell et al. (2005, 2009) for the period 1750-2000. FASST: emission changes from Stevenson et al. (2013) multiplied with FASST global forcing efficiencies**

| | Stevenson et al., 2013 | Shindell et al., 2005 | Shindell et al., 2009 | TM5-FASST |
|---|---|---|---|---|
| | Contribution to $O_3$ forcing (mWm$^{-2}$) | | | |
| $CH_4$ | $166 \pm 46$ | $200 \pm 40$ | 275 | 211 |
| $NO_x$ | $119 \pm 33$ | $60 \pm 30$ | 41 | 35 |
| CO | $58 \pm 13$ | | 48 | 67 |
| NMVOC | $35 \pm 9$ | | 7 | 39 |
| | Contribution to $CH_4$ forcing (mWm$^{-2}$) | | | |
| $CH_4$ | $533 \pm 39$ | $590 \pm 120$ | 530 | 528 |
| $NO_x$ | $-312 \pm 67$ | $-170 \pm 85$ | -130 | -95 |
| CO | $57 \pm 9$ | | | 58 |
| NMVOC | $22 \pm 18$ | | | 38 |

**Table 11. Regional-to-global direct radiative forcing efficiencies for PM$_{2.5}$ precursors (mW/m² /Tg of annual emissions) for the larger source-receptor regions in earlier studies, and from FASST, aggregated to similar regional definitions. Values in brackets represent 1 standard deviation from the respective reported model ensembles.**

| | | NAM | EUR | SAS | EAS | RUS | MEA |
|---|---|---|---|---|---|---|---|
| Stjern et al., 2016 | BC | 52 (±21) | 55 (±22) | 94 (±38) | 55 (±16) | 78 (±47) | 202 (±323) |
| | POM | -8 (±6) | -7 (±4) | -10 (±6) | -5 (±3) | -2 (±5) | -18 (±7) |
| | SO$_4$ (SO$_2$) | -5 (±2) | -6 (±2) | -8 (±4) | -4 (±1) | -4 (±1) | -10 (±7) |
| Yu et al., 2013 | BC | 27 (±15) | 37 (±19) | 25 (±15) | 28 (±20) | | |
| | POM | -4 (±2) | -4 (±2) | -4 (±2) | -4 (±2) | | |
| | SO$_4$ (SO$_2$) | -4 (±1) | -4 (±1) | -4 (±1) | -3 (±1) | | |
| FASST (RCP2000) | BC | 17 | 19 | 19 | 16 | 25 | 43 |
| | POM | -6 | -4 | -6 | -5 | -4 | -9 |
| | SO$_4$ (SO$_2$) | -3 | -3 | -4 | -2 | -2 | -7 |

**Table 12. Regional-to-global direct radiative forcing efficiencies for $O_3$ precursors (mW/m²/Tg of annual emissions) for the larger source-receptor regions in earlier work, and from FASST, aggregated to similar regional definitions, including direct $O_3$ forcing, feedbacks on $CH_4$ and long-term $O_3$ forcing from the latter. Values in brackets represent reported 1 standard deviation from the model ensemble in Fry et al., 2012.**

|  |  | East-Asia | Europe | N-America | South-Asia |
|---|---|---|---|---|---|
| Fry et al., 2012 | $NO_x$ | -0.31 (±0.6) | -0.80 (±0.5) | -0.53 (±0.6) | -1.17 (±2.2) |
|  | NMVOC | 0.50 (±0.2) | 0.45 (±0.2) | 0.47 (±0.2) | 0.72 (±0.2) |
|  | CO | 0.15 (±0.02) | 0.13 (±0.02) | 0.16 (±0.02) | 0.15 (±0.02) |
|  |  |  |  |  |  |
| FASST (RCP200) | $NO_x$ | -0.44 | -0.33 | -0.35 | -1.43 |
|  | NMVOC | 0.60 | 0.57 | 0.61 | 0.74 |
|  | CO | 0.18 | 0.15 | 0.15 | 0.19 |

**Table 13: Global GWP and GTP values 95% CI range (excluding Indirect Radiative Effects) from IPCC AR5 (Forster et al., 2007), and from FASST based on RCP year 2000 emissions and the regional forcing efficiencies listed in Table A6.2 of the SI (all numbers rounded to 2 significant figures).**

| | GWP20 | | GWP100 | | GTP20 | | GTP100 | |
|---|---|---|---|---|---|---|---|---|
| | AR5 | FASST | AR5 | FASST | AR5 | FASST | AR5 | FASST |
| $CH_4$ | (70, 98) | 78 | (24, 33) | 29 | (56, 79) | 66 | (3.6, 5.0) | 3.9 |
| BC | (940, 4100) | 880 | (257, 1100) | 240 | (270, 1200) | 340 | (35, 150) | 37 |
| OC | (-410, -89) | -280 | (-114, -25) | -77 | (-120, -26) | -110 | (-16, -3) | -12 |
| $SO_2$ | (-210,-70) | -150 | (-58, -19) | -40 | (-61, -20) | -57 | (-8, 38) | -6.2 |
| VOC | (8.3, 20) | 21 | (2.7, 6.3) | 7 | (4.4, 11) | 11 | (0.4, 0.9) | 1.2 |
| $NO_x$ | (12, 26)[a] (-220, -440)[b] | -31 | (-15, -7)[a] (-130, -64)[b] | -14 | (-120, -57) | -100 | (-3.9, -1.9) | -8 |
| CO | (6.0, 7.8) | 7.9 | (2, 3) | 2.6 | (3.7, 6.1) | 6.3 | (0.27, 0.55) | 0.42 |

a    Fuglestvedt et al. (2010)

b    Shindell et al., (2009)

**Table 14** Overview of previous studies on health impact of $PM_{2.5}$, together with FASST results for 2 different scenarios. Uncertainty ranges are as reported in the respective studies. The uncertainty range on FASST results includes the RR uncertainty only (Fig. S5.1 in the SI)

| Reference | Year evaluated | Method | threshold | Exposure - response function | Global deaths (millions) |
|---|---|---|---|---|---|
| | | Excluding mineral dust | | | |
| Fang et al., 2013 | 2000 | CTM | no | K2009[a] | 1.6 (1.2 – 1.9) |
| Silva et al., 2013 | 2000 | CTM | no | K2009 | 2.1 (1.3 -3.0) |
| Anenberg et al., 2010 | 2000 | CTM | 5.8µg m$^{-3}$ | K2009 | 2.7 (2.0 -3.4) |
| Evans et al., 2013 | 2004 | SAT | 5.8µg m$^{-3}$ | K2009 | 2.7 (1.9 - 3.5) |
| Lelieveld et al., 2013 | 2005 | CTM | no | K2009 | 2.2 (2.1 - 2.3) |
| **FASST (RCP)** | **2000** | **FASST** | **~7.3µg m$^{-3}$** | **K2009** | **2.5 (1.2 – 3.6)** |
| **FASST (RCP)** | **2000** | **FASST** | **~7.3µg m$^{-3}$** | **B2014[b]** | **2.1 (1.0 – 3.0)** |
| | | Including mineral dust | | | |
| Silva et al., 2016 | 2000 | ACCMIP CTM ensemble | ~7.3µg m$^{-3}$ | B2014 | 1.7 (1.3 – 2.1) |
| Evans et al. 2013 | 2004 | SAT | 5.8µg m$^{-3}$ | K2009 | 4.3 (2.9 – 5.4) |
| Lelieveld et al., 2015 | 2010 | CTM | ~7.3µg m$^{-3}$ | B2014 | 3.2 (1.5 - 4.6) |
| GBD2010 (Lim et al., 2012) | 2010 | Fused (FASST + SAT + ground based) | ~7.3µg m$^{-3}$ | B2014 | 3.2 (2.8 -3.6) |
| GBD2013 (Forouzanfar et al., 2015) | 2013 | Fused (FASST + SAT + ground based) | ~7.3µg m$^{-3}$ | B2014 | 2.9 (2.8 – 3.1) |
| GBD2015 (Cohen et al., 2017) | 2015 | Fused (FASST + SAT + ground based) | ~4.1µg m$^{-3}$ | B2014 | 4.2 (3.7 – 4.8) |
| **FASST (RCP)** | **2000** | **FASST** | **~7.3µg m$^{-3}$** | **K2009** | **3.6 (2.7 -4.5)** |
| **FASST (RCP)** | **2000** | **FASST** | **~7.3µg m$^{-3}$** | **B2014** | **2.6 (1.2 – 3.8)** |
| **FASST (HTAP2)** | **2010** | **FASST** | **~7.3µg m$^{-3}$** | **B2014** | **4.1 (2.0 - 5.9)** |

5 (a) Krewski et al., 2009
(b) Burnett et al., 2014

**Table 15 Overview of previous studies on long-term health impact of ozone, together with FASST results for 2 different scenarios**

| Ref | year | Method | threshold | Exposure-response function | Global deaths (thousands) |
|---|---|---|---|---|---|
| Anenberg et al., 2010 | 2000 | CTM | 33.3 | J2009 [a] | 470 (182 - 758) |
| Silva et al., 2013 | 2000 | ACCMIP CTM ensemble | 33.3 | J2009 | 380 (117 -750) |
| Lelieveld et al., 2015 | 2010 | CTM | ~37.6 | J2009 | 142 (90 -208) |
| GBD 2010 (Lim et al., 2012) | 2010 | FASST | ~37.6 | J2009 | 152 (52 – 270) |
| GBD 2013 (Forouzanfar et al., 2015) | 2013 | FASST | ~37.6 | J2009 | 217 (161 – 272) |
| GBF 2015 (Cohen et al., 2017) | 2015 | FASST | ~37.6 | J2009 | 254 (97 – 422) |
| **FASST (RCP)** | **2000** | **FASST** | **33.3** | **J2009** | **197 (66 – 315)** |
| **FASST (HTAP2)** | **2010** | **FASST** | **33.3** | **J2009** | **340 (116 – 544)** |

(a)  Jerrett et al., 2009

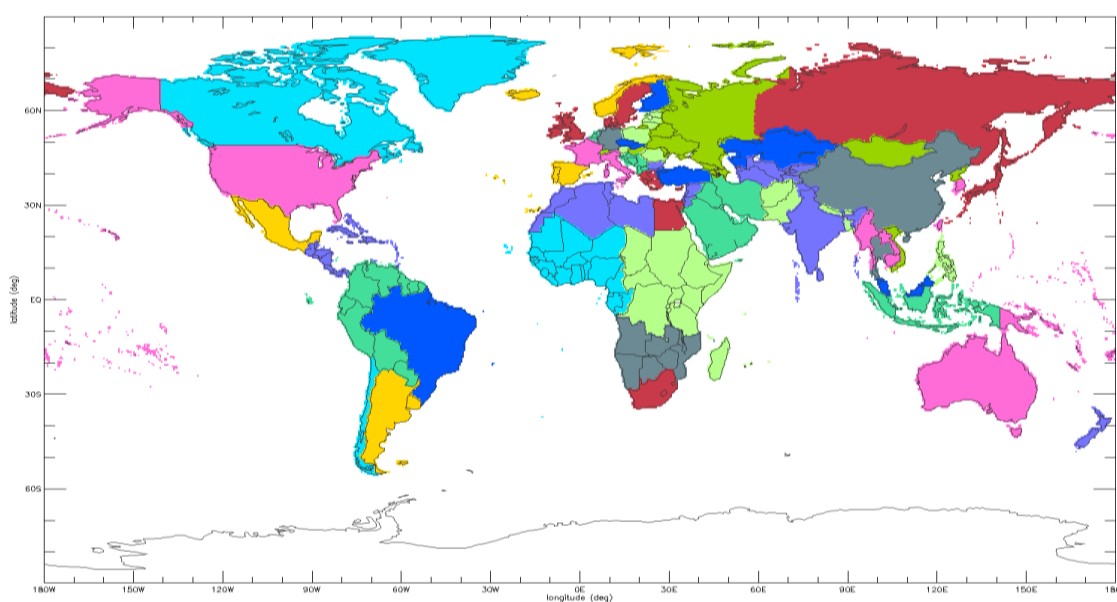

**Figure 1: 56 continental emission source regions in TM5-FASST. See Table S2.2 in the SI for the mapping between regions and countries**

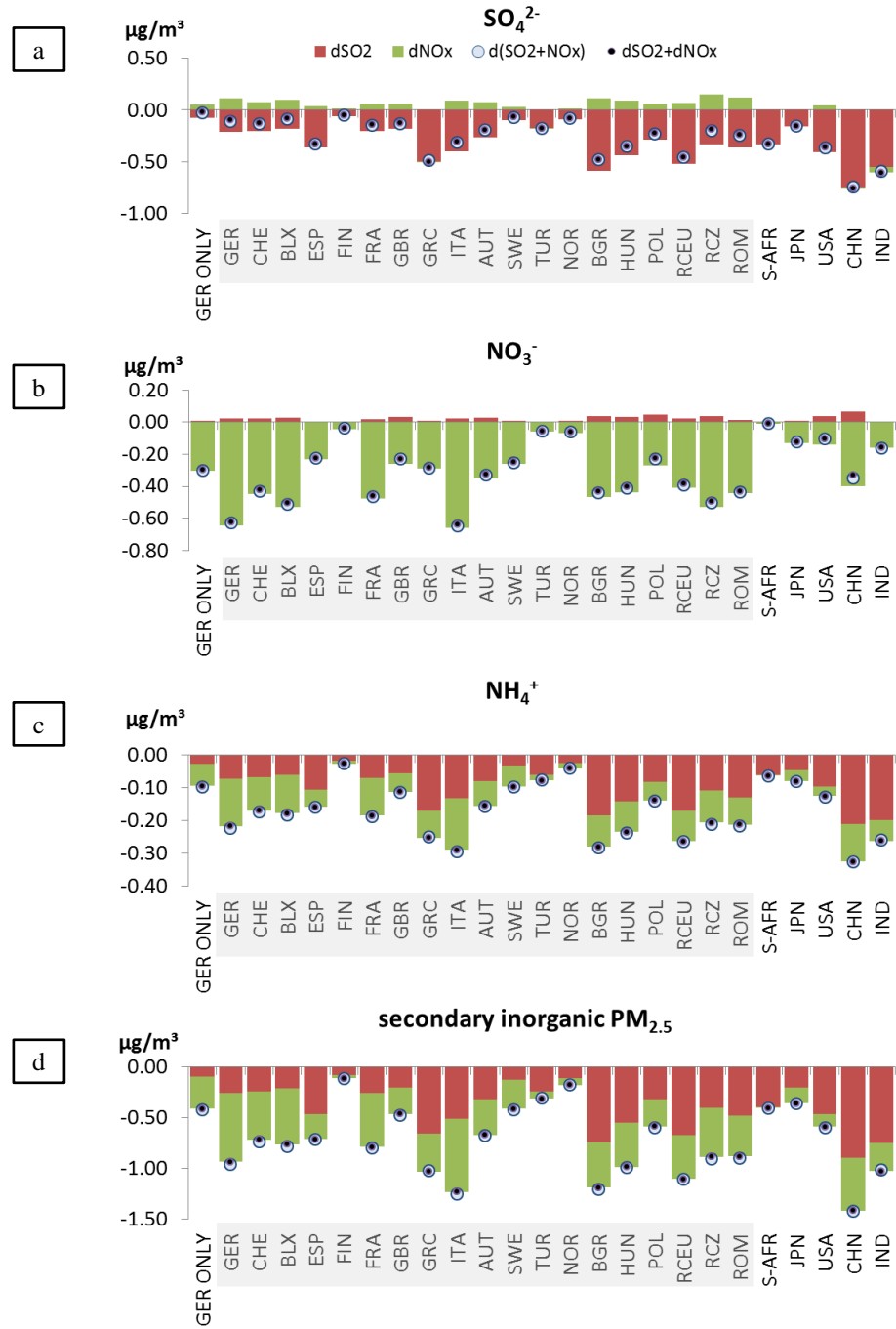

**Figure 2: TM5-CTM response in annual population-weighted mean sulfate (a), nitrate (b), ammonium (c) and total inorganic secondary PM₂.₅ (d) (as sum of the 3 components) upon emitted precursor perturbation of -20% for selected source regions (see SI table S2.2 for the region codes legend). Only the concentration change inside each source region is shown. Red bars: SO₂–only**

**perturbation (simulation P2); green bars: NO$_x$-only perturbation (simulation P3). Open circles: simultaneous (SO$_2$ + NO$_x$) perturbation (simulation P1). Black dots: P2 + P3. Shaded regions are perturbed simultaneously as one European region.**

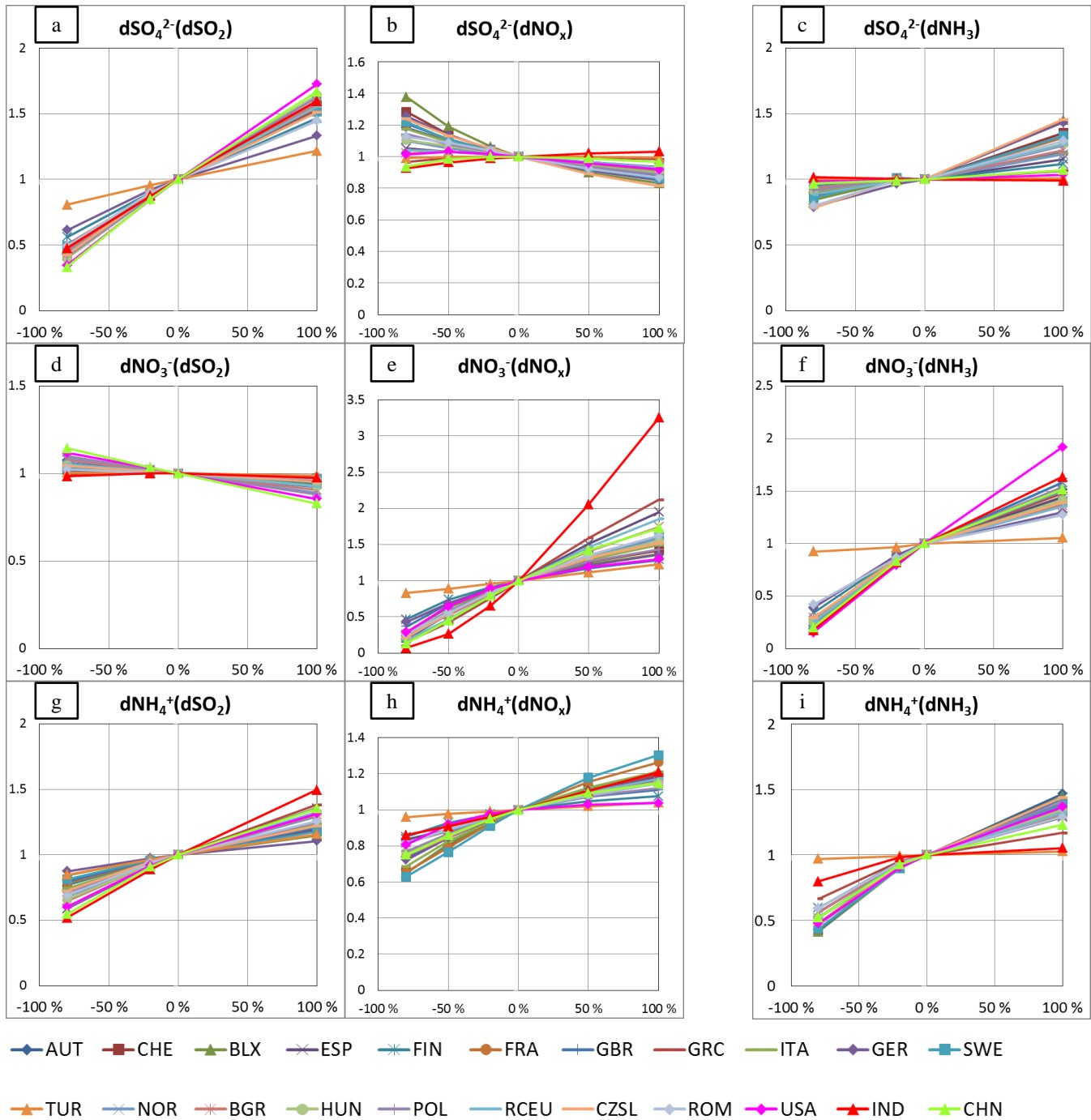

**Figure 3: TM5-CTM change in population-weighted regional mean secondary PM$_{2.5}$ components SO$_4^{2-}$ (a to c), NO$_3^-$ (d to f), NH$_4^+$ (g to i), relative to their respective base scenario concentration, as a function of precursor SO$_2$ (a, d, g), NO$_x$ (b, e, h) and NH$_3$ (c, f, i) emission perturbation strength for European receptor regions, USA, India and China. Perturbations were applied over all European regions simultaneously.**

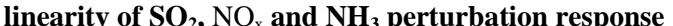

**Figure 4: Regional Secondary PM$_{2.5}$ (SO$_4^{2-}$+NO$_3^-$+NH$_4^+$) response to -80% and +100% single precursor emission perturbations for SO$_2$ (a), NO$_x$ (b), NH$_3$ (c) as well as the combined SO$_2$ + NO$_x$ perturbation (d). X-axis: Full TM5 model; Y-axis: Linear extrapolation of -20% perturbation (FASST approach). Each point corresponds to the population-weighted mean concentrations over a receptor region.**

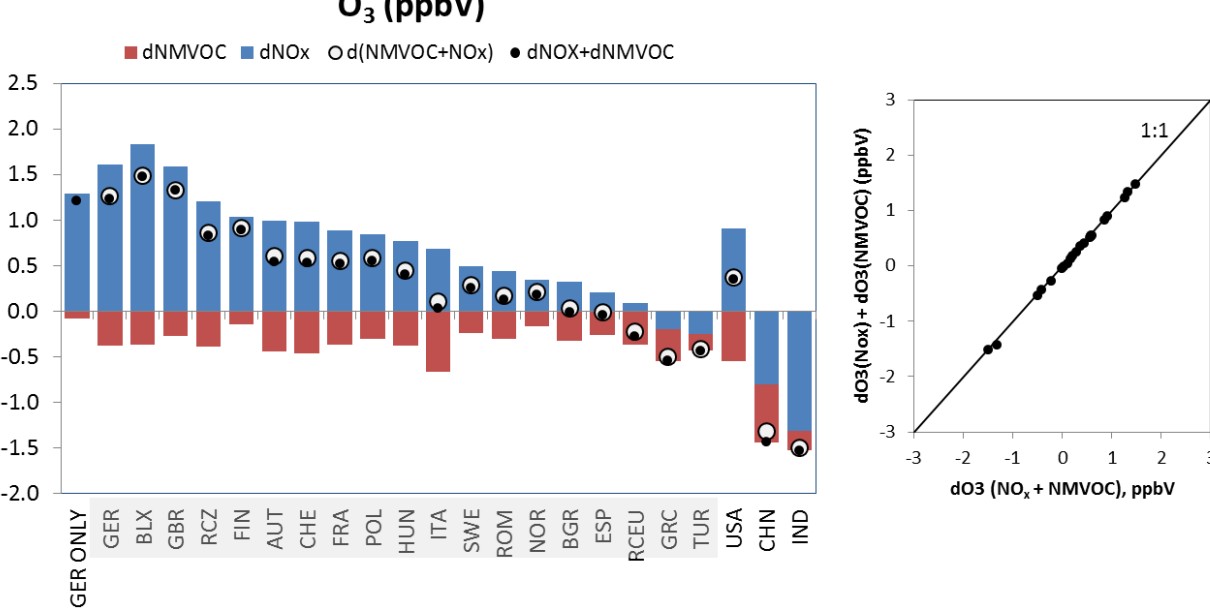

**Figure 5: TM5-CTM response in annual mean population-weighted O$_3$ concentration (in ppbV) upon emitted precursor perturbation of -20% for selected source receptor regions. European regions were perturbed simultaneously. Red bar: response form NMVOC–only perturbation (simulation P4); blue bar: response form NO$_x$-only perturbation (simulation P3). Open circles: response from simultaneous (NMVOC + NO$_x$) perturbation (simulation P5). Black dots: sum of individual responses. Shaded regions are perturbed simultaneously as one European region. Right panel: scatter plot between O$_3$ response to combined and summed individual responses.**

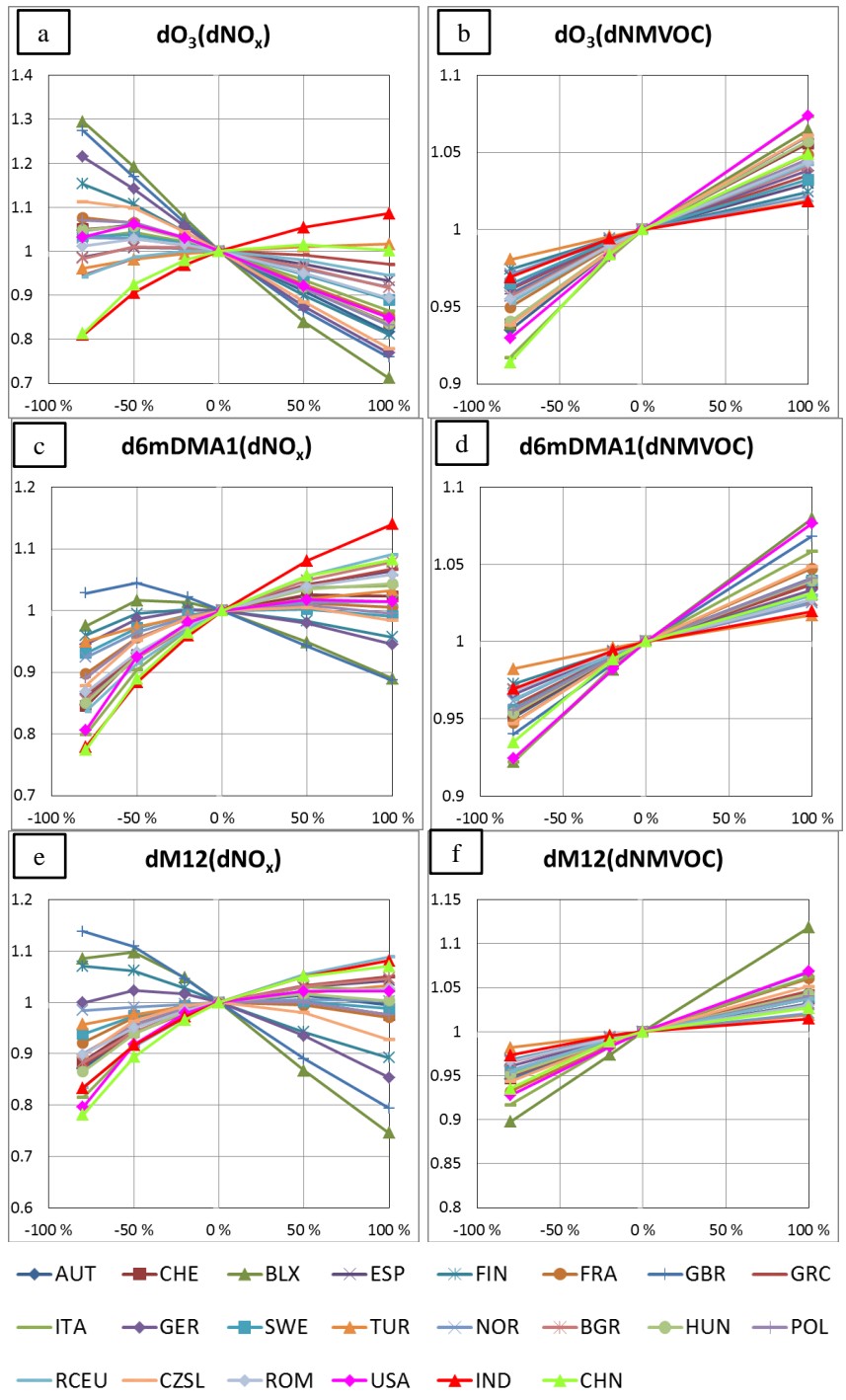

**Figure 6: TM5-CTM response in population weighted annual mean O₃ (a, b)and health exposure metric 6mDMA1 (c, d) , and in grid cell-area-weighted crop exposure metric M12 (e, f), relative to their respective base simulation values, as a function of**

**precursors $NO_x$ (a, c, e) and NMVOC (b, d, f) emission perturbation strength. European regions are perturbed simultaneously as one region.**

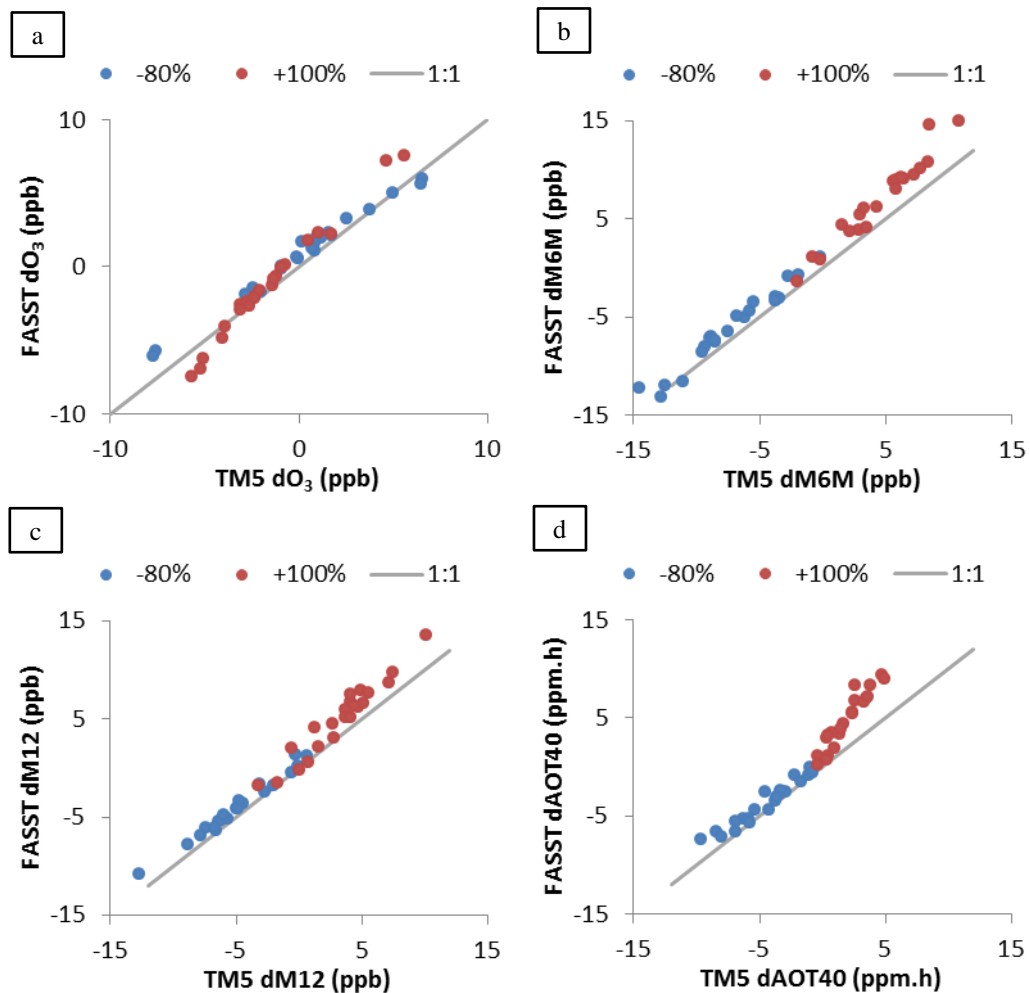

**Figure 7: Regional O₃ and O₃ exposure metrics responses to combined -80% and +100% precursor emission perturbations of NOₓ and NMVOC. (a) annual mean population-weighted O₃; (b) population-weighted 6mDMA1; (c) area-mean M12; (d) area-mean AOT40  X-axis: Full TM5 model; Y-axis: Linear extrapolation of -20% perturbation (FASST approach). Each point corresponds to the mean metric over a source region.**

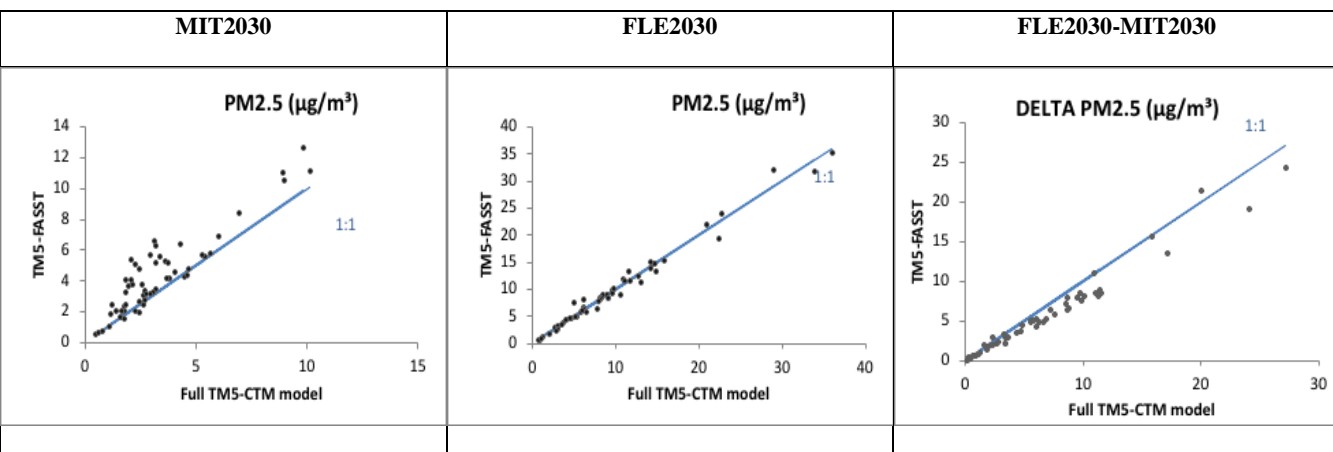

**Figure 8: Population-weighted mean PM$_{2.5}$ concentration computed with TM5-FASST versus TM5-CTM for low emission scenarios MIT2030 (left), high emission scenario FLE2030 (middle) and the change between the two. Each point represents the population-weighted mean over a TM5-FASST receptor region. Blue line: 1:1 relation.**

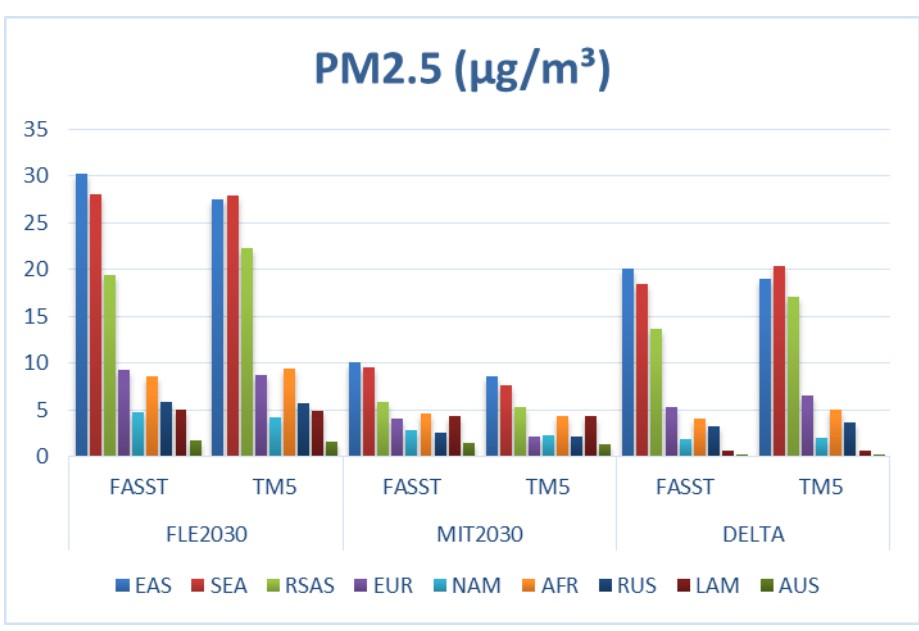

**Figure 9 Total population-weighted anthropogenic PM₂.₅ over larger FASST zoom areas, for the high (FLE2030) and low (MIT2030) emission scenarios, and the difference (delta) between both, computed with the full TM5 model and with FASST**

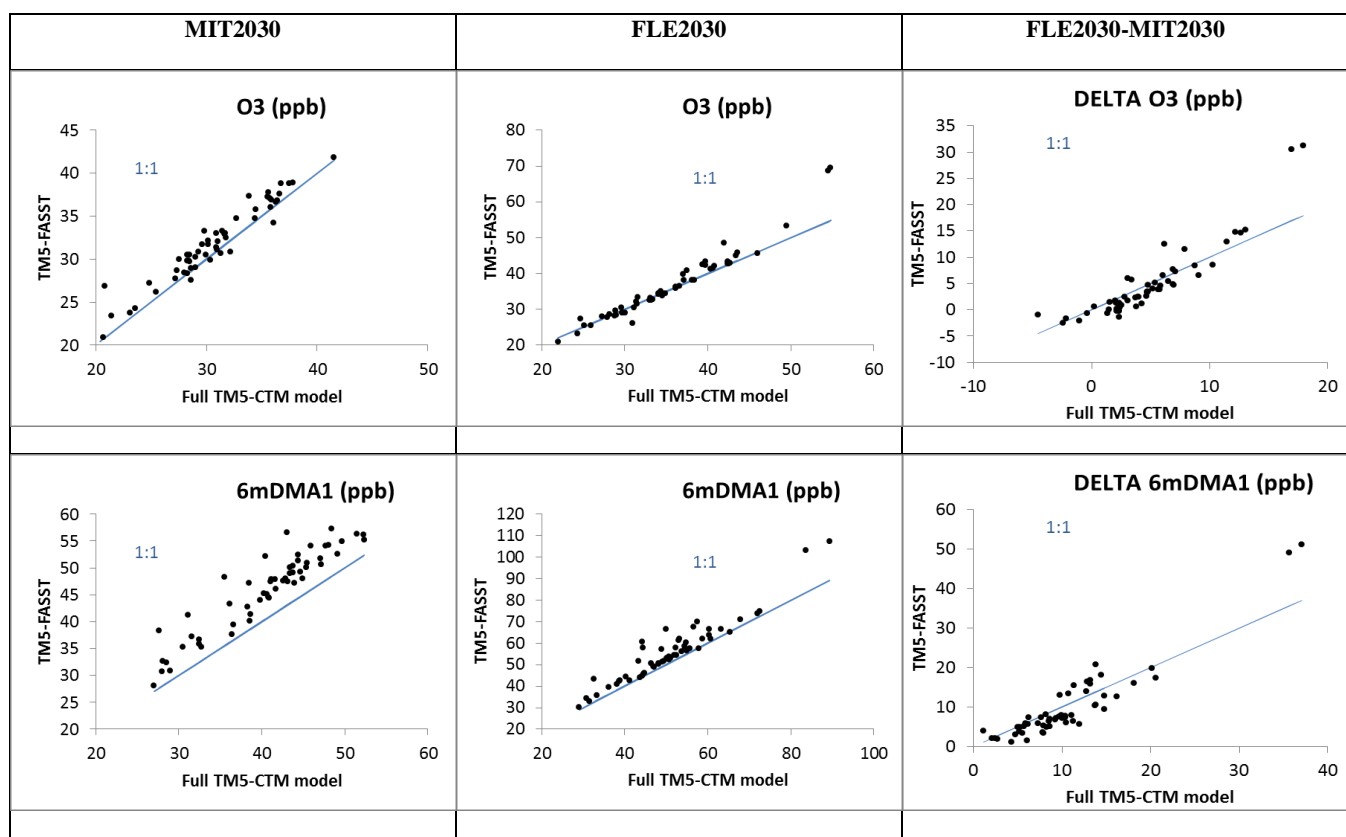

**Figure 10: Population-weighted mean annual ozone (top) and ozone exposure metric 6mDMA1 (bottom) computed with TM5-FASST versus TM5-CTM for low emission scenarios MIT2030 (left), high emission scenario FLE2030 (middle) and the change between the two (right). Each point represents the population-weighted mean over a TM5-FASST receptor region. Blue line: 1:1 relation.**

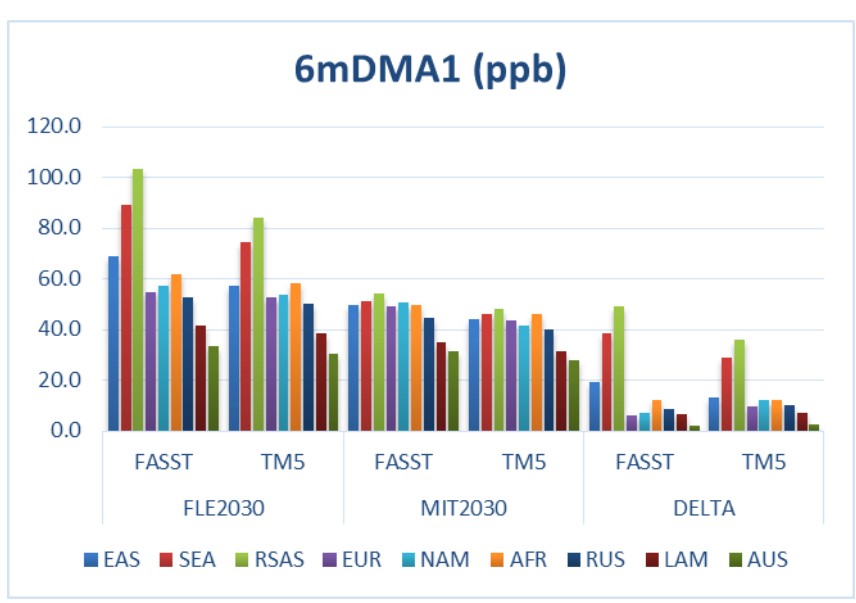

**Figure 11: Total population-weighted anthropogenic PM$_{2.5}$ over larger FASST zoom areas, for the high (FLE2030) and low (MIT2030) emission scenarios, and the difference (delta) between both, computed with the full TM5 model and with FASST**

Premature mortalities from PM$_{2.5}$

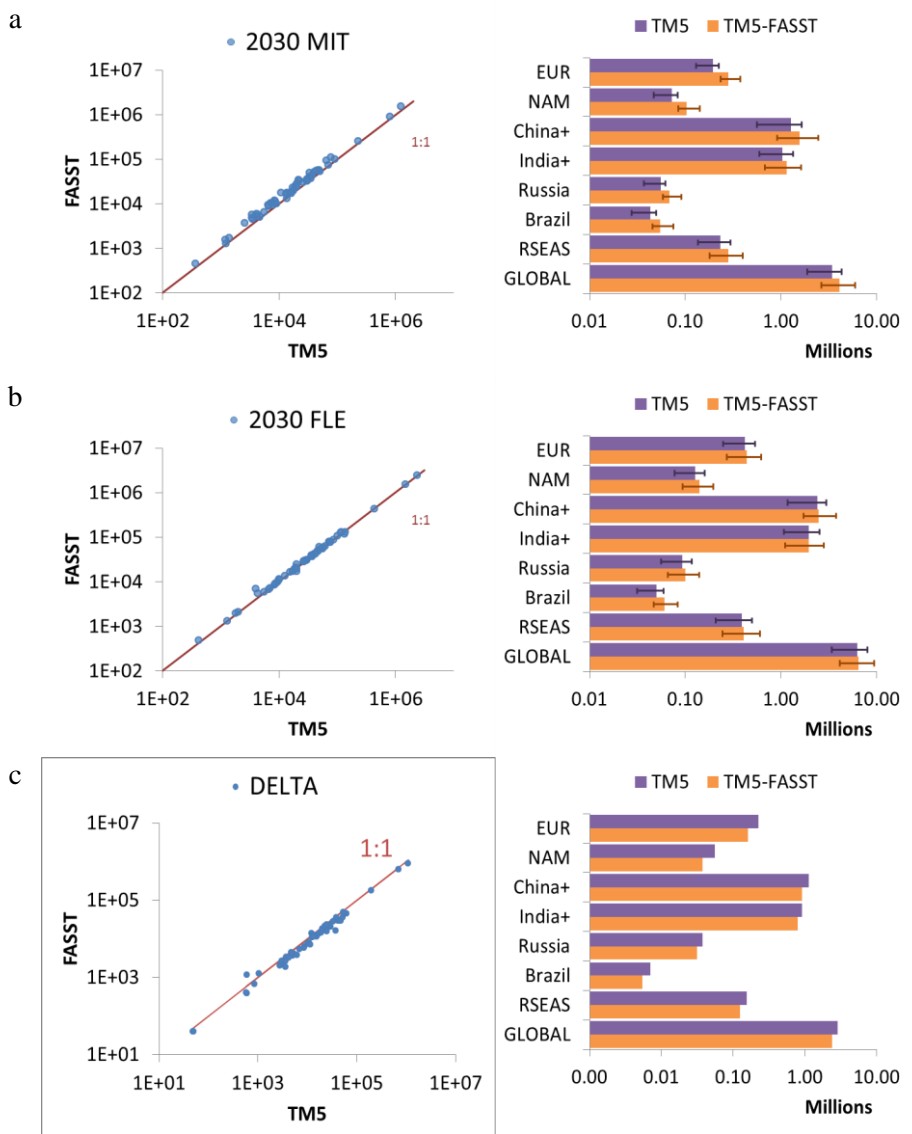

**Figure 12: FASST versus TM5 premature mortalities from exposure to PM$_{2.5}$ for MIT2030 (a) and FLE2030 (b) scenarios and the delta between both (c). Dots: aggregated over each FASST region. Bar plots: totals for selected world regions and global total. Error bars represent the 95% CI on the RR from the exposure-response function by Burnett et al. (2014)**

# Premature mortalities from O$_3$

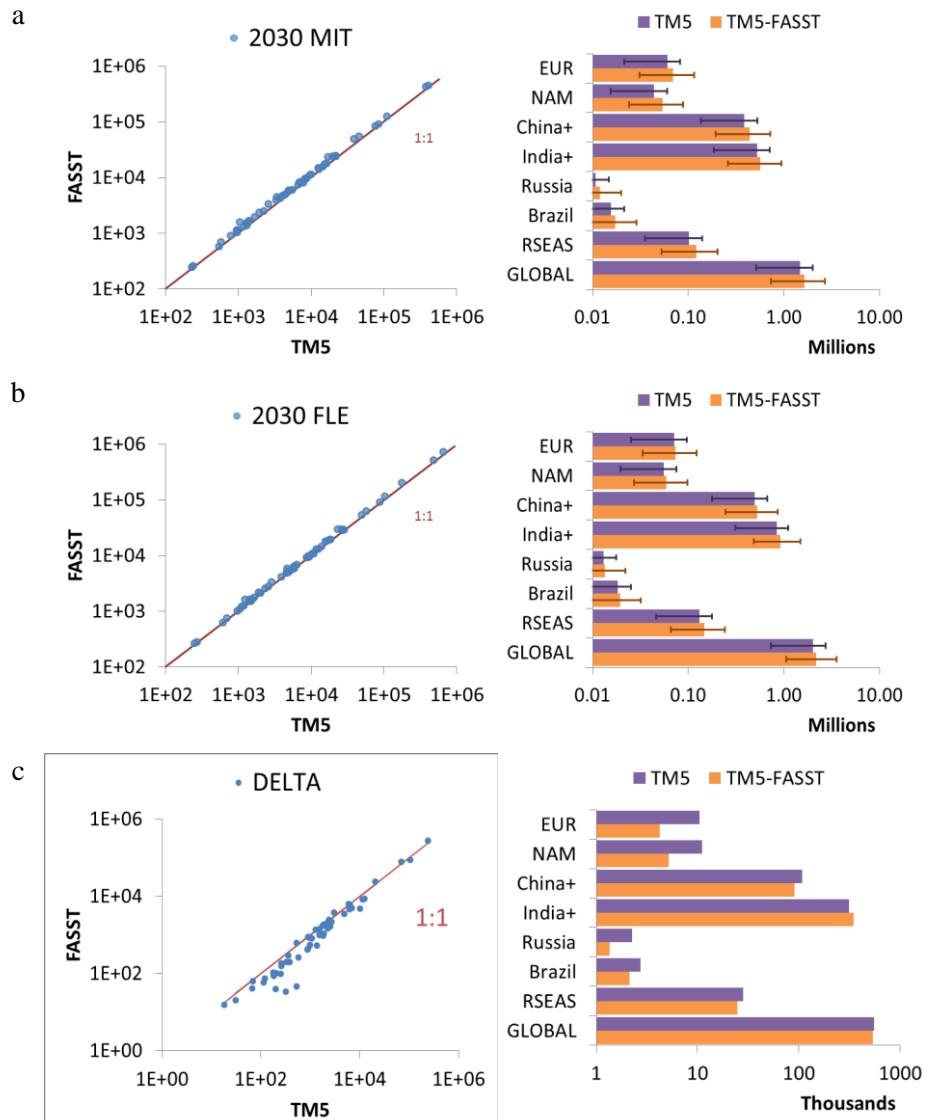

**Figure 13: FASST versus TM5 premature mortalities from exposure to O$_3$ for MIT2030 (a) and FLE2030 (b) scenarios and the delta between both (c). Dots: aggregated over each FASST region. Bar plots: totals for selected world regions and global total. Error bars represent the 95% CI on the exposure-response function (Jerrett et al., 2009).**

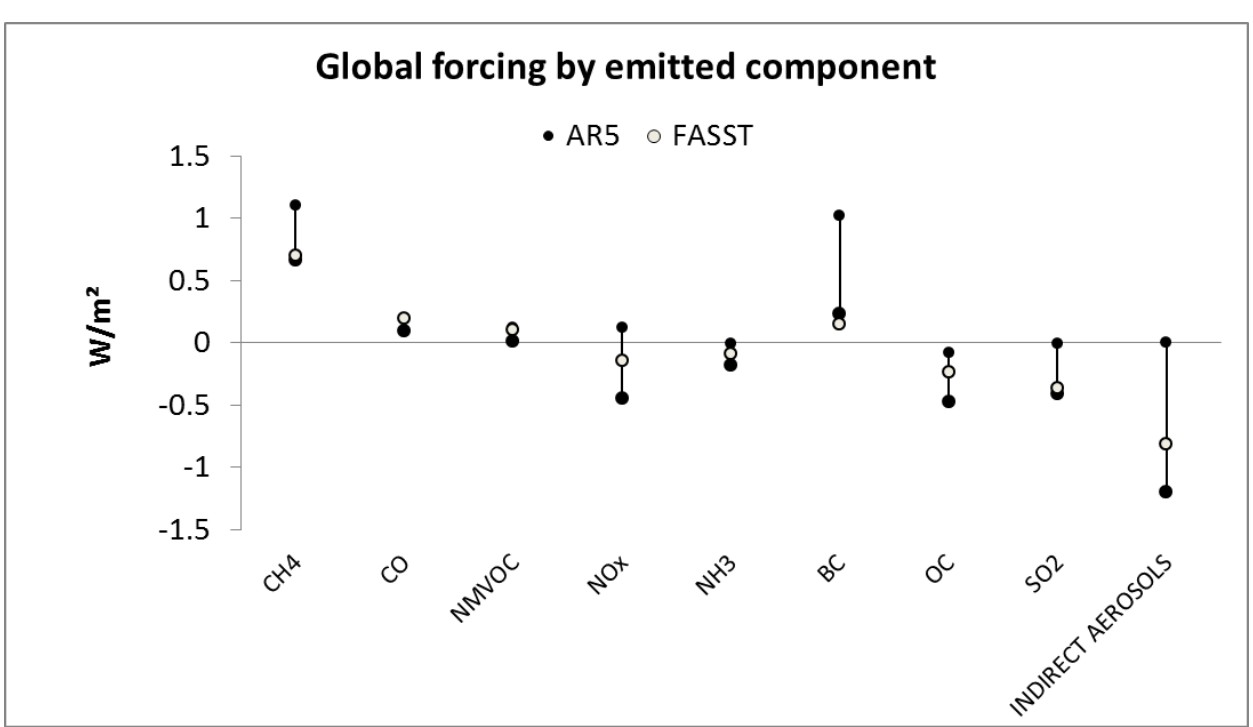

**Figure 14: Global anthropogenic radiative forcing by emitted component, from TM5-FASST forcing efficiencies applied on RCP (year 2000 anthropogenic emissions), and range of best anthropogenic forcings from AR5 (change over period 1750 – 2011)**

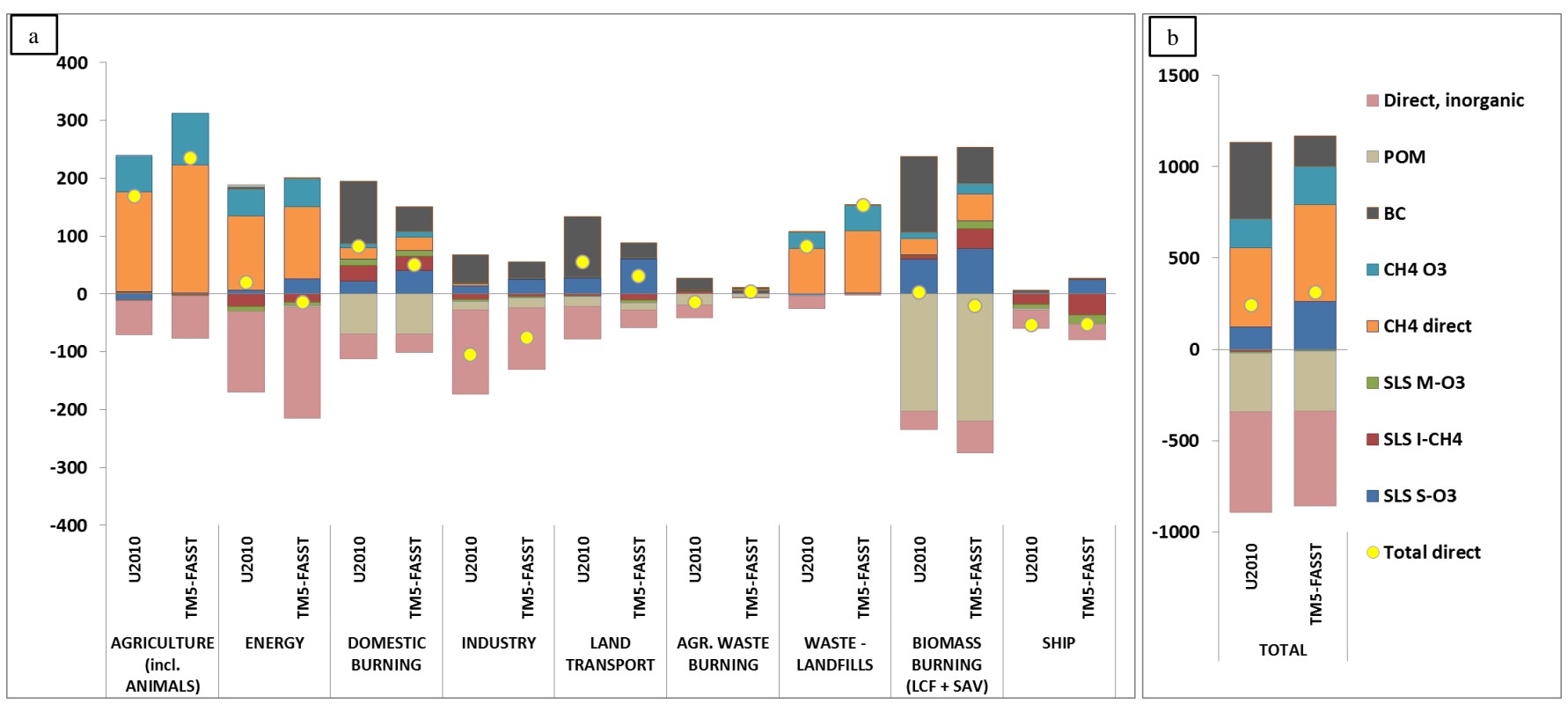

**Figure 15: Year 2000 radiative forcing from Unger et al. (2010), based on EDGAR year 2000 emissions and from TM5-FASST applied to RCP year 2000 (a) break-down by sector and by forcing component. Biomass burning includes both large scale fires and savannah burning; (b) total over all sectors. SLS S-O$_3$: direct contribution of short-lived species (SLS) to O$_3$; SLS I-CH$_4$: indirect contribution from SLS to CH$_4$; SLS M-O$_3$: indirect feedback from SLS on background ozone via the CH$_4$ feedback. CH$_4$ O$_3$: feedback of emitted CH$_4$ on background O$_3$**

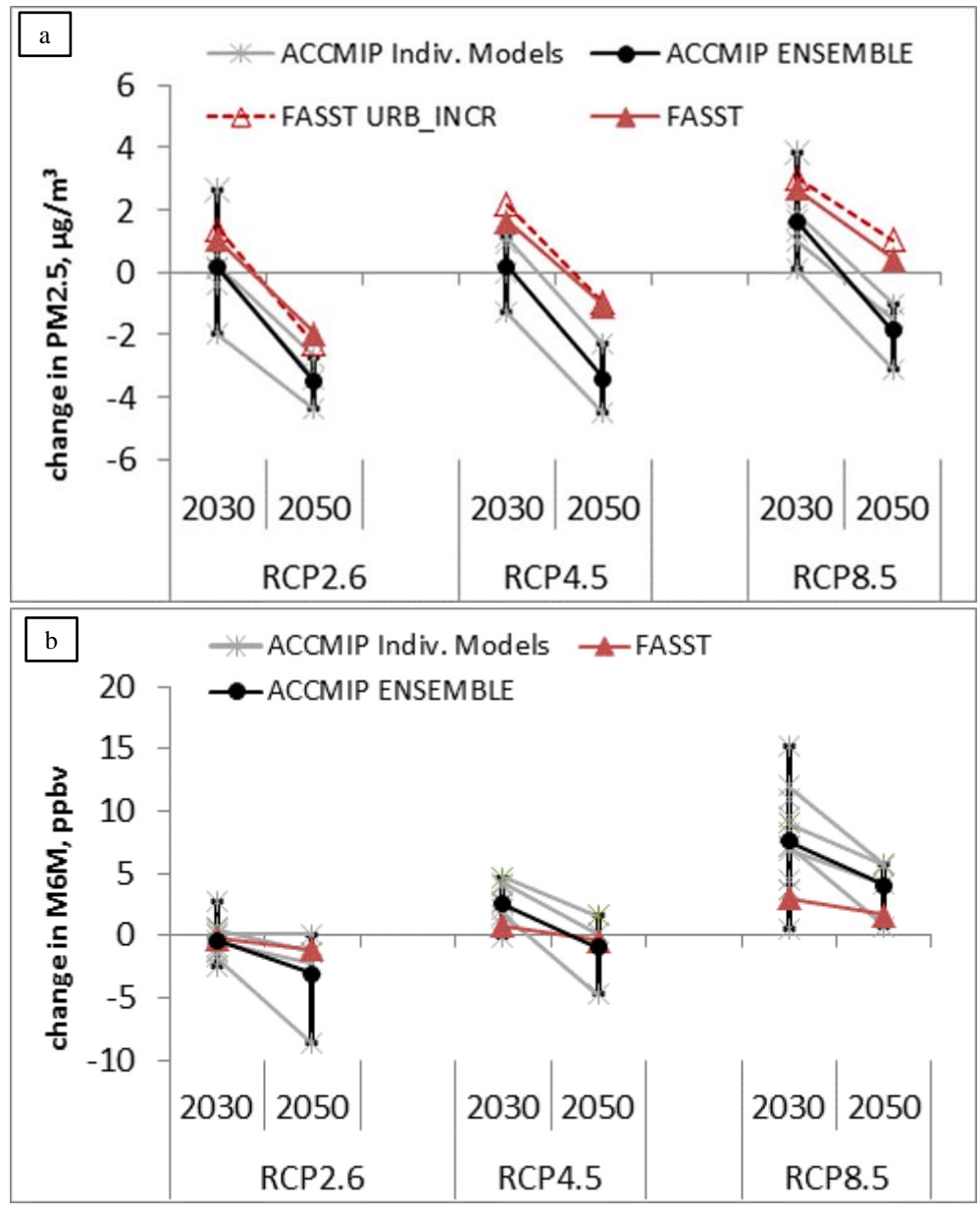

**Figure 16: Global population-weighted differences (scenario year minus year 2000) (a) in annual mean PM$_{2.5}$ concentrations and (b) in O$_3$ exposure metric 6mDMA1 for 3 RCP scenarios in each future year, from the ACCMIP model ensemble (Silva et al., 2016) (black symbols and lines) and TM5-FASST_v0 (red symbols and lines). FASST URB_INCR: including the urban increment correction. Grey symbols: results from individual ACCMIP models. Grey lines connect results from a single model. Not all models have provided data for all scenarios. ACCMIP error bars represent the range (min, max) across the ACCMIP ensemble.**

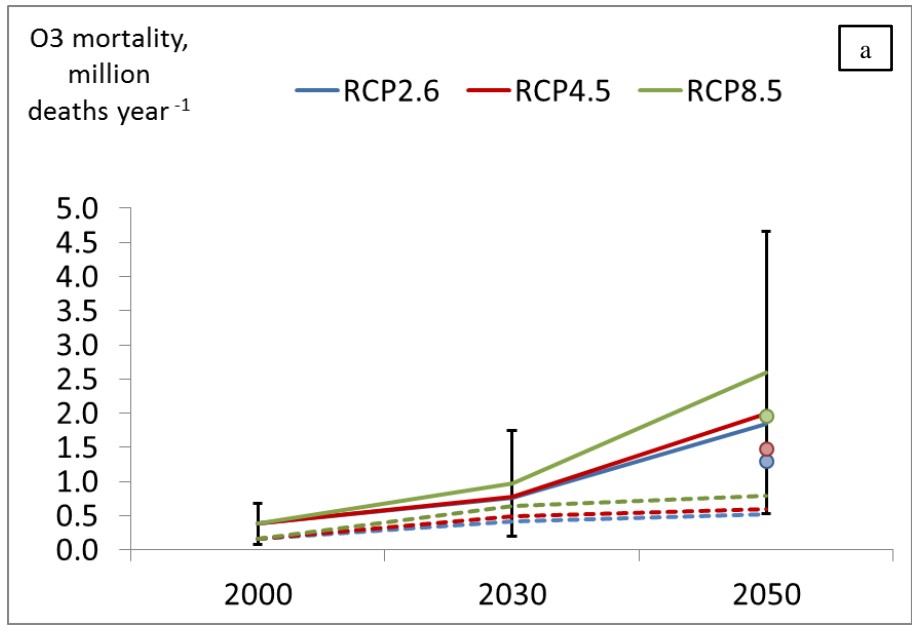

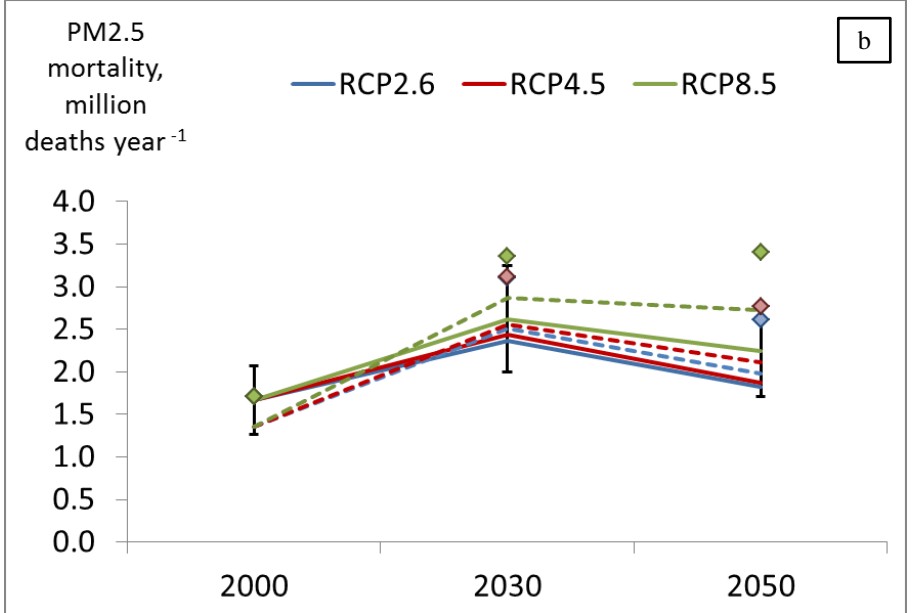

Figure 17. Trends in global burden on mortality of ozone (a) and PM$_{2.5}$ (b) from year 2000 to 2050 from the ACCMIP multi-model ensemble (Silva et al., 2016) (full lines) and TM5-FASST (dashed lines) for 3 RCP scenarios. The error bar on the year 2000 is the ACCMIP 95% CI including uncertainty in RR and across models. CI for 2030 and 2050 were not provided by ACCMIP, we use here the same relative error as for year 2000. Dots (O$_3$ mortality): adjusted TM5-FASST ozone mortalities for RCP 2050, using baseline respiratory mortalities consistent with Silva et al. (2016). Diamonds (PM$_{2.5}$ mortality): TM5-FASST estimate including the urban increment parameterization

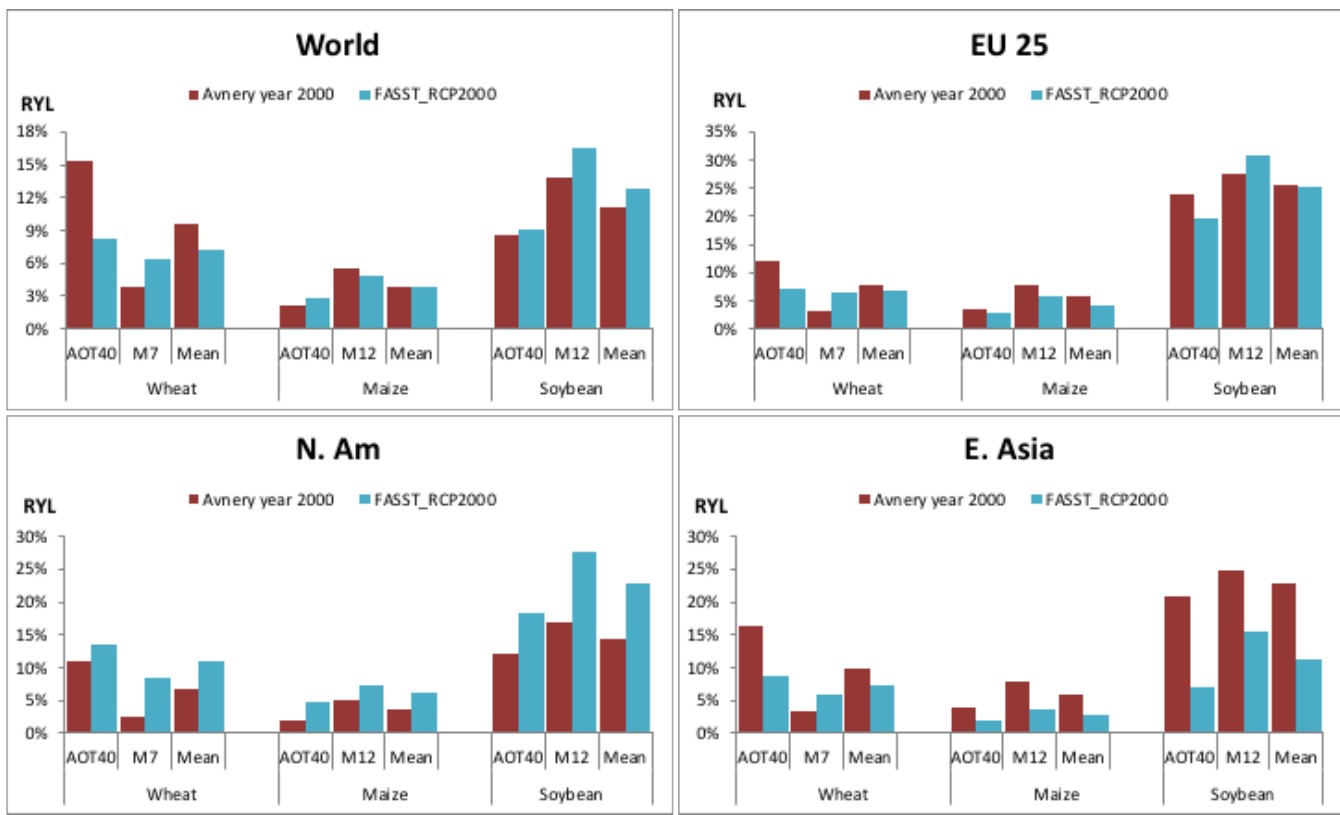

**Figure 18:Year 2000 global and regional ozone-induced relative yield losses for 3 major crops, from Avnery et al. (2011) and from TM5-FASST (RCP year 2000), estimated from the 2 common exposure metrics M7 and AOT40 (see text), as well as the mean of both.**