# Peer review of "TM5-FASST: a global atmospheric source-receptor model for rapid impact analysis of emission changes on air quality and short-lived climate pollutants"

_Atmospheric Chemistry and Physics, 2018_

## Referee Comment (RC1) · Anonymous Referee #1 · 27 Mar 2018

The manuscript presents a detailed summary of the methodology and validation for the TM5-FASST screening tool. TM5-FASST is a simplified tool that uses linear source-receptor relationships of air pollutant precursor species across 56 geographical source regions (plus aviation and shipping) to calculate the response in air pollutant concentrations at both the surface and 25 vertical layers in the atmosphere. The difference in concentrations can then be used to calculate the change in a number of air pollution impact metrics related to human health, climate and crop production. The tool allows for the impact from different emissions scenarios to be explored without the need to

run more detailed composition climate models. The manuscript provides a through description of the underlying methodology of TM5-FASST as well as an evaluation of the air pollutant predictions and impact metrics against a number of different sources. It provides a good reference for the TM5-FASST tool for use in future studies.

Major General Comments

1. Whilst I understand that TM5-FASST is not meant to replicate full scale model simulations, it would be good to bring together the limitations together into a more coherent section, possibly within the discussion section. Throughout the manuscript specific sections of the text mention aspects that TM5-FASST will not be able to predict e.g. changing spatial distribution of emissions and chemical regime. It would make sense for the reader to have these all in one place. Also I found little mention of how the fixed meteorological year of 2001 could potentially impact the prediction of pollutants in the future i.e. how would climate change affect predictions of future pollutants? Also the basis for the radiative forcing calculations is from a fixed meteorological year of 2001 and could have implications for the future calculation of effects. A more detailed mention of these issues would be good, perhaps in Section 4.

2. TM5-FASST and the validation of it using TM5 simulations have all been conducted using emissions inventory for the year 2000 as a baseline along with 20% perturbations from this base. How appropriate is it to use a base year of 2000 for validation purposes given the large recent changes in emissions over the last 10-15 years, particularly over East Asia where some emissions have changed by >20%. What impact would using more up to date emissions in the base scenario have the calculated source-receptor coefficients and would it significantly affect the magnitude of future predictions? It would be useful to provide information on how recent changes in emissions could impact TM5-FASST.

3. In Section 2.1, P4, Line 12 the manuscript mentions about the advent of finer resolution global models nearing 1°x1° horizontal resolution. I think it would be good

to make more comment on the applicability of the 1°x1° resolution when calculating country scale impacts. Is this resolution along with input information at similar resolution (e.g. emissions) sufficient to capture changes in pollutants at sub 100 km scales over countries such as the UK and Belgium/Luxembourg. I think that the urban adjustment of PM2.5 is a suitable attempt at this but I think it would good to have some further comment on the issue of resolution and the limitations provided by other inputs at this resolution e.g. emissions and meteorology.

4. Section 2.5 on health impacts provides a lot of details and is quite long compared to some others sections where most of the details are within a supplementary section. Also I found it a bit confusing to have two options for calculating PM2.5 health effects: the log-linear and integrated exposure-response functions (IER). I assume the output from FASST is only provided from one (Figure 15)? The paragraph on page 10 Lines 8 to 13 does not seem to provide clarity on which method is preferred and could be re-worded. Therefore Section 2.5 could be potentially made more concise by removing the details on the log-linear method to the supplementary. This would allow the main text to focus more on the IER method by Burnett et al., (2014), which is the current methodology used within the Global Burden of Disease study.

5. In section 3.1.1 when making a comparison of the additivity of emission perturbations for PM2.5 individual changes for SO2, NOx and NH3 is shown on Figure 3 and 4 but in Figure 2 there is no effect from NH3 emissions. Whereas, in Figure S7.1 and S7.2 the 3 individual responses are shown along with the combined response on PM2.5 (sum of all 3). However, the effect for combined emissions is only for SO2 and NOx in Figure 2 and 4 and does not include any addition from NH3. Why has the contribution from NH3 not been included within some of the combined emission changes in PM2.5? There seems to be a bit of inconsistency here, especially when considering that NH3 emissions can be important for NO3 aerosol formation.

6. Within section 3 on the evaluation of TM5-FASST numerous references are made to the ability of FASST to predict TM5 concentrations or other metrics using the gradient

of the straight line fit as an estimate of bias. I have noticed a couple of times in the text where FASST is stated to over or under estimate the comparison but the details in the figure do not agree with this statement, which could be due to the use of the gradient. I think that a more appropriate bias statistic such as normalised mean bias (or something similar) could be used to provide an evaluation of FASST rather than this simple linear fit. This occurs throughout Section 3 and please check that all comments are appropriate to the relevant figures.

7. Within Section 3 a comparison has been made with air pollutant concentrations, health and climate metrics. However, no comparison has been made to other studies on the crop relevant metrics. The comparison of crop relevant metrics seems to have been excluded from the comparison. Is it possible to compare the results from FASST to other studies that have looked at the air pollution impact on crops to provide some evaluation of these metrics?

8. Please could the author make sure that all the equations provided within the manuscript are appropriately numbered. It appears that some have been but not all.

Minor Specific Comments

Section 2.1, P3, Line 12 – Brackets needed round O3 as first time defined as ozone.

Section 2.1, P3, line 14 – When describing the particulate matter components I think some mention needs to be made here about Secondary Organic Aerosol (SOA). I think this comes later in the manuscript (section 2.3 P6) but I feel it would also be worth mentioning here with the initial model description.

Section 2.1, P3, Line 26 – 'Although for most health and ecosystem impacts only the surface level fields are required, base simulation and perturbed pollutants concentrations were calculated and stored for the 25 vertical levels of the model as monthly means, and some air quality-relevant parameters as hourly or daily fields.' – I think some mention of the fact that to calculate climate relevant impacts requires 3D information of constituents and not just surface fields.

Section 2.3, P4, Line 25 – reference should be made to the underlying effects of the particular meteorological year used i.e. 2001 in this case.

Section 2.3, P5, Line 4 – '(Where j =i in the case of a primary component)' – maybe this could be changed to '(where the concentration of a primary pollutants is directly related to its emission)'.

Section 2.3, P6, Lines 1 – 7 – There seems to be confusion between the labelling of emitted precursors and concentrations of components as in this section they both seemed to have been referred to as j. Please clarify which letter is meant to represent each.

Section 2.3, P6, Equation 1 – Are these Source receptor coefficients calculated on the monthly or annual response between the precursor emission and pollutant? This needs to be stated within the description of the equation.

Section 2.3, P6, Line 21 - 24 – It is unclear to me how secondary organic aerosol (SOA) is included within the TM5-FASST tool as a component of PM2.5. Does it form part of the POM and what fraction of the primary emissions are used?

Section 2.3, P7, Line 3 – The combination of emissions perturbation scenarios is given in Table 2. Did the base simulation not conduct emission perturbation scenarios for all 56 continental regions? I thought that this would have been essential to enable to the calculation of changes in concentrations in TM5 but Table 2 does not seem to imply this. Clarification required.

Section 2.3, P7, Line 15 to 18 – The change in CH4 burden in TM5 from the HTAP1 perturbation simulations is stated as being an emission perturbation of 77 Tg/year. Could the authors provide information on how this was obtained.

Section 2.3, P7, Lines 22 to 28 – It is stated that FASST does not include impacts on O3 from perturbations in CO emissions. I am not sure why this has not been included

in the development of FASST along with other O3 precursor emissions of NOx and NMVOCs. Within this section it states that there is a dedicated CO emission perturbation experiment conducted with TM5 as part of HTAP1 available and that the impacts on O3 are not insignificant. Therefore I wonder why the information from the TM5 CO experiments have not been included previously within FASST?

Section 2.4, P8 – Maybe this section should be labelled as something like 'Urban Adjustments in PM2.5 for Health Calculation' to better identify what is being done here. I am assuming that the adjusted PM2.5 concentrations are only used within the calculation of health impacts?

Section 2.4, P8, Lines 25 -26 – Is the CIESIN population dataset the default one used within FASST as this seems to have been used to calculate the default urban increment factors in Table S4.2? Might be worth included which one is recommended for use.

Section 2.5, P9, Line 10 – Check definition of AF here as this does not match up with what is provided further down the page, just above line 20.

Section 2.5, P10, Lines 17 to 24 – I think a comment is required here to state how the recent updates in the epidemiological evidence for health effects could impact on the predictions in FASST i.e. will they be cause an underestimate or overestimate.

Section 2.6, P11, Line 14 – 'Both Mi metrics . . .' should be changed to 'Both metrics (Mi) . . .'

Section 2.6, P11, Line 15 – How is the growing season defined when calculating the crop metrics?

Section 2.6, P11, Line 16 – RYL is defined as the crop relative yield. Should this be the relative yield loss? Also the coefficients a,b,c within the equation for RYL need more explanation.

Section 2.7.1, P12, Lines 10 to 12 – Are these two sentences on the basic radiative properties of aerosols relevant? Including some text on the following lines would

be good to discuss how the treatment of externally mixed aerosols alters the radiative forcing calculations when compared to internally mixed ones (Lesins et al., 2002; Klingmüller et al., 2014).

Section 2.7.2, P12 – I think this sections needs to be made clearer. I am struggling to make the link between the output from FASST and the calculation of indirect aerosol forcing. How is done? What fields from FASST are used to calculate it? Needs to explain the methodology better for the reader.

Section 2.7.2, P12, Line 29 – Add year used to meteorological data

Section 2.7.2, P12, Line 30 – missing word 'using' between after 'calculated'. Also it is probably worth stating here or in the supplementary section S6 the equations used to calculate cloud droplet number concentrations and cloud effective radius.

Section 2.7.3, P13 – Like section 2.7.2. I think this section needs to be made clearer to highlight what output is being used from FASST to compute O3 and CH4 radiative forcings. There is a lot of details of what is included but I struggled to follow the basic principle of FASST output + forcing efficiency = radiative forcing. I think the description of what is done in FASST should come first at the start of this paragraph and then follow with the description of what it takes account of.

Section 2.7.3, P13, Lines 4 to 6 – How do these STOCHEM calculations compare to the ACCMIP multi-model mean and is it still appropriate?

Section 2.7.3, P13, Line 32 – For regions not covered by the major HTAP1 source could the 'rest of the world' CO forcing efficiency not be used from Table S6.3 rather than a global average?

Section 2.7.4, P14, Line 7 – Are the emission based forcing efficiencies those in Table S6.2 to S6.5? Can a reference be put in to these in the main text?

Section 3, P15, Lines 19 to 21 – Simplify point 1 to read better.

Section 3.1, P16, Line 2 – reference is made to Annex 4 of the SI. Pleases clarify this reference as there is no Annex 4.

Section 3.1.1, P16, Lines 12 to 14 – Is there a reason for the particular representative source regions selected in Table 2 e.g. South Africa for NOx.

Section 3.1.1, P16, Lines 19 to 22 – The explanation on these lines could be simplified.

Section 3.1.1, P16, Lines 29 to 31 – Also there is a larger response to NO3 from increasing NOx emissions over India. Do you think that is this a particular issue for TM5 over India? Does this is cause issues for future prediction of NO3 aerosol from changes in NOx emissions over India?

Section 3.1.1, P17, Lines 8 to 11 – I don't think you can say that errors in the -80% case are larger than +100% for NOx. They look similar to me.

Section 3.1.2, P17, Lines 18 to 19 – Can you include references to back up the fact that combined NOx and NMVOCs emission perturbations will behave more linearly?

Section 3.1.2, P17, Line 31 – remove 'also here'

Section 3.1.2, P17, Line 31 to 32 – Good agreement is found everywhere apart from China, Why?

Section 3.1.2, P18, Line 2 – change 'Europa' to Europe

Section 3.1.2, P18, Lines 16 to 18 – If anything I would say FASST overestimates the change in TM5 (be it positive or negative) most of the time as the -80% points on the scatter plot tend to always above the 1:1 line (see major point 6 above).

Section 3.1.2, P18, Lines 21 to 22 – I am not sure that the linear fit is that good for the change in annual mean O3 in Figure 7a as there seems to be distinctive curvature in the +100% simulation for larger O3 reductions. I anticipate that this will be larger for certain months. The non-linear behaviour seems to occur to a lesser extent for other O3 metrics where a linear approximation is probably more justified. I think a change

of wording for this statement is required to reflect the fact that a linear approximation does not represent the non-linear chemistry effects for large emission perturbation.

Section 3.1.2, P18, Lines 25 to 28 – Check percentage numbers are correct as they don't appear to be the same as that shown on Figure S7.4 or in Table 3 e.g. -5 to 13% for M12 where on the Figure S7.4c I can't see anything below 0.

Section 3.1.2, P18, Lines 28 to 30 – Same as above but for NMVOC.

Section 3.1.2, P19, Line 1 – Same as above but for combined emission perturbation.

Section 3.2, P19, Line 10 – remove 'e'

Section 3.2, P19, Line 25 to 26 –In both scenarios emissions can change by >80% over some regions and precursors. The ability of FASST to predict such changes over regions needs to be highlighted in the results based on the breakdown of the linear approach for O3 at such high emission perturbations.

Section 3.2, P20, Lines 3 to 5 – If it is more policy relevant to consider the change in pollutant concentrations between two scenarios than absolute concentrations, and FASST is a tool for the assessment of policy measures, then why is the difference not shown in place of absolute concentrations? Might be worth showing the change in concentrations in the main text and the absolute concentrations in the supplementary. Also it might be better to show the change between FLE and BASE, and MIT and BASE separately rather than the different between the two future scenarios.

Section 3.2, P20, Line11 to 12 – I would say that FASST tends to underestimate the magnitude of change in TM5 for both annual mean and M6M O3, as most points are below the 1:1 line. (see major point 6 above).

Section 3.2, P20, Lines 15 to 21 – Only a very small discussion on the future evaluation of health metrics. Maybe expand slightly to include different regions and that FASST always overpredicts compared to TM5.

Section 3.3.2, P22, Line 1 – relate discussion of text to labels on Figure 13 or define labels with more description in Figure 13 caption.

Section 3.3.3, P22, Line 13 – replace 'were' with 'where' and remove 'and'.

Section 3.3.4, P22, Line 22 – remove 'implemented in FASST'

Section 3.3.4, P22, Line 24 – replace 'death cause' with 'a cause of death'.

Section 3.3.4, P22, Line 25 to 27 – Could the difference in population and mortality rates between the two studies lead to some of the differences in Figure 14?

Section 3.3.4, P23, Line 9 to 11 – How can FASST account for inter-model variability in its results? I think that this is mentioned as a future development so needs to be linked to that here.

Section 3.3.4, P23, Line 17 – replace 'While also' with 'Whilst calculated'.

Section 3.3.4, P23, Line 18 to 21 – Why does the different baseline mortality and population statistics have such a big impact on O3 mortality rates but not PM2.5?

Section 3.3.4, P23, Line 27 to 31 – Could a little more discussion on regional mortality burdens be put into the main text. Interesting differences between regions and for RCP2.6 vs RCP8.5.

Section 4, P24, Line 17 to 18 – Make statement in this sentence less strong by inserting 'tend to' between 'metrics' and 'remain'.

Section 4, P24, Lines 21 to 23 – I think the first two sentences could be re-written to simply specify that because the emissions and meteorology are fixed the source receptor matrices remain fixed. Also I think the work 'arbitrary' should be removed.

Section 4, P24, Lines 25 – remove repetitive statement of 'compared to the base simulation year 2000'.

Section 4, P24, Lines 27 – remove 'be'

Section 4, P24, Lines 30 to 31 – reword sentence to 'It can be expected that errors will be larger for the newer generation scenarios with dynamic allocation of emission across countries and macro-regions'.

Section 4, P25, Lines 5 to 7 – Sectors are mentioned that can't be assessed but little has been mentioned about shipping and aviation which can be assessed and are specifically included as a source region in FASST. I think it is worth mentioning these source regions in this section.

Section 5, P25, Line 32 – removal of '. . ..' at end of page.

Section 5, P26, Line 6 – subscripts for O3 and PM2.5 required.

Section 5, P26, Line 19 – Slightly more detail could be provided on how the HTAP2 modelling exercise will inform/improve TM5-FASST, especially as TM5 was not a model that participated in HTAP2. Figure 14 – I find that the grey lines mask out the black lines in some instances and I think the Figure would look better if the grey lines could be made less bold or more transparent. Also I am not sure why there is a different number of grey lines on each part of the Figure. Did a different number of models submit results for each experiment?

Supplementary Material

Table S3 – Certain lines in the table seem to be missing any information. e.g. P5 Germany, P4 USA, P5 Japan.

Figure S3.3 – Why has the sign been reversed? For a 20% reduction in CH4 you would expect a decrease in O3 concentrations but the figure shows positive changes. This seems confusing.

Section S4.1, Equation 4.4 – I am not sure I can follow how the INCR formulation was derived and whyt it includes the (fup)2 terms.

Figure S5 – hard to decipher the different lines on the graph. Cannot see red lines

most of the time. Please make clearer.

Section S6.1, P24, Line 166 – 'Table S7.1' should be Table S6.1.

References

Burnett, R. T., Arden Pope, C., Ezzati, M., Olives, C., Lim, S. S., Mehta, S., Shin, H. H., Singh, G., Hubbell, B., Brauer, M., Ross Anderson, H., Smith, K. R., Balmes, J. R., Bruce, N. G., Kan, H., Laden, F., Prüss-Ustün, A., Turner, M. C., Gapstur, S. M., Diver, W. R. and Cohen, A.: An integrated risk function for estimating the global burden of disease attributable to ambient fine particulate matter exposure, Environ. Health Perspect., 122, 397–403, doi:10.1289/ehp.1307049, 2014.

Klingmüller, K., Steil, B., Brühl, C., Tost, H. and Lelieveld, J.: Sensitivity of aerosol radiative effects to different mixing assumptions in the AEROPT 1.0 submodel of the EMAC atmospheric-chemistry–climate model, Geosci. Model Dev., 7(5), 2503–2516, doi:10.5194/gmd-7-2503-2014, 2014.

Lesins, G., Chylek, P. and Lohmann, U.: A study of internal and external mixing scenarios and its effect on aerosol optical properties and direct radiative forcing, J. Geophys. Res. Atmos., 107(D10), AAC 5-1-AAC 5-12, doi:10.1029/2001JD000973, 2002.

---

## Referee Comment (RC2) · Anonymous Referee #2 · 17 Apr 2018

The manuscript by Van Dingenen et al. presents and evaluates TM5-FASST, a reduced form air quality assessment tool. The manuscript is long, but does a thorough job of both presenting the computational methods used for formulating the FASST tool and the types of impacts that it calculates (from health impacts to climate) as well as evaluating the tool against simulations from the full TM5 model as well as results in the literature. I have some additional questions in a few areas, described below, but in general was satisfied / impressed with the evaluation and performance. The writing could use a bit more editing for grammar and some of the figures need clarification

on units, axis, etc. Addressing these will amount to moderate revisions and some additional evaluations.

However, my only main concern about would be if this article should be moved to GMD instead of ACP, as the emphasis really is on the tool development and evaluation; there is not any content on application of the tool to new science or policy questions. It may not thus fit the scope of ACP.

Comments:

2.1-8: Why are reduced form or source-receptor models needed in the first place? I think there's a significant point to be made here about the complexity of air quality modeling vs the level of sophistication and computational intensity that can be acceptable to the decision-making community. But the article as presently writing misses this point, so the justification for the tool isn't readily apparent.

Intro: Overall the introduction is rather brief. There are other reduced form models on regional scales that are used for different purposes (in the US and Asia, in particular). There are also theoretical advantages (quick) and disadvantages (approximations of linearity; enforced aggregation at pre-defined scales; outdated emissions inventories or old atmospheric conditions) of reduced form models. A lot more thought could be put into discussion and introducing these issues. This is ACP, not GMD, so more than just a model description is expected.

3.18: This sentence is a bit too vague to be useful. The authors should mention what type of model updates have been made (emissions? aerosols? etc) and why they are deemed to not be relevant for this current work.

This does raise the question of uncertainties introduced in this tool owing to use of a single year (was 2001 an average year, in terms of temp, precipitation, etc.?) to approximate a reasonable climatology, as well as this use of a year that is significantly older than most present applications, considering decadal-scale climate change.

4.29: There is extensive research on the chemical oxidation of elemental carbon and the role this plays on the lifetime of this species in the atmosphere. Comment on why this is not included.

5.1: I'm not sure what is a "parameter" in this context – please explain.

5.1 - 14: It seems like some discussion of the fact that this functional relationship is only approximate is warranted. Instead, it is presented here as if the actual functional relationship is known, where in fact just a local linear approximation is used. This must have some limitations. For example, what is the basis for the statement later on this page that -20% perturbation is small enough to evaluate sensitivities and large enough for extrapolation? I recognize that -20% is a commonly used modeling experiment, but it is also commonly known that this approach has limitations for source attribution that are well documented in the literature (compared to tagging, 2nd order methods, or other).

6: The notation in equations (1) and (2) is not correct. In equation 1, there is an inconsistency between the description of the notation for the concentrations vs emissions species ($i$ and $j$) and what is written in the equation. Assuming the equation is correct, the text should refer to change in concentration of component $j$ (not $i$) owing to emitted precursor $i$ (not $j$).

In equation (2), the notation on the summations is not complete nor correct. The first sum should be from $x = 1$ (below the sum) to $n_x$ (written above the sum), and the second should be for $i = 1$ (below the sum) to $n_i(j)$ (above the sum). It's also not clear why $y$ would be bold in this equation. As explained in the text, the number of precursor pollutants ($n_i$) depends on the pollutant response in consideration, hence $n_i$ is $n_i(j)$.

So the pollutant responses are dry aerosol concentrations? At what T,P,RH?

6.21: This equation needs to be corrected following suggestions for equation (2).

6.23: It is oxymoronic to refer to secondary biogenic POM. This would just be secondary biogenic OM.

6.24: Can the authors comment on how neglect of anthropogenic SOA might be biasing the results of this tool?

6.25: Just because the impacts are annual in nature doesn't mean the emissions contributions to the impacts are seasonally consistent. Surely the impact of NOx on ammonium nitrate and O3 is quite different in different seasons; it's not clear why one would have access to this information but not use it.

7.7-21: I got a bit lost in this discussion of the way CH4 concentrations responses are treated. It would be good if this section could be expanded and formalized a bit better, using equations where useful, such that the approach could be evaluated and replicated.

It also wasn't clear to me – is NOx allowed to impact CH4, particularly for the purposes of climate imapacts?

Many of the studies in the table are a bit out-of-date, as they would be around atmospheric conditions / emissions levels that are rather old, or in comparison to datasets that have greatly matured (for example comparison to satellite-based NO2 retrievals, which are now much more accurate and consistent across retrievals than in the study of van Noije 2006.

8.3-4: Statements like this could be supported by reference many articles on the topic, including evaluation of how much this matters for different species at different scales.

8.13: Is there a reason why primary PM2.5 from industrial sources would also not be expected to contribute to the local urban increment? Or is this source just not very large?

Section 2.4 and SIS4 are useful in understanding the urban increment, and some evaluation of improvement in performance compared to satellite-derived PM2.5 is included. However, the evidence is a bit indirect. I'd like to see a comparison of native 1x1 and

urban downscaled BC concentration to in situ measurements from urban monitoring sites, such as are available in the US.

11.25-27: This justification would be improved if the authors were a bit more quantitative. Also, if the lowest model level O3 compares favorably to the surface O3 measurements, this begs the question then of why the modeled O3 thought the lower atmosphere in TM5 is biased low (as surface-level concentrations would be lower than 30 m concentrations).

12.21: The authors evaluate their approach to calculating direct aerosol radiative forcing by providing plots of the species specific forcings in Fig S6.1 and noting they are "reliable results." However, I don't have any sense of what makes these results reliable. What features of the distributions shown in these plots are those that we would expect, easily explain, or could compare to observations or other modeling studies? The BC RF in the eastern part of Antarctica exhibits a strange horizontal strip that I'm not sure about. Also, the figure legend on S6.1 is redundant and the units on both of the color bars are incorrect.

Section 2.7.2: The tool does not include the substantial non-direct cloud interactions for BC, nor the impact of BC on snow / ice albedo. These factors contribute significantly to the targeting of BC-rich sources for SLCP mitigation. Comment on how omission would affect TM5-FASST results.

Fig 2: What is the mechanism by which the perturbation in NOx emissions causes a reduction in SO4 in IND (as opposed to an increase in all other regions)?

18.1-2: It seems NOx levels in the US and Europe are much lower now, and I'm not sure these titrations still exist; they are at least less persistent in the summer. See for example recent article by Jin, Fiore, et al., JGR, 2017.

18.6: The statement that the sensitivities of impact-relevant O3 metrics (M6M and M12) are more linear than for annual average pop-weighted O3 is not supported by

the results shown in Fig 6. The responses to NOx emission changes seem to be more nonlinear for M6M or M12 in some cases, such as GBR as well as others. The text should be revised accordingly.

18.20: AOT40 would focus on high O3 values. It's not clear to my, chemically speaking, why this would be expected to response more nonlinearly than other metrics. Presumably larger O3 values are occurring more in the summer. Earlier it was claimed that summer sensitivities would be more linear... so I'm a bit confused here.

18.31: Why is that the case, chemically speaking?

20.4-14: I find it interesting that the change in PM2.5 is predicted by FASST better than absolute concentrations (which I would expect) but that the change in O3 metrics is predicted more poorly than absolute concentrations (would not expect). Do the authors have any thoughts about the reasons behind the latter?

20.15-22: That's a reasonable comparison. I also wonder though what is the total number of estimated premature deaths associated with PM2.5 and O3, and how these numbers compare to those in the literature (from e.g. GBD), for present day conditions. This would help evaluate the accuracy of the absolute estimates in addition to estimates of changes.

20.25 - 21.7: I'm I incorrect in thinking that many of the pre-industrial to present IPCC RF's also include an 80% reduction in biomass burning sources? If so, this might further explain why the IPCC values are on the higher side. Also, IPCC estimates and those in Bond include RF of BC on snow, which I don't see as being accounted for in FASST.

21.22: How doe they know it's owing to different OH levels and lifetimes rather than to different emissions (line 21.14)?

Section 3.3.2: The evaluation of global sector and species specific RF looks good. A key feature of FASST is regional specificity; could they also compare to some studies

in the literature that have evaluation the RF of regionally specific emissions by species or sector?

23.25: Does including this correction for changing mortality rates though lead to worse agreement between ACCMIP and FAST for PM2.5 related deaths (Fig 15)?

Section 4: Good discussion. Some caveats about missing accurate treatment of SOA? Or carbonaceous aerosol aging? And possibly being a bit more clear about the limits of the emissions perturbations magnitudes that should be used with this tool (e.g., x2? x5? x10?).

Technical / editorial comments:

1.15: as broad range of pollutant-related impacts, related to—> as a broad range of pollutant-related impacts on

1:21 are not compromising –> do not compromise

1.22: I'm not sure that evaluation of the model proves that it is useful for science-policy analysis (that type of proof would require demonstration of actual use for such purposes), so this sentence should be rephrased to be a bit more accurate.

1.24: I'm not really sure that this sentence means. Suggest rewriting or omitting? If keeping, change frame to framework?

2.7: Seems there is a closing parenthesis missing after the Amann citation.

3.2: from literature –> from the literature

3.3: discussing –> discussion of

3.4: like. . .the HTAP1 –> similar to how the development of TM5-FASST was built upon extending the HTAP1

3.4-8: I don't think readers unfamiliar with HTAP would understand this sentence. Also, I don't see how this manuscript could influence the HTAP2 exercise, in terms of timing,

as the latter is essentially complete.

3.13: Something is strange with the size of the subscripts in SO2 and PM2.5

3.25: pollutants –> pollutant

3.28: It was already stated that the meteorology is from ECMWF, so this is redundant. Maybe add the part about the year 2001 to the first sentence of this paragraph.

4.15: together an –> together with an

4.22: of anthropogenic –> of anthropogenic emissions

7.2: Formatting error in SOMO35.

8.11: Further,

SI.97: the. The

SI.S5.1 caption says "MEAS" is "routine monitoring programs" whereas the text on line 95 says the high-resolution dataset is the satellite-derived product from van Donkelaar et al. (2010). Please update the figure caption to be more precise.

14.17: where

16.8: perturbed with –> perturbed by

16.17: Germany as –> Germany as an

16.30: response linearity towards NOx emissions —> linearity of the response to NOx emissions

19.10: e selected

20.23: confront –> evaluate

Fig 13: x-axis missing labels.

23.18: extend –> extent

25.30: Strange to end with "..."

---

## Author Comment (AC1) · 6 Sep 2018

We thank reviewer 1 for the insightful comments, and for pointing to inconsistencies. We apologize for needing more time than anticipated to address all comments, but we believe that we have been able to address most issues, and that we have significantly strengthened the manuscript.

Before addressing the comments we would like to mention that we have modified the abbreviation of the $O_3$ health exposure metric (6 monthly daily maximum 1-h concentration) from M6M to 6mDMA1 (and accordingly M3M to 3mDMA1) as the latter seems to be commonly used in other works.

In the following we have placed the numbered reviewer comments in boxes. Our reply to the reviewer is in blue font, the changes to the manuscript in red font.

We also attach a revised version of manuscript and supplement with tracked changes compared to the first version.

REVIEWER 1 comments:

The manuscript presents a detailed summary of the methodology and validation for the TM5-FASST screening tool. TM5-FASST is a simplified tool that uses linear source receptor relationships of air pollutant precursor species across 56 geographical source regions (plus aviation and shipping) to calculate the response in air pollutant concentrations at both the surface and 25 vertical layers in the atmosphere. The difference in concentrations can then be used to calculate the change in a number of air pollution impact metrics related to human health, climate and crop production. The tool allows for the impact from different emissions scenarios to be explored without the need to run more detailed composition climate models. The manuscript provides a through description of the underlying methodology of TM5-FASST as well as an evaluation of the air pollutant predictions and impact metrics against a number of different sources. It provides a good reference for the TM5-FASST tool for use in future studies.

Major General Comments

1) Whilst I understand that TM5-FASST is not meant to replicate full scale model simulations, it would be good to bring together the limitations together into a more coherent section, possibly within the discussion section. Throughout the manuscript specific sections of the text mention aspects that TM5-FASST will not be able to predict e.g. changing spatial distribution of emissions and chemical regime. It would make sense for the reader to have these all in one place.

REPLY: Thank you for the suggestion. We have substantially edited and extended the discussion section to address the limitations of the tool..

CHANGES TO MANUSCRIPT: *New section 4*:

[revised manuscript text omitted]

2) Also I found little mention of how the fixed meteorological year of 2001 could potentially impact the prediction of pollutants in the future i.e. how would climate change affect predictions of future pollutants?
Also the basis for the radiative forcing calculations is from a fixed meteorological year of 2001 and could have implications for the future calculation of effects. A more detailed mention of these issues would be good, perhaps in Section 4.

REPLY: This is an issue raised by both reviewers. We agree with the reviewer that the year 2001 meteorology is somewhat outdated. The perturbation runs for constructing the SR library of FASST were performed with the TM5 model set-up defined in the first phase of HTAP1 (during the period 2008 – 2011) and because of the computational costs, an update with more recent meteorology was not possible (TM5 is not taking part in HTAP2 where meteorological year 2010 has been used). A systematic check of the representativeness of this particular year for each of the FASST regions is beyond the scope of this study, in the first place because FASST is considered to be a screening tool focussing on impacts of emission changes. However we have substantially extended the discussion on the use of a single meteorological year.

CHANGES TO MANUSCRIPT:   *Added to Section 2.1 P5 L10*

Meteorological fields are obtained from the ECMWF operational forecast representative for the year 2001. The implications of using a single meteorological year will be discussed in section 4.2.

*Discussion section 4.2  added  as included above.*

3) TM5-FASST and the validation of it using TM5 simulations have all been conducted using emissions inventory for the year 2000 as a baseline along with 20% perturbations from this base. How appropriate is it to use a base year of 2000 for validation purposes given the large recent changes in emissions over the last 10-15 years, particularly over East Asia where some emissions have changed by >20%. What impact would using more up to date emissions in the base scenario have the calculated source-receptor coefficients and would it significantly affect the magnitude of future predictions? It would be useful to provide information on how recent changes in emissions could impact TM5-FASST.

REPLY: This is certainly an issue of concern, but at the same time difficult to address in a quantitative way. Although an independent set of SR simulations departing from a different reference scenario is not available, in the manuscript we included a validation of the linear scaling approach beyond the -20% perturbation, based on a number of additional perturbation simulations with TM5 for selected key regions, including East-Asia. In these test cases, the emissions of individual precursors where decreased by -80% relative to the reference emissions of the year 2000, while other precursors were kept at the year 2000 emissions. These simulations are not exactly testing the emission-response sensitivity for a different reference case, but they do provide a validation of the linear approach.
A second validation method, discussed as well in the paper, uses exactly emission scenarios that are strongly different from the reference year 2000 case for all precursors simultaneously (i.e. GEA FLE2030 and MIT2030 scenarios), where FASST uses the sensitivities based on year 2000 and compares the outcome  with TM5, to some extend addressing the issue raised by the reviewer. The magnitudes of these emission changes are representative for more recent scenarios. A general observation is that FASST somewhat over-predicts resulting O3 and $PM_{2.5}$ concentrations (compared to the full TM5 model) for large (i.e. greater than say 50%) emission perturbations in either direction, but this does not compromise its usefulness as a tool to explore air pollution scenarios in a multi-pollutant/multi-impact  framework. Indeed, as demonstrated in section 3.3, regional key features and trends, as well as inter-regional differences or similarities resulting from future RCP scenarios up to 2050 are reproduced within the variability of the ACCMIP air quality model ensemble.
We dedicate more discussion on these results, including a more systematic statistical analysis of the performance of FASST versus TM5.  In the final discussion we refer to the new round of perturbation simulations performed in the frame of HTAP2.

[revised manuscript text omitted]

4) In Section 2.1, P4, Line 12 the manuscript mentions about the advent of finer resolution global models nearing 1◦x1◦ horizontal resolution. I think it would be good to make more comment on the applicability of the 1◦x1◦ resolution when calculating country scale impacts. Is this resolution along with input information at similar resolution (e.g. emissions) sufficient to capture changes in pollutants at sub 100 km scales over countries such as the UK and Belgium/Luxembourg. I think that the urban adjustment of $PM_{2.5}$ is a suitable attempt at this but I think it would good to have some further comment on the issue of resolution and the limitations provided by other inputs at this resolution e.g. emissions and meteorology

REPLY: We agree with this critique, even if TM5 during the last decade or so has been amongst the global models with highest grid resolution that have made global studies on health impacts of air pollution. Further, the FASST source regions are defined such to include several gridboxes, e.g. Belgium/The Netherlands/Luxembourg are aggregated into a single region. We address the comment in the following ways:

1) Section 2.4, initially dedicated to the sub-grid adjustment for urban concentrations, has now been extended to include a quick analysis of the TM5 base simulation at resolution 6°x4°, 3°x2° and 1°x1° to illustrate the impact of resolution on concentrations and emission-concentration response sensitivities, with more detailed information and figures provided in the SI.

2) The paper already included a methodology to partly address the sub-grid gradients with a parametrized approach; in the connected annex S4 in the SI we now explicitly compare FASST $PM_{2.5}$ with a high-resolution satellite product.

CHANGES TO MANUSCRIPT:
*Expanded section 2.1 P5 L26*

With the introduction of massive parallel computing, however, this comparative advantage is now slowly disappearing, and global model resolutions of 1°x1° or finer are now becoming more common (see the model descriptions in this special issue, e.g. Liang et al., 2018). The model grid resolution influences the predicted pollutant concentrations as well as the estimated population exposure, especially near urban areas where strong gradients occur in population density and pollutant levels, which cannot be resolved by the 1°x1° resolution. In section 2.4 we describe a methodology to improve population $PM_{2.5}$ exposure estimates by applying sub-grid concentration adjustments based on high-resolution ancillary data. The bias introduced by model resolution affects as well computed SR matrices, e.g. off-setting the share of 'local' versus 'imported' pollution in a given receptor region. We will discuss this aspect more in detail in section 4.3.

*(Section 4.3: see reply to comment 1)*

5) Section 2.5 on health impacts provides a lot of details and is quite long compared to some others sections where most of the details are within a supplementary section. Also I found it a bit confusing to have two options for calculating $PM_{2.5}$ health effects: the log-linear and integrated exposure-response functions (IER). I assume the output from FASST is only provided from one (Figure 15)? The paragraph on page 10 Lines 8 to 13 does not seem to provide clarity on which method is preferred and could be re-worded. Therefore Section 2.5 could be potentially made more concise by removing the details on the log-linear method to the supplementary. This would allow the main text to focus more on the IER method by Burnett et al., (2014), which is the current methodology used within the Global Burden of Disease study.

REPLY: The reason for the relatively detailed description of health impacts calculations in TM5-FASST is that most users and publications tend to focus on this aspects- and because differences in methodologies are an important reason for differences in calculated health outcomes. We agree however that most of the description could move to the SI.

Most recently published global studies on health impacts of ambient air pollution use one of the two methodologies for $PM_{2.5}$ (i.e. log-lin and/or GBD) and both methodologies appear in WHO recommendations for Europe. We included both methods in the FASST output to facilitate comparison with other studies. The two calculations also provide an additional perspective on the uncertainty of the health impact outcome. (Upon request of Ref #2 we included an additional intercomparison of present-day mortalities with other studies, using both methodologies). However we agree with the reviewer that it was not clearly stated which method was used in Fig. 15 (now Fig. 17) – in this case, as the Silva study was based on GBD, we also used the result following the GBD methodology.

CHANGES TO MANUSCRIPT:
*As suggested by the reviewer, we have moved a large part of the description of the health methodology (section 2.6) to the SI and kept only the GBD methodology in the main text, while mentioning that the tool includes the log-lin method as well.*
*We also mention now specifically in section 3.3.5 (Health impacts) that the methodology is based on GBD.*

**2.5 Health impacts**

TM5-FASST provides output of annual mean $PM_{2.5}$ and $O_3$ health metrics (3-monthly and 6-monthly mean of daily maximum hourly $O_3$ (3mDMA1, 6mDMA1), and the sum of the maximal 8-hourly mean above a threshold of 35 ppbV (SOMO35) or without threshold (SOMO0), as well as annual mean $NO_x$ and $SO_2$ concentrations at grid resolution of 1°x1°. These are the metrics consistent with underlying epidemiological studies (Jerrett et al., 2009; Krewski et al., 2009; Pope et al., 2002). The population-weighted pollutant exposure metrics grid maps, in combination with any consistent population grid map, are thus available for human health impact assessment. The TM5-FASST_v0 tool provides a set of standard methodologies, including default population and health statistics, to quantify the number of air quality-related premature deaths from $PM_{2.5}$ and $O_3$.

Health impacts from $PM_{2.5}$ are calculated as the number of annual premature mortalities from 5 causes of death, following the Global Burden of Disease methodology (Lim et al., 2012): ischemic heart disease (IHD), chronic obstructive pulmonary disease (COPD), stroke, lung cancer (LC) and acute lower respiratory airways infections (ALRI) whereas mortalities from exposure to $O_3$ are related to respiratory disease.

Cause-specific excess mortalities are calculated at grid cell level using a population-attributable fraction approach as described in Murray et al. (2003) from $\Delta Mort = m_0 \times AF \times Pop$, where $m_0$ is the baseline mortality rate for the exposed population, $AF = (RR-1)/RR$ is the fraction of total mortalities attributed to the risk factor (exposure to air pollution), $RR$ = relative risk of death attributable to a change in population-weighted mean pollutant concentration, and $Pop$ is the exposed population (adults ≥ 30 years old, except for ALRI for which infant population <5 years old was considered). $RR$ for $PM_{2.5}$ exposure is calculated from the Integrated Exposure-Response functions (IER) developed by Burnett et al. (2014), and first applied in e.g. the Global Burden of Disease study (Lim et al., 2012).

In order to facilitate comparison with earlier studies, TM5-FASST provides as well mortality estimates based on a log-linear exposure response function $RR = \exp^{\beta \Delta PM_{2.5}}$ where $\beta$ is the concentration–response factor (CRF; i.e., the estimated slope of the log-linear relation between concentration and mortality) and $\Delta PM_{2.5}$ is the change in concentration. More details on the health impact methodologies, as well as sources for currently implemented population and baseline mortality statistics and their projections in TM5-FASST_v0 are given in section S5 of the SI.

For $O_3$ exposure, $RR = e^{\beta(\Delta 6mDMA1)}$ , $\beta$ is the concentration–response factor, and $RR$ = 1.040 [95% confidence interval (CI): 1.013, 1.067] for a 10 ppb increase in 6mDMA1 according to Jerrett et al. (2009). We apply a default counterfactual concentration of 33.3 ppbV, the minimum 6mDMA1 exposure level in the Jerrett et al. (2009) epidemiological study.

We note that the coefficients in the IER functions used in the GBD assessments have been recently updated due to methodological improvements in the curve fitting, leading to generally higher RR and mortality estimates (Cohen et al.,2017; Forouzanfar et al., 2016). In particular, the theoretical minimum risk exposure level was assigned a uniform distribution of 2.4–5.9 $\mu g/m^3$ for $PM_{2.5}$, bounded by the minimum and fifth percentiles of exposure distributions from outdoor air pollution cohort studies, compared to the presently used range of  5.8 - 8.8  $\mu g\ m^{-3}$ which would increase the health impact from $PM_{2.5}$ in relatively clean areas.  Further, a recent health impact assessment (Malley et al., 2017), using updated RR estimate and exposure parameters from the epidemiological study by Turner et al. (2016), estimates 1.04–1.23 million respiratory deaths in adults attributable to $O_3$ exposure, compared with 0.40–0.55 million respiratory deaths attributable to $O_3$ exposure based on the earlier (Jerrett et al., 2009) risk estimate and parameters. These recent updates have not been included in the current version of TM5-FASST. Health impacts from exposure to other pollutants ($NO_2$, $SO_2$ for example) are currently not being evaluated in TM5-FASST-v0

*In section 3.3.5  P27 L19*

The analysis by Silva et al. (2016) used the same methodology implemented in FASST for estimating premature mortalities from $PM_{2.5}$ and $O_3$ (i.e. Burnett et al., 2014 as in the Global Burden of Disease study and Jerrett et al., 2009 respectively)
* * *
6) In section 3.1.1 when making a comparison of the additivity of emission perturbations for $PM_{2.5}$ individual changes for SO2, NOx and NH3 is shown on Figure 3 and 4 but in Figure 2 there is no effect from NH3 emissions. Whereas, in Figure S7.1 and S7.2 the 3 individual responses are shown along with the combined response on $PM_{2.5}$ (sum of all 3). However, the effect for combined emissions is only for SO2 and NOx in Figure 2 and 4 and does not include any addition from NH3. Why has the contribution from NH3 not been included within some of the combined emission changes in PM2.5? There seems to be a bit of inconsistency here, especially when considering that NH3 emissions can be important for NO3 aerosol formation.
* * *
REPLY: In first instance we have evaluated separately the 'additivity' and 'linearity' issues. Figure 2 demonstrates the additivity assumption of NOx and SO2 perturbations. This requires model simulations for (1) SO2 only perturbation (2) NOx only perturbation and (3) simultaneous NOx+SO2 perturbation, all of the same magnitude. For each of these perturbation experiments, the effect on SO4, NO3 and NH4 in $PM_{2.5}$ is available.

However due to lack of CPU resources, similar analyses for combined SO2+NH3 and NOx+NH3 perturbations have not been performed unfortunately.  Because only separate NH3 perturbations are available we cannot provide the equivalent figures for these combinations. We therefore assume additivity for the combined perturbations of NH3 with SO2 and NOx respectively.  To some extend one may argue that source regions of NH3 on the one hand, and SO2 and NOx on the other are less aligned, and that control strategies are different/independent hence simultaneous reductions are less pertinent, but we recognize this is a caveat in the FASST methodology.

CHANGES TO MANUSCRIPT:

**3.1 Validation against the full TM5 model: additivity and linearity**

We recall that the TM5-FASST computes concentrations and metrics based on a perturbation approach, i.e. the linearization applies only on the difference between scenario and reference emission. Therefore we focus on evaluating the perturbation response, i.e. the second term in the right hand side of Eq. 2.

The standard set of -20% emission perturbation simulations, available for all 56 continental source regions and constituting the kernel of TM5-FASST_v0 are simulations P1 (perturbation of $SO_2$, $NO_x$, BC and POM), P2 ($SO_2$ only), and P4 ($NH_3$ and NMVOC) shown in Table 2. Additional standard -20% perturbation experiments P3 ($NO_x$ only) and P5 ($NO_x$ and NMVOC), as well as an additional set of perturbation simulations P1' to P5' over the range [-80%, +100%], listed in Table S3 of the SI, have been performed for a limited selection of representative source regions (Europe, USA, China, India, Japan) due to limited CPU resources. For the same reason, no combined perturbation studies are available for ($SO_2$ + $NH_3$) and ($NO_x$ + $NH_3$) for a systematic evaluation of additivity and linearity. The available [-80%, +100%] perturbations are used to validate the linearized reduced-form approach against the full TM5 model, exploring chemical feedback mechanisms (additivity) and extrapolation of the -20% response sensitivity towards larger emission perturbation magnitudes (linearity). This is in particular relevant for the $NO_x$ - NMVOC - $O_3$ chemistry and for the secondary $PM_{2.5}$ components $NO_3^-$ - $SO_4^{2-}$ - $NH_4^+$. These mechanisms could also be important for organic aerosol, but we remind that in this study organic aerosol formation was parameterized as pseudo-emissions.

**3.1.1 Additivity and linearity of secondary inorganic $PM_{2.5}$ response:**

Experiment P1, where BC, POM, $SO_2$ and $NO_x$ emissions are simultaneously perturbed by -20% relative to base simulation P0, delivers SR matrices for primary components BC and POM, and a first-order approximation for the precursors $SO_2$ and $NO_x$ whose emissions do not only affect $SO_2$ and $NO_x$ gas concentrations but also lead to several secondary products ($SO_2$ forms ammonium sulfate, $NO_x$ leads to $O_3$, ammonium nitrate). Experiment P2 perturbs $SO_2$ only, while experiment P3 perturbs $NO_x$ only (in this latter case, to limit the computational cost, computed for a limited set of representative source regions only).

We first test the hypothesis that the $PM_{2.5}$ response to the combined ($NO_x$ + $SO_2$) -20% perturbation (P1) can be approximated by the sum of the single precursor perturbations responses (P2 + P3). Figure 2 summarizes the resulting change in $SO_4^{2-}$, $NO_3^-$, $NH_4^+$ and total inorganic $PM_{2.5}$ respectively for the selected source regions. For Europe, the emission perturbations were applied over all European countries simultaneously, hence the responses are partly due to inter-regional transport from other countries. Following findings result from the perturbation experiments P1, P2 and P3:

1. Sulfate shows a minor response to $NO_x$ emissions, and likewise nitrate responds only slightly to $SO_2$ emissions and both perturbations are additive. In general the response is one order of magnitude lower than the direct formation of $SO_4^{2-}$ and $NO_3^-$ from $SO_2$ and $NO_x$ respectively (Fig. 2a, b);
2. $NH_4$ responds to $NO_x$ and $SO_2$ emissions with comparable magnitudes and in an additive way (Fig. 2c);
3. The response of total sulfate, nitrate and ammonium to a combined $NO_x$ and $SO_2$ -20% perturbation can be approximated by the sum of the responses to the individual perturbations, i.e. P1 ≈ P2+P3 (Fig. 2d). Scatterplots between P1 and P2+P3 for the regional averaged individual secondary products and total inorganic $PM_{2.5}$ are shown in Fig. S7.1 of the SI.

From the combined [$SO_2$+$NO_x$] perturbation (P1), and the separate $SO_2$ perturbation simulations (P2), both available for all source regions, the missing $NO_x$ SR matrices have been gap-filled using (P1 – P2). By lack of simulations for combined ($SO_2$ + $NH_3$) or ($NO_x$ + $NH_3$) perturbations we assume additivity for simultaneous $NH_3$, $SO_2$ and $NO_x$ perturbations, i.e. the response is computed from a linear combination of P2, P3 and P4.

7) Within section 3 on the evaluation of TM5-FASST numerous references are made to the ability of FASST to predict TM5 concentrations or other metrics using the gradient of the straight line fit as an estimate of bias. I have noticed a couple of times in the text where FASST is stated to over or under estimate the comparison but the details in the figure do not agree with this statement, which could be due to the use of the gradient. I think that a more appropriate bias statistic such as normalised mean bias (or something similar) could be used to provide an evaluation of FASST rather than this simple linear fit. This occurs throughout Section 3 and please check that all comments are appropriate to the relevant figures.

REPLY:
This point is well taken, the slope of the fit was indeed not the most appropriate choice for evaluating the performance of FASST. We have omitted the linear fit in the figures, and leave only the 1:1 line as a reference. Instead we have calculated Normalized Mean Bias (NMB), Mean Bias (MB) and correlation coefficient as validation metrics in a consistent way across sections 3.1 and 3.2 when compare FASST to the full TM5 model, where

NMB = $\overline{(FASST - TM5)}/\overline{TM5}$

MB = $\overline{(FASST - TM5)}$

$\bar{Y}$ = average of all grid cells in region

Further, in section 3.1 (linearization error under strong emission perturbations) we focus on evaluating the perturbation term (delta), putting additional statistics on the total concentrations in the SI. In section 3.2 (comparison with high/low GEA emission scenarios) we show and discuss both totals for individual scenarios and deltas in the main text.

CHANGES TO MANUSCRIPT;
*New discussion in section 3.1.2 P21 L6*

Figure 7 illustrates the performance of the TM5-FASST approach versus TM5 for regional-mean annual mean ozone, health exposure metric 6mDMA1 (both evaluated as population-weighted mean), and for the crop-relevant exposure metrics AOT40 and M12 (both evaluated as area-weighted mean) over the extended emission perturbation range. In most cases the response (i.e. the *change* between base and perturbed case) to emission perturbations lies above the 1:1 line across the 4 metrics, indicating that FASST tends to over-predict the resulting metric (as a sum of base concentration and perturbation). Of the four presented metrics, AOT40 is clearly the least robust one, which can be expected for a threshold-based metric that has been linearized. Tables 5 to 7 give the statistical metrics for the grid-to-grid comparison of the perturbation term between FASST and TM5 for the health exposure metric 6mDMA1, and crop exposure metrics AOT40 and M12 respectively. Statistical metrics for the total absolute concentrations (base concentration + perturbation term) are given in Tables S7.2 to S7.4 in the SI. As anticipated, the $NO_x$-only perturbation terms are showing the highest deviation, in particular for a doubling of emissions, however combined $NO_x$-NMVOC perturbations are reproduced fairly well for all regions, staying within 33% for a -80% perturbation for all 3 exposure metrics, and within 38% for an emission doubling for 6mDMA1 and M12, while the AOT40 metric is overestimated by 76 to 126% for emission doubling. The total resulting concentration over the entire perturbation range for single and combined $NO_x$ and NMVOC perturbation agrees within 5% for 6mDMA1 and M12, and within 64% for AOT40. The mean bias is positive for both perturbations, for all metrics and over all analysed regions, except for crop metric M12 under a doubling of NMVOC emissions over Europe showing a small negative bias. The deviations for individual European receptor regions under single and combined NMVOC and $NO_x$ perturbations for health and crop exposure metrics are shown in Figs. S7.4 to S7.6 of the SI.

8) Within Section 3 a comparison has been made with air pollutant concentrations, health and climate metrics. However, no comparison has been made to other studies on the crop relevant metrics. The comparison of crop relevant metrics seems to have been excluded from the comparison. Is it possible to compare the results from FASST to other studies that have looked at the air pollution impact on crops to provide some evaluation of these metrics?

REPLY: We now include an intercomparison with a study on present-day global and regional crop losses

CHANGES TO MANUSCRIPT:
*New section 3.3.6*

**3.3.6 Present day O3 – crop losses**

Avnery et al. (2011) evaluate year 2000 global and regional $O_3$-induced crop losses for wheat, maize and soy bean, based on the same crop ozone exposure metrics as used in FASST, obtained with a global chemical transport model at 2.8°x2.8° resolution. Figure 18 compares their results (in terms of relative yield loss) with FASST (TM5) results based on RCP year 2000 for the globe and 3 selected key regions (Europe, North-America and East Asia). Despite the less-robust quantification of crop impacts from $O_3$ in a linearized reduced-form model set-up, we find that FASST reproduces the major features and trends across regions and crop varieties. Differences may be attributed to a variety of factors, including model resolution, model O3 chemistry processes, emissions, definition of crop growing season and crop spatial distribution.
*And new Figure 18:*

[Figure]

Figure 18:Year 2000 global and regional ozone-induced relative yield losses for 3 major crops, from Avnery et al. (2011) and from TM5-FASST (RCP year 2000), estimated from the 2 common exposure metrics M7 and AOT40 (see text), as well as the mean of both.

9) Please could the author make sure that all the equations provided within the manuscript are appropriately numbered. It appears that some have been but not all.

REPLY: OK, done

**Minor Specific Comments:**

10) Section 2.1, P3, Line 12 – Brackets needed round O3 as first time defined as ozone

REPLY: OK done

11) Section 2.1, P3, line 14 – When describing the particulate matter components I think some mention needs to be made here about Secondary Organic Aerosol (SOA). I think this comes later in the manuscript (section 2.3 P6) but I feel it would also be worth mentioning here with the initial model description

REPLY: Agree

CHANGES TO MANUSCRIPT: *P4 L16*

Biogenic secondary aerosol (BSOA) was included following the AEROCOM recommendation (Dentener et al., 2006; Kanakidou et al., 2005) which parameterized BSOA formation from natural VOC emissions as a fixed fraction of the primary emissions. The relative fraction compared to the anthropogenic POM emissions varied spatially, with a higher contribution in regions were the emissions of terpene emissions were higher.
SOA from anthropogenic emission was not explicitly included in the current simulations.

*And to the discussion: P31 L17*

The omission of secondary organic PM in TM5 is estimated to introduce a low bias in the base concentration of the order of 0.1 µg m$^{-3}$ as global mean however with regional levels in Central Europe and China up to 1 µg m$^{-3}$ in areas where levels of primary organic matter are reaching 20 µg m$^{-3}$ (Farina et al., 2010) indicating a relatively low contribution of SOA to total PM$_{2.5}$.

12) Section 2.1, P3, Line 26 – 'Although for most health and ecosystem impacts only the surface level fields are required, base simulation and perturbed pollutants concentrations were calculated and stored for the 25 vertical levels of the model as monthly means, and some air quality-relevant parameters as hourly or daily fields.' – I think some mention of the fact that to calculate climate relevant impacts requires 3D information of constituents and not just surface fields

REPLY: Agree

CHANGES TO MANUSCRIPT: *changed the relevant phrase to: (P5 L7)*

Although for most health and ecosystem impacts only the surface level fields are required, climate metrics (e.g. radiative forcing) require the full vertical column and profile information. Therefore base simulation and perturbed pollutant concentrations were calculated and stored for the 25 vertical levels of the model as monthly means, and some air quality-relevant parameters as hourly or daily fields.

13) Section 2.3, P4, Line 25 – reference should be made to the underlying effects of the particular meteorological year used i.e. 2001 in this case.

REPLY: We do not fully agree that this addition would fit in here as the phrase describes a general feature of AQ-SRM. However we added it in the 3th par where TM5-FASST_v0 is introduced.

CHANGES TO MANUSCRIPT: *P6 L27*

In the current version v0 of TM5-FASST the emission-concentration relationship is locally approximated by a linear function expressing the change in pollutant concentration in the receptor region upon a change in precursor emissions in the source region with the generic form $dC_y = SRC \times dE_x$ where $dC_y$ equals the change in the pollutant concentration compared to a reference concentration in receptor region $y$, $dE_x$ is the change in precursor emission compared to a reference emission in source region $x$, and SRC the source-receptor coefficient for the specific compound and source-receptor pair – in this case emulating atmospheric processes linked to the meteorology in 2001.
* * *
14) Section 2.3, P5, Line 4 – '(Where j =i in the case of a primary component)' – maybe this could be changed to '(where the concentration of a primary pollutants is directly related to its emission)'
* * *
REPLY: We intend here that a primary component does not change chemically after its emission.

CHANGES TO MANUSCRIPT: *As the phrase is rather redundant we removed it.*
* * *
15) Section 2.3, P6, Lines 1 – 7 – There seems to be confusion between the labelling of emitted precursors and concentrations of components as in this section they both seemed to have been referred to as j. Please clarify which letter is meant to represent each
* * *
REPLY: This was indeed wrongly indexed, thanks for spotting.

CHANGES TO MANUSCRIPT*: P7 L29*

For each receptor point $y$ (i.e. each model vertical level 1°x1° grid cell), the change in concentration of component $j$ in receptor $y$ resulting from a -20% perturbation of emitted precursor $i$ in source region $x$, is expressed by a unique SR coefficient $A_{ij}[x,y]$:

$$A_{ij}[x,y] = \frac{\Delta C_j(y)}{\Delta E_i(x)} \text{ with } \Delta E_i(x)=0.2E_{i,base}(x) \tag{1}$$

The total concentration of component $j$ in receptor region $y$, resulting from arbitrary emissions of *all $n_i$* precursors $i$ at *all $n_x$* source regions $x$, is obtained as a perturbation on the base-simulation concentration, by summing up all the respective SR coefficients scaled with the actual emission perturbation:

$$C_j(y) = C_{j,base}(y) + \sum_{k=1}^{n_x} \sum_{i=1}^{n_i} A_{ij}[x_k,y] \cdot \left[E_i(x_k) - E_{i,base}(x_k)\right] \tag{2}$$
* * *
16) Section 2.3, P6, Equation 1 – Are these Source receptor coefficients calculated on the monthly or annual response between the precursor emission and pollutant? This needs to be stated within the description of the equation.
* * *
REPLY: Emission perturbations are implemented on annual basis, and the change in the source-receptor pollutant concentrations are evaluated on an annual basis as well. However some exposure metrics are based on seasonal values (e.g. crop growing season, human exposure to O3 during highest 6 monthly mean of hourly maximum values). We extended the paragraph, including as well additional information on the treatment of residual water in PM$_{2.5}$ to address an issue raised by Ref #2.

CHANGES TO MANUSCRIPT: *added phrase after Eq. 1(P8 L1)*

In the present version TM5-FASST_v0, the SR coefficients for pollutant concentrations are stored as annual mean responses to annual emission changes. Individual $PM_{2.5}$ components SRs are stored as dry mass ($\mu g\ m^{-3}$). $PM_{2.5}$ residual water at 35% is optionally calculated a posteriori for sensitivity studies, assuming mass growth factors for ammonium salts of 1.27 (Tang, 1996) and for sea-salt of 1.15 (Ming and Russell, 2001). The presence of residual water in $PM_{2.5}$ is not irrelevant: epidemiological studies establishing $PM_{2.5}$ exposure-response functions are commonly based on monitoring data of gravimetrically determined PM2.5, for which measurement protocols foresee filter conditioning at 30 – 50% RH. Therefore, although most health impact modelling studies consider dry $PM_{2.5}$ mass, the residual water fraction should in principle be included in modelled PM2.5.
We also established SR matrices linking annual emissions to specific $O_3$ exposure metrics that are based on seasonal or hourly $O_3$ concentrations (e.g. crop exposure metrics based on daytime ozone during crop growing season, human exposure to $O_3$ during highest 6 monthly mean of hourly maximum values).

*And deleted the phrase below Eq (3):*

"In TM5-FASST_v0 the monthly perturbations are aggregated to annual emission-concentration SR matrices, as the health, climate and vegetation impact metrics used in this version are also aggregated to annual values."
* * *
17) Section 2.3, P6, Line 21 - 24 – It is unclear to me how secondary organic aerosol (SOA) is included within the TM5-FASST tool as a component of PM2.5. Does it form part of the POM and what fraction of the primary emissions are used
* * *
REPLY:  This is partly addressed in our reply to comment 10). Further we specify now that the perturbation simulations are made for anthropogenic components only.

CHANGES TO MANUSCRIPT: *Paragraph was modified as follows (P7 L14)*

The SR matrices, describing the concentration response in each receptor upon a change in emissions in each source region, have been derived from a set of simulations with the full chemical transport model TM5 by applying -20% emission perturbations for each of the 56 defined source regions (plus shipping and aviation), for all relevant anthropogenic precursor components, in comparison to a set of unperturbed simulations, hereafter denoted as 'base simulations'. Emissions from biogenic organic components were included as a spatial/temporally varying component, but did not vary in the model emission sensitivity simulations. Consequently absolute concentrations of BSOA were identical across base and perturbation simulations and no SR coefficients are available.
* * *
18) Section 2.3, P7, Line 3 – The combination of emissions perturbation scenarios is given in Table 2. Did the base simulation not conduct emission perturbation scenarios for all 56 continental regions? I thought that this would have been essential to enable to the calculation of changes in concentrations in TM5 but Table 2 does not seem to imply this. Clarification required
* * *
REPLY: We agree that the phrase is formulated confusingly and deserves more clarification. The purpose of the perturbation simulations is indeed to obtain SR matrices for each precursor, and for each of the source regions, but it was not required to run all individual perturbations for all

regions. Table 2 explains in brief the purpose of each simulation and section 3.1 explains in detail how the various simulations are combined to get to the full set.

CHANGES TO MANUSCRIPT: *Added to section 2.3: (P9 L8)*

The -20% perturbation simulations were performed for the combination of precursors given in Table 2, with P0 the unperturbed reference simulation, and P1 through P5 -20% perturbations for combined or single precursors. Due to limited CPU availability, precursors that are expected not to interact chemically are perturbed simultaneously, with P1 combining $SO_2$, $NO_x$, BC, and POM and P4 combining $NH_3$ and NMVOC. P1 and P4 were computed for each of the 56 continental source regions plus shipping (P1 and P4) and aviation (P1). Additionally, a $SO_2$-only perturbation was computed for all individual source regions and shipping (P2) and $NO_x$-only for a selection of key source regions (P3). Finally a set of combined $NO_x$ + NMVOC perturbation simulations (P5) was performed for a set of key regions.
For a limited set of representative source regions, an additional wider range of emission perturbations $P_i'$ [-80% to +100%] has been applied to evaluate possible non-linearities in the emission-concentration relationships. The list of these additional perturbation simulations is given in Table S3 of the SI. In section 3.1 we explain how this set of perturbation runs is combined into FASST to obtain a complete set of source-receptor matrices for each precursor and source region.

*Modified Section 3.1*

**3.1 Validation against the full TM5 model: additivity and linearity**
We recall that the TM5-FASST computes concentrations and metrics based on a perturbation approach, i.e. the linearization applies only on the difference between scenario and reference emission. Therefore we focus on evaluating the perturbation response, i.e. the second term in the right hand side of Eq. 2.
The standard set of -20% emission perturbation simulations, available for all 56 continental source regions and constituting the kernel of TM5-FASST_v0 are simulations P1 (perturbation of $SO_2$, $NO_x$, BC and POM), P2 ($SO_2$ only), and P4 ($NH_3$ and NMVOC) shown in Table 2. Additional standard -20% perturbation experiments P3 ($NO_x$ only) and P5 ($NO_x$ and NMVOC), as well as an additional set of perturbation simulations P1' to P5' over the range [-80%, +100%], listed in Table S3 of the SI, have been performed for a limited selection of representative source regions (Europe, USA, China, India, Japan) due to limited CPU resources. For the same reason, no combined perturbation studies are available for ($SO_2$ + $NH_3$) and ($NO_x$ + $NH_3$) for a systematic evaluation of additivity and linearity. The available [-80%, +100%] perturbations are used to validate the linearized reduced-form approach against the full TM5 model, exploring chemical feedback mechanisms (additivity) and extrapolation of the -20% response sensitivity towards larger emission perturbation magnitudes (linearity). This is in particular relevant for the $NO_x$ - NMVOC - $O_3$ chemistry and for the secondary $PM_{2.5}$ components $NO_3^-$ - $SO_4^{2-}$ - $NH_4^+$. These mechanisms could also be important for organic aerosol, but we remind that in this study organic aerosol formation was parameterized as pseudo-emissions.
* * *
19) Section 2.3, P7, Line 15 to 18 – The change in CH4 burden in TM5 from the HTAP1 perturbation simulations is stated as being an emission perturbation of 77 Tg/year. Could the authors provide information on how this was obtained
* * *
REPLY: The value comes from the assumption that the imposed $CH_4$ steady state concentration is the result of a balanced emission on the one hand and the chemical loss by oxidation by OH on the other hand (neglecting the lower-order losses to soil and stratosphere). As the TM5 model keeps track of the total amount of $CH_4$ oxidized, the implied change in emission is simply obtained from the difference in total amount of $CH_4$ oxidized in 1 year between the two runs.

We agree this could be explained better. In order to address a similar comment from Ref. #2 we have moved the details of the methodology to the SI, and modified the text as follows:

CHANGES TO MANUSCRIPT: *Main text, P9 L25:*

Annex S3 in the SI provides more details on the methodology applied to convert the $CH_4$ concentration perturbation into a $CH_4$ emission-based perturbation

*Annex S3:*

**S3.1 $CH_4$ – $O_3$ source-receptor relations from HTAP1 perturbation experiments:**
$CH_4$ emissions lead to a change in $CH_4$ concentrations with a perturbation response time of about 12 years. In order to avoid expensive transient computations, HTAP1 simulations SR1 and SR2 with prescribed fixed $CH_4$ concentrations (1760 ppb and 1408 ppb, see Dentener et al., 2010) were used to establish $CH_4$ – $O_3$ response sensitivities. Previous transient modeling studies have shown that a change in steady-state $CH_4$ abundance can be traced back to a sustained change in emissions, but the relation is not linear because an increase in $CH_4$ emissions removes an additional fraction of atmospheric OH (the major sink for $CH_4$) and prolongs the lifetime of $CH_4$ (Fiore et al., 2002, 2008; Prather et al., 2001).
In a steady-state situation, the $CH_4$ concentration is the result of balanced sources and sinks. In the HTAP1 experiments, keeping all other emissions constant, the change in the amount of $CH_4$ loss (mainly by OH oxidation with a lifetime of ca. 9 years, neglecting loss to soils and stratosphere with lifetimes of ca.160 and 120 years respectively (Prather et al., 2001) ) under the prescribed change in $CH_4$ abundance should therefore be balanced by an equal and opposite source which we consider as an "effective emission". The amount of $CH_4$ oxidized by OH in one year being diagnosed by the model, the resulting difference between the reference and perturbation experiment of -77 Tg sets the balancing "effective" emission rate to 77Tg/yr, which is then used to normalize the resulting $O_3$ and $O_3$ metrics response to a $CH_4$ emission change.

The same perturbation experiments also allow us to establish the $CH_4$ self-feedback factor F describing the relation between a change in emission and the change in resulting steady-state concentration:
$$\frac{C_2}{C_1} = \left(\frac{E_2}{E_1}\right)^F \tag{S3.1}$$
With $CH_4$ concentrations prescribed, $CH_4$ emissions were not included in the SR1 and SR2 experiments. The feedback factor F is derived from model-diagnosed respective $CH_4$ burdens (B) and total lifetimes (LT) as follows (Fiore et al., 2009; Wild and Prather, 2000):
$F=1/(1-s)$
$s = \partial\ln(LT) / \partial\ln(B)$
TM5 returns s = 0.33 which can be compared to a range of values between 0.25-and 0.31 in IPCC-TAR (Prather et al., 2001, Table 4.2) , resulting in a TM5-inherent calculated feedback factor F=1.5. This factor can be used to estimate the corresponding SR2-SR1 change in $CH_4$ emission in a second way. From Eq. S3.1 we find that a 20% decrease in $CH_4$ abundance corresponds to a 14% decrease in total $CH_4$ emissions. Kirschke et al. (2013) estimate total $CH_4$ emissions in the 2000s in the range 550 – 680 Tg yr[-1] from which we obtain an estimated emission change between the HTAP SR1 and SR2 experiments in the range 77 – 95 Tg yr[-1], in line with our steady-state loss-balancing approach.

20) Section 2.3, P7, Lines 22 to 28 – It is stated that FASST does not include impacts on O3 from perturbations in CO emissions. I am not sure why this has not been included in the development of FASST along with other O3 precursor emissions of NOx and NMVOCs. Within this section it states that there is a dedicated CO emission perturbation experiment conducted with TM5 as part of HTAP1 available and that the impacts on O3 are not insignificant. Therefore I wonder why the information from the TM5 CO experiments have not been included previously within FASST?

REPLY:  This is indeed a missing link in the TM5-FASST model which we hope to address in a future version of the tool. Also here, missing CPU resources did not allow for dedicated CO perturbation simulations in each of the 56 source regions. Indeed from HTAP1, source receptor relations between large rectangular source areas (not aligned with political borders and coast lines, and including ocean) are available but we did not attempt to remap those on the FASST 56 continental regions, given expected differences in CO lifetimes for emissions from these regions. With HTAP2 source regions better aligned with the FASST ones, there may be possibilities to rely on those in future developments. This caveat has been mentioned in the discussion.

21) Section 2.4, P8 – Maybe this section should be labelled as something like 'Urban Adjustments in PM$_{2.5}$ for Health Calculation' to better identify what is being done here. I am assuming that the adjusted PM$_{2.5}$ concentrations are only used within the calculation of health impacts

REPLY:  Thank you for the suggestion – indeed this is relevant for the exposure of population. As this section now also includes a discussion on the impact of grid resolution (see reply to comment 4) we have modified the title

CHANGES TO MANUSCRIPT: *Title changed to:*

**2.4  PM2.5 adjustments in urban regions for health impact evaluation**

22) Section 2.4, P8, Lines 25 -26 – Is the CIESIN population dataset the default one used within FASST as this seems to have been used to calculate the default urban increment factors in Table S4.2? Might be worth included which one is recommended for use.

REPLY: The CIESIN dataset is the one with the highest resolution and therefore most suitable for a sub-grid correction. The 'default' regional increment factors are indeed based on CIESIN year 2000 data, but they are static and therefore do not change with scenario years. The public web tool always uses these default factors, but the (not-public) 'research version' has the option to include more appropriate population data sets.

CHANGES TO MANUSCRIPT: *we added the following phrase in the conclusion section: (P33 L30)*

This version offers the possibility to explore built-in as well as user-defined scenarios, using static default urban increment correction factors and crop production data. A more sophisticated in-house research version with gridded output and flexibility in the choice of gridded ancillary data (population grid maps, scenario-specific urban increment factors, crop distribution) is under continuous development and has been applied for the assessments listed in table S1.

23) Section 2.5, P9, Line 10 – Check definition of AF here as this does not match up with what is provided further down the page, just above line 20

REPLY: In fact it does correspond: 1-1/RR = (RR-1)/RR. But as the right-hand form is probably more legible we changed it to the latter. The part of the text above line 20 containing the larger equation has been moved to the SI following comment 4.

CHANGES TO MANUSCRIPT: *changed the phrase to: (P11 L21)*

… where $m_0$ is the baseline mortality rate for the exposed population, *AF = (RR-1)/RR* is the fraction of total mortalities attributed to the risk factor (exposure to air pollution)
* * *
24) Section 2.5, P10, Lines 17 to 24 – I think a comment is required here to state how the recent updates in the epidemiological evidence for health effects could impact on the predictions in FASST i.e. will they be cause an underestimate or overestimate.
* * *
REPLY: We have added a line to clarify the impact of the new parameter on the estimated health impact for PM2.5.

CHANGES TO MANUSCRIPT: *extended the phrase as follows: (P12 L6)*

In particular, the theoretical minimum risk exposure level was assigned a uniform distribution of 2.4–5.9 $\mu g/m^3$ for $PM_{2.5}$, bounded by the minimum and fifth percentiles of exposure distributions from outdoor air pollution cohort studies, compared to the presently used range of 5.8 - 8.8 $\mu g/m^3$ which would increase the health impact from $PM_{2.5}$ in relatively clean areas.
* * *
25) Section 2.6, P11, Line 14 – 'Both Mi metrics …' should be changed to 'Both metrics (Mi) …
* * *
REPLY: OK done
* * *
26) Section 2.6, P11, Line 15 – How is the growing season defined when calculating the crop metrics?
* * *
REPLY: As reported in the text, the growing seasons for the respective crops are retrieved from the gridded GAEZ data set. To clarify this more, we have extended the description of methodology related to the definition of the crop season.

CHANGES TO MANUSCRIPT: *modified section 2.6 as follows:*

**2.6 Crop impacts**

The methodology applied in TM5-FASST to calculate the impacts on four crop types (wheat, maize, rice, and soy bean) is based on Van Dingenen et al. (2009). In brief, TM5 base and -20% perturbation simulations of gridded crop $O_3$ exposure metrics (averaged or accumulated over the crop growing season) are overlaid with crop suitability grid maps to evaluate receptor region-averaged exposure metrics SR coefficients. Gridded crop data (length and centre of growing period, as well as a gridded crop-specific suitability index, based on average climate 1961 – 1990) have been updated compared to Van Dingenen et al. (2009), using the more recent and detailed Global Agro-Ecological Zones (GAEZ) data set (IIASA and FAO, 2012, available at http://www.gaez.iiasa.ac.at/).
Available crop ozone exposure metrics are 3-monthly accumulated ozone above 40 ppbV (AOT40) and seasonal mean 7 hr or 12 hr day-time ozone concentration (M7, M12) for which exposure-response functions are available from the literature (Mills et al., 2007; Wang and Mauzerall, 2004). Both metrics ($M_i$) are calculated as the 3-monthly mean daytime (09:00 – 15:59 for M7, 08:00 – 19:59 for M12) ozone concentration, evaluated over the 3 months centred on the midpoint of the location-dependent crop-growing season provided by the GAEZ data set.

Note that in the GAEZ methodology, the theoretical growing season is determined based on prevailing temperatures and water balance calculations for a reference crop, and can range between 0 and 365 days, however our approach always considers 3 months as the standard metric accumulation or averaging period.

27) Section 2.6, P11, Line 16 – RYL is defined as the crop relative yield. Should this be the relative yield loss? Also the coefficients a,b,c within the equation for RYL need more explanation

REPLY:  indeed, "RYL" was wrongly positioned in the phrase. We have included a table with the values of the coefficients in the equations. While in the Weibull function the a and b parameters are pure mathematical shape coefficients, the c coefficients sets the lower threshold value for zero impact. We included this as well.

CHANGES TO MANUSCRIPT:

*Modified following section (P12 L25):*
Both metrics ($M_i$) are calculated as the 3-monthly mean daytime (09:00 – 15:59 for M7, 08:00 – 19:59 for M12) ozone concentration, evaluated over the 3 months centred on the midpoint of the location-dependent crop-growing season.
The crop relative yield loss (RYL) is calculated as linear function from AOT40 and from a Weibull-type exposure-response as a function of Mi:

$$RYL[AOT40] = a \times AOT40 \tag{5}$$

$$\left. \begin{array}{ll} RYL(M_i) = 1 - \dfrac{exp\left[-\left(\frac{Mi}{a}\right)^b\right]}{exp\left[-\left(\frac{c}{a}\right)^b\right]} & M_i \geq c \\[4mm] RYL(M_i) = 0 & M_i < c \end{array} \right\} \tag{6}$$

The parameter values in the exposure response functions are given in Table 3. Note that for $M_i$ = *c, RYL = 0* hence *c* is the lower $M_i$ threshold for visible crop damage. Also here, the non-linear shape of the RYL(Mi) function requires the ΔRYL for 2 scenarios (S1, S2) being evaluated as RYL($M_{i,S2}$) – RYL ($M_{i,1}$), and not as RYL ($M_{i,S2}$- $M_{i,S1}$).

28) Section 2.7.1, P12, Lines 10 to 12 – Are these two sentences on the basic radiative properties of aerosols relevant? Including some text on the following lines would be good to discuss how the treatment of externally mixed aerosols alters the radiative forcing calculations when compared to internally mixed ones (Lesins et al., 2002; Klingmüller et al., 2014).

REPLY:  We agree on the redundancy of the two sentences and removed them. We included a brief discussion on the impact of the introduced simplifications regarding mixing state as well as the use of integrated column burden instead of resolved vertical profiles. With respect to the mixing state we rather refer to Bond et al. (2013) who considered various additional processes affecting the BC absorption coefficient

CHANGES TO MANUSCRIPT:  *Added following text: (P14 L15)*

Neglecting the aerosol mixing state and using column-integrated mass rather than vertical profiles introduces additional uncertainties in the resulting forcing efficiencies. Accounting for internal mixing may increase the BC absorption by 50 to 200% (Bond et al., 2013), while including the vertical profile would weaken BC forcing and increase SO4 forcing (Stjern et al.

2016). Further, the BC forcing contribution through the impact on snow and ice is not included, nor are semi- and indirect effects of BC on clouds. Our evaluation of pre-industrial to present radiative forcing in the validation section demonstrates that, in the context of the reduced-form FASST approach, the applied method however provides useful results..
* * *
29) Section 2.7.2, P12 – I think this sections needs to be made clearer. I am struggling to make the link between the output from FASST and the calculation of indirect aerosol forcing. How is done? What fields from FASST are used to calculate it? Needs to explain the methodology better for the reader.
* * *
REPLY: Apologies if the manuscript lacked clarity on this issue. Equation (7) explains how FASST SR matrices for radiative forcing are obtained: the change in forcings (both direct and indirect) for the perturbation experiments are computed from TM5-output using normalized forcing efficiencies. FASST then simply contains a SR coefficient to be multiplied with the emission change to obtain a forcing change. Sections 2.7.1 and 2.7.2. describe the underlying methodology in TM5. We have added some more clarification as follows:

CHANGES TO MANUSCRIPT: *modified section 2.7.3 as follows:*

**2.7.3 Radiative forcing by $O_3$ and $CH_4$**

Using TM5 output, indirect forcing is evaluated considering only the so far best studied first indirect effect, and using the method described by Boucher and Lohmann (1995). Fast feedbacks on cloud lifetimes and precipitation were not included in this off-line approach. This simplified method uses TM5 3D time-varying fields of SO4 concentrations, cloud liquid water content, and cloud cover (the latter from the parent ECMWF meteorological data). The parameterization uses the cloud information (liquid water content and cloud cover) from the driving ECMWF re-analysis data (year 2001). Fast feedbacks on cloud lifetimes and precipitation were not included in this off-line approach. The cloud droplet number concentrations and cloud droplet effective radius were calculated following Boucher and Lohmann (1995) separating continental and maritime clouds. The equations are given in section S6 of the SI. The global indirect forcing field associated with sulfate aerosols is shown in Fig. S6.1(d) of the SI. Indirect forcing by clouds remains however highly uncertain, and although FASST evaluates its magnitude, it is often not included in our analyses.
* * *
30) Section 2.7.2, P12, Line 29 – Add year used to meteorological data
* * *
REPLY: done (see previous comment)
* * *
31) Section 2.7.2, P12, Line 30 – missing word 'using' between after 'calculated'. Also it is probably worth stating here or in the supplementary section S6 the equations used to calculate cloud droplet number concentrations and cloud effective radius
* * *
REPLY: done

CHANGES TO MANUSCRIPT:
*Following section was added to section S6 of the SI:*

Indirect forcing:
The cloud droplet number concentrations (*CDNC*) were calculated using the following set of equations from Boucher and Lohmann (1995), separating continental and maritime clouds:

$$CDNC_{cont}^{St} = 10^{2.24+0.257log(m_{SO_4})}$$

$$CDNC_{cont}^{Cu} = 10^{2.54+0.186log(m_{SO_4})}$$
$$CDNC_{ocean} = 10^{2.06+0.48log(m_{SO_4})}$$

Following Boucher and Lohmann (1995), the cloud droplet effective radius is calculated from the mean volume cloud droplet radius:

$$r_e = 1.1 \left( \frac{l\rho_{air}}{(4/3)\pi\rho_{water}CDNC} \right)^{1/3}$$

Where $l$ = cloud liquid water content, $\rho_{air}$ = air density, $\rho_{water}$ = water density

32) Section 2.7.3, P13 – Like section 2.7.2. I think this section needs to be made clearer to highlight what output is being used from FASST to compute $O_3$ and $CH_4$ radiative forcings. There is a lot of details of what is included but I struggled to follow the basic principle of FASST output + forcing efficiency = radiative forcing. I think the description of what is done in FASST should come first at the start of this paragraph and then follow with the description of what it takes account of.

REPLY: We apologize for the lack of clarity. The section was indeed not very clear in explaining the methodology used in TM5 and how this is transferred into FASST. We have modified the introductory part of section 2.7 to explain the general approach: TM5 provides radiative forcing output from a built-in methodology, and the forcing SRs in FASST are simply based on emission-normalized delta's between base and perturbation experiments. The subsequent sections then explain in more detail how forcing is calculated in TM5.
Further we have shortened section 2.7.3 and moved the details of the methodology to the SI (new section S6.2)

CHANGES TO MANUSCRIPT: *we modified the introductory part of the section and the section addressing radiative forcing by O3 and CH4 as follows:*

**2.7 Climate metrics**
We make use of the available 3D aerosol and $O_3$ fields in the -20% emission perturbation simulations with TM5 to derive the change in global forcing for each of the perturbed emitted precursors. The region-to-global radiative forcing SR for precursor *j*, emitted from region *k*, is calculated as the emission-normalized change in global radiative forcing between the TM5 base and the corresponding -20% emission perturbation experiment:

$$SR\_RF_k^j = \frac{RF\_PERT[j,k] - RF\_BASE}{0.2E_k^j} \ [W/m^2]/[kg/yr] \tag{7}$$

where RF_PERT and RF_BASE are the TM5 global radiative forcings for the perturbation and base simulations respectively, and $E_k^j$ is the annual base emission of precursor *j* from region *k*. For each emitted pollutant (primary and secondary) the resulting normalized global forcing responses are then further used to calculate the global warming potential (GWP) and global temperature potential (GTP) for a series of time horizons H. In this way, a set of climate metrics is calculated with a consistent methodology as the air quality metrics, health and ecosystem impacts calculated from the concentration and deposition fields. In this section we describe in more detail the applied methodologies in TM5 to obtain the radiative forcing from aerosols, clouds and gases, as well as the derivation of the GWP and GTP metrics.

(…)

**2.7.3 Radiative forcing by $O_3$ and $CH_4$**
Using TM5 output, radiative forcing (RF) by ozone is approximated using the forcing efficiencies obtained by the STOCHEM model as described in Dentener et al. (2005), normalized by the ozone columns obtained in that study. Here we use annual averaged forcing based on the

RF computations provided as monthly averages by D. Stevenson (personal communication, 2004). The radiative transfer model was based on Edwards and Slingo (1996). These forcings account for stratospheric adjustment, assuming the fixed dynamical heating approximation, which reduces instantaneous forcings by ~22%.

For $CH_4$ the RF associated with the base simulation was taken from the equations in the IPCC-Third Assessment Report (TAR) (Table 6.2 of Ramaswamy et al., 2001). Using the HTAP1 calculated relationship between CH4 concentration and emission, and the same equations, we evaluated a globally uniform value of 2.5 mW/m² per Tg $CH_4$ emitted. (Dentener et al., 2010). It includes both the direct $CH_4$ greenhouse gas (GHG) forcing (1.8 mW/m²) as well as the long-term feedback of $CH_4$ on hemispheric $O_3$ (0.7 mW/m²).

From the TM5 perturbation experiments we derive as well region-to-global radiative forcing SRs for precursors ($NO_x$, NMVOC, CO and $SO_2$) through their feedback on the $CH_4$ lifetime and subsequently on long-term hemispheric $O_3$ levels. Hence, the greenhouse gas radiative forcing contribution of each ozone precursor consists of 3 components: a direct effect through the production of $O_3$, a contribution by a change in $CH_4$ through modified OH levels (including a self-feedback factor accounting for the modified $CH_4$ lifetime), and a long-term contribution via the feedback of $CH_4$ on hemispheric ozone. The details of the applied methodology are given in section S6.2 of the SI.

In its current version, TM5-FASST_v0 provides the steady-state concentrations and forcing response of the long-term $O_3$ and $CH_4$ feedback of sustained precursor emissions, i.e. it does not include transient computations that take into account the time lag between emission and establishment of the steady-state concentration of the long-term $O_3$ and $CH_4$ responses.

*And in the SI:*

**S6.2 Secondary forcing feedbacks of O3 precursors on CH4 and background O3**

Emissions of short-lived species (NOx, NMVOC, CO, $SO_2$) influence the atmospheric OH burden and therefore the $CH_4$ atmospheric lifetime, which in turn contributes to long-term change in CH4 and background ozone. Hence, the total forcing contribution from O3 precursors consists of a short-term direct contribution from immediate O3 formation (S-O3), and secondary contributions from $CH_4$ (I-CH4) and a long-term feedback from this $CH_4$ on background O3 (M-O3).

We apply the formulation by (Fiore et al., 2009; Prather et al., 2001; West et al., 2007) to calculate the secondary change in steady-state $CH_4$ from SLS emissions, using the TM5 perturbation experiments for FASST (see section S3). TM5 diagnoses the $CH_4$ loss by oxidation for reference and perturbation run (where the emissions of SLS are decreased with -20%), from which we calculate the $CH_4$ oxidation lifetime ratio between reference and perturbation:

$$\frac{LT_P}{LT_{Ref}} = \frac{CH4\_ox_P}{CH4\_ox_{Ref}}$$

[S6.5]

Where LT is the CH4 lifetime against loss by OH oxidation, and $CH4\_ox$ = the amount (Tg) of CH4 oxidized.

The new steady-state methane concentration $M$ due to the changing lifetime from perturbation experiment P, induced by $O_3$ precursor emissions follows from (Fiore et al., 2008, 2009; Wild and Prather, 2000):

$M = M_0 \times \left(\frac{LT_P}{LT_{ref}}\right)^F$ where $M_0 = 1760$ ppb, the reference $CH_4$ concentration and F = 1.5, determined from the HTAP1 CH4 perturbation experiments, as described in section S3.

The change in $CH_4$ forcing (I-CH4) associated with the change to the new steady-state concentration is obtained from IPCC AR5 equations:

$$\Delta F = \alpha\left(\sqrt{M} - \sqrt{M_0}\right) - \left(f(M, N_0) - f(M_0, N_0)\right) \qquad [S6.6]$$

$$f(M, N) = 0.47ln[1 + 2.01 \times 10^{-5}(MN)^{0.75} + 5.31 \times 10^{-15}M(MN)^{1.52}] \qquad [S6.7]$$

Where $M$, $M_0$ = $CH_4$ concentration in ppb, $N_0$ = $N_2O$ (=320 ppb)

The associated long-term O3 forcing (M-O3) per Tg precursor emitted is obtained by scaling linearly the change in $O_3$ forcing obtained in the HTAP1 $CH_4$ perturbation simulation (SR2–SR1), with the change in $CH_4$ obtained above, and normalizing by the precursor emission change (Fiore et al., 2009)

$$\Delta F = \frac{\Delta F_{O3}[SR2-SR1]}{M_{SR2} - M_{SR1}}(M - M_0) \qquad [S6.8]$$

The response of $CH_4$ and $O_3$ forcing to CO emission changes (for which no regional TM5-FASST perturbation model simulations were performed) was taken from TM5-CTM simulations performed for the HTAP1 assessment (Dentener et al., 2010) using the average forcing efficiency for North America, Europe, South-Asia and East-Asia. For regions not covered by the HTAP1 regions, the HTAP1 rest-of-the-world forcing efficiency was used.
The resulting region-to-globe emission-based forcing efficiencies are given in Tables S6.2 to S6.5 for aerosols, CO, $CH_4$ and other $O_3$ precursors respectively.
* * *
33) Section 2.7.3, P13, Lines 4 to 6 – How do these STOCHEM calculations compare to the ACCMIP multi-model mean and is it still appropriate?
* * *
REPLY: We have not made ourselves the comparison between STOCHEM and ACCMIP normalized O3 radiative forcings. However Stevenson et al. (2013)calculated a global normalized RF of 42 mWm$^{-2}$ DU$^{-1}$, while two other model studies find values of about 36 mWm$^{-2}$ DU$^{-1}$. In this study a value 30 mWm$^{-2}$ DU$^{-1}$ was found, broadly in line with the global numbers above. The results of Stevenson et al. (2013) were not available when the RF module was developed, and indeed updating the radiative transfer code, including ozone vertical profiles (instead of using fixed ozone columns) would be obvious candidates for improvement.
* * *
34) Section 2.7.3, P13, Line 32 – For regions not covered by the major HTAP1 source could the 'rest of the world' CO forcing efficiency not be used from Table S6.3 rather than a global average?
* * *
REPLY: This is indeed a correct observation; we have corrected the text and the values in Table S6.3.
* * *
35) Section 2.7.4, P14, Line 7 – Are the emission based forcing efficiencies those in Table S6.2 to S6.5? Can a reference be put in to these in the main text?
* * *
REPLY: OK done

CHANGES TO MANUSCRIPT: *we refer to the relevant tables in the SI in the respective sections 2.7.1 (aerosols)*

*(P14 L22)* The regional emission-normalized forcing SRs for aerosol precursors (in W m$^{-2}$ Tg$^{-1}$) are given in Table S6.2 of the SI.

*2.7.2 (indirect forcing)*
*(P15 L6)* The global indirect forcing field associated with sulfate aerosols is shown in Fig. S6.1(d) of the SI an regional forcing SRs are listed in Table S6.2

*(P15 L25)* The details of the applied methodology for direct and indirect $CH_4$ forcing SRs are given in section S6.2 of the SI, including tables with the regional forcing efficiencies for all precursors (Tables S6.3 to S6.5).

*And in the first line of section 2.7.4:*
*(P16 L2)* The obtained emission-based forcing efficiencies (Tables S6.2 to S6.5 in the SI) are immediately useful for evaluating a set of short-lived climate pollutant climate metrics.
* * *
36) Section 3, P15, Lines 19 to 21 – Simplify point 1 to read better
* * *
REPLY: agree, we have rephrased the introduction of this section as follows

CHANGES TO MANUSCRIPT:

**3 Results: validation of the reduced-form TM5-FASST**

In this section we focus on the validation of regionally aggregated TM5-FASST_v0 outcomes (pollutant concentrations, exposure metrics, impacts), addressing specifically:
1    The additivity of individual pollutant responses as an approximation to obtain the response to combined precursor perturbations,
2    The linearity of the emission responses over perturbation ranges extending beyond the -20% perturbation
3    The FASST outcome versus TM5 for a set of global future emission scenarios that differ significantly from the reference scenario
4    FASST key-impact outcomes versus results from the literature for some selected case studies, with a focus on climate metrics, health impacts and crops.
* * *
37) Section 3.1, P16, Line 2 – reference is made to Annex 4 of the SI. Pleases clarify this reference as there is no Annex 4
* * *
REPLY: Indeed thanks for spotting.

CHANGES TO MANUSCRIPT: *reference is now correctly made to Table S3 (P17 L26)*

Additional standard -20% perturbation experiments P3 ($NO_x$ only) and P5 ($NO_x$ and NMVOC), as well as an additional set of perturbation simulations P1' to P5' over the range [-80%, +100%], listed in Table S3 of the SI, have been performed for a limited selection of representative source regions (Europe, USA, China, India, Japan) due to limited CPU resources.
* * *
38) Section 3.1.1, P16, Lines 12 to 14 – Is there a reason for the particular representative source regions selected in Table 2 e.g. South Africa for NOx
* * *
REPLY: In order to optimize computing time, NOx-only as well as the combined NOx-NMVOC perturbation regions were selected based on their presumed relevance in terms of impact, pace of expected emission changes in the future and geographical representativeness. South Africa was included as a case of rapidly developing economy in the Southern hemisphere and a possible case where it may be "safer" to explicitly calculate the NOx SR rather than applying gap filling.
* * *
39) Section 3.1.1, P16, Lines 19 to 22 – The explanation on these lines could be simplified
* * *
REPLY: done

CHANGES TO MANUSCRIPT: *we have rewritten the first part of section 3.1.1 as follows:*

**3.1.1 Additivity and linearity of secondary inorganic $PM_{2.5}$ response:**

Experiment P1, where BC, POM, $SO_2$ and $NO_x$ emissions are simultaneously perturbed by -20% relative to base simulation P0, delivers SR matrices for primary components BC and POM, and a first-order approximation for the precursors $SO_2$ and $NO_x$ whose emissions do not only affect $SO_2$ and $NO_x$ gas concentrations but also lead to several secondary products ($SO_2$ forms ammonium sulfate, $NO_x$ leads to $O_3$ and ammonium nitrate). Experiment P2 perturbs $SO_2$ only, while experiment P3 perturbs $NO_x$ only (in this latter case, to limit the computational cost, computed for a limited set of representative source regions only).

We first test the hypothesis that the $PM_{2.5}$ response to the combined ($NO_x$ + $SO_2$) -20% perturbation (P1) can be approximated by the sum of the single precursor perturbations responses (P2 + P3). Figure 2 summarizes the resulting change in $SO_4^{2-}$, $NO_3^-$, $NH_4^+$ and total inorganic $PM_{2.5}$ respectively for the selected source regionsFor Europe, the emission perturbations were applied over all European countries simultaneously, hence the responses are partly due to inter-regional transport from other countries. Following findings result from the perturbation experiments P1, P2 and P3:

    (1) Sulfate shows a minor response to $NO_x$ emissions, and likewise nitrate responds only slightly to $SO_2$ emissions and both perturbations are additive. In general the response is one order of magnitude lower than the direct formation of $SO_4^{2-}$ and $NO_3^-$ from $SO_2$ and $NO_x$ respectively.(Fig. 2a, b).

    (2) $NH_4$ responds to $NO_x$ and $SO_2$ emissions with comparable magnitudes and in an additive way (Fig. 2c)

    (3) A simultaneous -20% emission perturbation of $SO_2$ and $NO_x$ behaves in an additive manner for what concerns the formation of secondary $PM_{2.5}$, i.e. the response of total sulfate, nitrate and ammonium to a combined $NO_x$ and $SO_2$ perturbation can be approximated by the sum of the responses to the individual perturbations (Fig. 2d), i.e. P1 ≈ P2+P3. Scatterplots between P1 and P2+P3 for the regional averaged individual secondary products and total inorganic $PM_{2.5}$ are shown in Fig. S7.1of the SI
* * *
40) Section 3.1.1, P16, Lines 29 to 31 – Also there is a larger response to NO3 from increasing NOx emissions over India. Do you think that is this a particular issue for TM5 over India? Does this is cause issues for future prediction of NO3 aerosol from changes in NOx emissions over India?
* * *
REPLY: The reviewer correctly notices the large sensitivity of aerosol nitrate formation to NOx emissions in India. It is difficult to say whether this is a specific feature of TM5, or a more general feature of others models, as we are not aware of published sensitivity studies on NOx - aerosol NO3 in India. Moreover to our knowledge there are hardly any reliable NO3 observations available from India that could corroborate the calculated sensitivity. We will however highlight this feature in our paper, with a specific recommendation to devote more multi-model studies to this.

CHANGES TO MANUSCRIPT: *Modified / added following phrases: (P19 L2)*

The figure illustrates the general near-linear behaviour of regionally aggregated responses to single precursor emission perturbations for all regions, except for India where the linearity of the response to $NO_x$ emissions breaks down for emission reductions beyond -50%. For India we further observe a relatively strong nitrate response to NOx emissions, with $NO_3^-$ increasing by a factor of 3 for a doubling of $NO_x$ emissions. We are not aware of reliable observations or other published NOx-aerosol sensitivity studies from that region that could corroborate the calculated

sensitivity. Because such a feature may strongly affect projected future PM$_{2.5}$ levels and associated impacts, we recommend devoting regional multi-model studies to this aspect.
* * *
41) Section 3.1.1, P17, Lines 8 to 11 – I don't think you can say that errors in the -80% case are larger than +100% for NOx. They look similar to me
* * *
REPLY: This part of the section has been rewritten to comply with earlier comments on statistic metrics

CHANGES TO MANUSCRIPT: *We have rephrased the part of section3.1.1 dealing with linearity test under the large perturbations as follows: (P18 L30)*

Next we evaluate the hypothesis that the -20% perturbation responses can be extrapolated towards any perturbation range, as an approximation of a full TM5 simulation. Figure 3 shows, for the selected regions listed in Table S3 of the SI, the TM5 computed relative change in secondary PM$_{2.5}$ concentration versus the relative change in precursor emission in the range [-80%, +100%]. The figure illustrates the general near-linear behaviour of regionally aggregated responses to single precursor emission perturbations for all regions, except for India where the linearity of the response to NO$_x$ emissions breaks down for emission reductions beyond -50%. For India we further observe a remarkably strong nitrate response to NO$_x$ emissions, with NO$_3^-$ increasing by a factor of 3 for a doubling of NO$_x$ emissions, although the responses shown in Fig. 2 indicate that absolute changes (in µg m$^{-3}$) in NO$_3$ are relatively low and that secondary PM$_{2.5}$ in this region is dominated by SO$_4$. We are not aware of reliable observations or other published NO$_x$-aerosol sensitivity studies from that region that could corroborate this calculated sensitivity. Because such a feature may strongly affect projected future PM$_{2.5}$ levels and associated impacts, we recommend regional multi-model studies devote attention this feature Because the TM5-FASST linearization is based on the extrapolation of the -20% perturbation slope, concave-shaped trends in Fig. 3 indicate a tendency of TM5-FASST to over-predict secondary PM$_{2.5}$ at large negative or positive emission perturbations, and opposite for convex-shaped trends. Figure 4 illustrates the error introduced in regional secondary PM$_{2.5}$ concentrations responses when linearly extrapolating the regional -20% perturbation sensitivities to -80% (blue dots) and +100% (red dots) perturbations respectively. While the scatter plots for the single perturbations (Fig. 4 a,b,c) evaluate the linearity of the single responses, the panel showing the combined (SO$_2$+NO$_x$) perturbation (Fig. 4d) is a test for the linearity combined with additivity of SO$_2$ and NO$_x$ perturbations over the considered range. In general, the linear approximation leads to a slight over-prediction of the resulting secondary PM$_{2.5}$ (i.e. the sum of sulfate, nitrate and ammonium) for all regions considered, in either perturbation direction. Table 4 shows regional statistical validation metrics (normalized mean bias NMB [%], mean bias MB [µg m$^{-3}$], and correlation coefficient, definitions are given in the Table Notes) for the grid-to-grid comparison between TM5-FASST and TM5-CTM of the response to the [-80%, 100%] perturbation simulations (with Europe presented as a single region). In terms of NMB, the FASST linearisation performs worst for the NO$_x$ perturbations, with almost a factor 2 overestimate in Japan for an emission doubling. However, because of the already low NO$_x$ emissions in this region, the absolute error (MB) remains below 0.2µg m$^{-3}$. In all considered perturbation cases, FASST shows a positive MB, except for the NO$_x$ perturbation in India. In general, the highest NMB are observed for the regions where secondary PM$_{2.5}$ shows low response sensitivity to the applied perturbations and where the impact on the total PM$_{2.5}$ is therefore relatively low. Indeed, when considering the total resulting secondary PM$_{2.5}$ (i.e. the full right-hand side of Eq. 2, including the PM$_{2.5}$ base-concentration term containing primary and secondary components), regional averaged FASST secondary PM$_{2.5}$ values stay within 15% of TM5 (see Table S7.1of the SI). A break-down for the individual receptor regions within the European

zoom region of the linearisation error on the resulting total secondary PM$_{2.5}$ from individual and combined precursor perturbations is shown in Fig. S7.3 of the SI.
* * *
42) Section 3.1.2, P17, Lines 18 to 19 – Can you include references to back up the fact that combined NOx and NMVOCs emission perturbations will behave more linearly?
* * *
REPLY: We do not exactly say that combined and aligned NOx-VOC emission changes (in general) are behaving linearly, but, seen the fact that the ratio NOx/NMVOC determines the O$_3$ formation regime, combined emission changes of the same relative size and sign (in the way we applied them e.g. to establish the combined -20% perturbation responses) will not change the emission ratio and therefore preserve the O3 formation regime implying a linear behaviour. This is an implication of the statement made in the first phrase where we provide references.

CHANGES TO MANUSCRIPT*: we adapted the phrase as follows: (P20 L5)*

Because the NO$_x$/NMVOC ratio determines the O$_3$ response to emission changes, a perturbation with simultaneous NO$_x$ and NMVOC emission changes of the same relative size is expected to behave more linearly than single perturbations since the chemical regime remains similar.
* * *
43) Section 3.1.2, P17, Line 31 – remove 'also here'
* * *
REPLY:  done
* * *
44) Section 3.1.2, P17, Line 31 to 32 – Good agreement is found everywhere apart from China, Why?
* * *
REPLY: We presume the reviewer is referring to Fig. 5. Indeed for China the agreement between combined and sum of individual responses is – in absolute terms – slightly worse than most other regions, but in relative terms the sum of perturbations is within 10% of the combined one. We have added a scatterplot to Figure 5 to illustrate the over-all validity of additivity.
The underlying reason for the small deviations between combined and sum-of-individual responses has not been investigated in detail but, as stated above, is most probably linked to the fact that changing a single precursor emission strength changes the NOx/NMVOC ratio and could affect the O3 emission response regime.

CHANGES TO MANUSCRIPT: *P20 L19*

As shown in Fig. 5, for the -20% perturbations we find good agreement between the combined (NO$_x$ + NMVOC) perturbation (open circles) with the sum of the individual precursor perturbation (black dots). This occurs even in situations where titration by NO causes a reverse response in O$_3$ concentration as is the case in most of Europe and the USA, indicating that a -20% perturbation in individual precursors appears not to change the prevailing O3 regime.

*We also added a scatter plot to Fig. 5 to demonstrate the very good correspondence.*

[Figure]

Figure 5: TM5-CTM response in annual mean population-weighted $O_3$ concentration (in ppbV) upon emitted precursor perturbation of -20% for selected source receptor regions. European regions were perturbed simultaneously. Red bar: response form NMVOC–only perturbation (simulation P4); blue bar: response form $NO_x$-only perturbation (simulation P3). Open circles: response from simultaneous (NMVOC + $NO_x$) perturbation (simulation P5). Black dots: sum of individual responses. Shaded regions are perturbed simultaneously as one European region. Right panel: scatter plot between $O_3$ response to combined and summed individual responses.
* * *
45) Section 3.1.2, P18, Line 2 – change 'Europa' to Europe

REPLY: done.
* * *
46) Section 3.1.2, P18, Lines 16 to 18 – If anything I would say FASST overestimates the change in TM5 (be it positive or negative) most of the time as the -80% points on the scatter plot tend to always above the 1:1 line (see major point 6 above).

REPLY: For a negative emission change, an origin-forced response slope below 1 (with points lying above the 1:1 line) indicates that the response between unperturbed and perturbed in FASST is lower than TM5, hence FASST underestimates the response upon an emission decrease and consequently overestimates the resulting concentration which is the sum of base and perturbation response (Eq. 2). A response slope larger than one for a positive emission change also corresponds to an over-prediction of the total concentration. We describe now more clearly in section 3.1 that we are evaluating the perturbation response (the change) and how an under/overestimation affects the total resulting concentration.

CHANGES TO MANUSCRIPT: *most of the section has been rewritten as follows: (P21 L6)*

Figure 7 illustrates the performance of the TM5-FASST approach versus TM5 for regional-mean annual mean ozone, health exposure metric 6mDMA1 (both evaluated as population-weighted mean), and for the crop-relevant exposure metrics AOT40 and M12 (both evaluated as area-weighted mean) over the extended emission perturbation range. In most cases the response (i.e. the *change* between base and perturbed case) to emission perturbations lies above the 1:1 line across the 4 metrics, indicating that FASST tends to over-predict the resulting metric (as a sum

of base concentration and perturbation). Of the four presented metrics, AOT40 is clearly the least robust one, which can be expected for a threshold-based metric that has been linearized. Tables 5 to 7 give the statistical metrics for the grid-to-grid comparison of the perturbation term between FASST and TM5 for the health exposure metric 6mDMA1, and crop exposure metrics AOT40 and M12 respectively. Statistical metrics for the total absolute concentrations (base concentration + perturbation term) are given in Tables S7.2 to S7.4 in the SI. As anticipated, the $NO_x$-only perturbation terms are showing the highest deviation, in particular for a doubling of emissions, however combined $NO_x$-NMVOC perturbations are reproduced fairly well for all regions, staying within 33% for a -80% perturbation for all 3 exposure metrics, and within 38% for an emission doubling for 6mDMA1 and M12, while the AOT40 metric is overestimated by 76 to 126% for emission doubling. The total resulting concentration over the entire perturbation range for single and combined $NO_x$ and NMVOC perturbation agrees within 5% for 6mDMA1 and M12, and within 64% for AOT40. The mean bias is positive for both perturbations, for all metrics and over all analysed regions, except for crop metric M12 under a doubling of NMVOC emissions over Europe showing a small negative bias. The deviations for individual European receptor regions under single and combined NMVOC and $NO_x$ perturbations for health and crop exposure metrics are shown in Figs. S7.4 to S7.6 of the SI.
* * *
47) Section 3.1.2, P18, Lines 21 to 22 – I am not sure that the linear fit is that good for the change in annual mean O3 in Figure 7a as there seems to be distinctive curvature in the +100% simulation for larger O3 reductions. I anticipate that this will be larger for certain months. The non-linear behaviour seems to occur to a lesser extent for other O3 metrics where a linear approximation is probably more justified. I think a change of wording for this statement is required to reflect the fact that a linear approximation does not represent the non-linear chemistry effects for large emission perturbation.
* * *
REPLY: The linear fits in Figure 7 were used as a guide to evaluate the overall correspondence of regional mean O3 metrics versus TM5, they are not the linear approximations used in FASST. (Each dot is obtained applying the region-specific SR coefficients for the respective precursors). Because this seems to cause confusion with the reader, we omitted the fittings and present the figure now only with the 1:1 line as a reference. Our statement refers to the observation that – except for AOT40 – the regional mean ozone metrics are relatively well represented by FASST (i.e. close to the 1:1 line) and in particular the FASST approximation reproduces the negative response to emission doubling (and positive response to emission reduction), typical for the titration regime.

CHANGES TO MANUSCRIPT: *We deleted the section using the slopes of the linear fits in Figs. 4 and 7– see also changes mentioned in previous comment*
* * *
48) Section 3.1.2, P18, Lines 25 to 28 – Check percentage numbers are correct as they don't appear to be the same as that shown on Figure S7.4 or in Table 3 e.g. -5 to 13% for M12 where on the Figure S7.4c I can't see anything below 0
49) Section 3.1.2, P18, Lines 28 to 30 – Same as above but for NMVOC
50) Section 3.1.2, P19, Line 1 – Same as above but for combined emission perturbation.
* * *
REPLY TO 47- 49:
The inconsistencies between values in the text and the figures were a consequence of a different statistical evaluation method, more in particular: the text vales were referring to the mean of all individual grid cell relative deviations, whereas the graphs were referring to the NMB as defined above (major comment 6). We report the values now consistently as NMB in text and figures.

> 51) Section 3.2, P19, Line 10 – remove 'e'

REPLY:  done

> 52) Section 3.2, P19, Line 25 to 26 –In both scenarios emissions can change by >80% over some regions and precursors. The ability of FASST to predict such changes over regions needs to be highlighted in the results based on the breakdown of the linear approach for O3 at such high emission perturbations.

REPLY: Indeed a valid suggestion.

CHANGES TO MANUSCRIPT:
- *First of all, we became aware that the numbers reported in Table S8 (emission % changes relative to FASST reference) were wrong for all regions except Asia and Global – they have now been corrected (this does not affect the reported results)*
- *The introductory part of section 3.2 has been rewritten/rearranged mentioning some features of the scenario emissions, pointing to possible issues with combined emission changes that could not be addressed in the dedicated additivity/linearity simulations*
- *We have added new Figures, demonstrating that FASST does capture regional features both for low and high emission scenarios*

*We modified the relevant paragraph to:*

[revised manuscript text omitted]
 than absolute concentrations, and FASST is a tool for the assessment of policy measures, then why is the difference not shown in place of absolute concentrations? Might be worth showing the change in concentrations in the main text and the absolute concentrations in the supplementary. Also it might be better to show the change between FLE and BASE, and MIT and BASE separately rather than the different between the two future scenarios

REPLY: We agree with the comment that from policy relevance perspective, putting more emphasis on the deltas makes sense. However, using in TM5FASST the RCP reference year 2000 as a common reference scenario is not very useful as here we are looking at a different scenario family (GEA) and a different year (2030). From policy perspective, comparing a 'policy' case (here: MIT2030) with a 'non-policy' case (here: FLE2030) for a given year immediately reveals the benefits of policy action. We therefore prefer to present the delta between the two GEA scenarios (with the additional benefit that this reduces the number of figures when showing the delta).

CHANGES TO MANUSCRIPT
*As mentioned in the reply to the previous comment, we have rewritten most of the section. We include and discuss now both delta and totals for the two scenarios.*
* * *
54) Section 3.2, P20, Line11 to 12 – I would say that FASST tends to underestimate the magnitude of change in TM5 for both annual mean and M6M O3, as most points are below the 1:1 line. (see major point 6 above).

REPLY: The referee made a correct observation; the slope is misleading here. This has been addressed with the changes made in text and the new figures (see previous comments)
* * *
55) Section 3.2, P20, Lines 15 to 21 – Only a very small discussion on the future evaluation of health metrics. Maybe expand slightly to include different regions and that FASST always overpredicts compared to TM5

REPLY: Agree, we have expanded the discussion of the intercomparison of the health impacts and included as well the delta in mortalities as from policy perspective this is relevant.

CHANGES TO MANUSCRIPT:
- *Added additional panels c to Figs. 10 and 11 showing the delta mortalities for* $PM_{2.5}$ *and O3 respectively*
- *Modified the health impact discussion as follows: (P23 L22)*

A major issue in air pollution or policy intervention impact assessments is the impact on human health; therefore we also evaluate the TM5-FASST outcome on air pollution premature mortalities with the TM5-based outcome, applying the same methodology on both TM5 and FASST outcomes. We evaluate mortalities from $PM_{2.5}$ using the IER functions (Burnett et al., 2014) and $O_3$ mortalities using the log-linear ER functions and RR's from Jerrett et al. (2009) respectively. Figure 12 ($PM_{2.5}$) and Fig. 13 ($O_3$) illustrate how FASST-computed mortalities compare to TM5, both as absolute numbers for each scenario, as well as the delta (i.e. the health benefit for MIT2030 relative to FLE2030). Regional differences in premature mortality numbers are mainly driven by population numbers. In line with the findings for the exposure metrics ($PM_{2.5}$ and 6mDMA1) FASST in general over-predicts the absolute mortality numbers, in particular in the low-emission case. For MIT2030, global $PM_{2.5}$ mortalities are overestimated by 19%, in Europe and North-America FASST even by 43%. In the FLE2030 case, we find a better agreement, with a global mortality over-prediction of 3% (for Europe and North-America 5% and 11% respectively). For the latter scenario, the highest deviation is found in Latin America (10 – 20%). $O_3$ mortalities are overestimated globally by 11% (7%) with regional agreement within 20% (14%) for MIT2030 (FLE2030). However, as shown by the error bars, the difference between FASST and TM5 is smaller than the uncertainty on the mortalities resulting from the uncertainty on RR's only. The potential health benefit of the mitigation versus the non-mitigation scenario (calculated as FLE2030 minus MIT2030 mortalities) is shown in Figs. 12c and 13c. Globally, FASST underestimates the reduction in global $PM_{2.5}$ mortalities by 17% with regional deviations ranging between -30% for Europe and North-America, and -12% for India. The global health benefit for ozone is underestimate by 2% for $O_3$, however as a net result of 11% overestimation in India and 12 to 59% underestimation in the other regions. The numbers corresponding to Figs. 12 and 13 are provided in Table S8.4 and S8.5 of the SI.
The error ranges presented here are obviously linked to the choice of the test scenarios and will for any particular scenario depend on the magnitude and the relative sign of the emission changes relative to RCP2000, but given the amplitude of the emission change for the currently two selected scenarios relative to RCP2000, these results support the usefulness of TM5-FASST as a tool for quick scenario screening.
* * *
56) Section 3.3.2, P22, Line 1 – relate discussion of text to labels on Figure 13 or define labels with more description in Figure 13 caption
* * *
REPLY: We presume this refers in particular to the labels in the b-panel (SLS M-O3 etc...). We have added the explanation in the figure caption.

CHANGES TO MANUSCRIPT*: changed the relevant section to*: (P24 L2)

Figure 15b shows the break-down by forcing component, including the direct contributions by aerosols, by short-lived precursors to $O_3$ (SLS S-$O_3$), their indirect effect on $CH_4$ (SLS I-$CH_4$) and associated long-term $O_3$ (SLS M-$O_3$), as well as $CH_4$ forcing from direct $CH_4$ emissions and its associated feedback on background ozone (CH3 $O_3$). Fig. 15a separates the contributions by emission sector..

*And similar in the caption of Fig. 13 modified to:*

Figure 13: Year 2000 radiative forcing from Unger et al. (2010), based on EDGAR year 2000 emissions and from TM5-FASST applied to RCP year 2000 (a) break-down by sector and by forcing component. Biomass burning includes both large scale fires and savannah burning; (b) total over all sectors. SLS S-$O_3$: direct contribution of short-lived species (SLS) to $O_3$; SLS I-$CH_4$: indirect contribution from SLS to $CH_4$; SLS M-$O_3$: indirect feedback from SLS on background ozone via the $CH_4$ feedback. $CH_4$ $O_3$: feedback of emitted $CH_4$ on background $O_3$
* * *
57) Section 3.3.3, P22, Line 13 – replace 'were' with 'where' and remove 'and'.

REPLY:  done
* * *
58) Section 3.3.4, P22, Line 22 – remove 'implemented in FASST

REPLY:  done
* * *
59) Section 3.3.4, P22, Line 24 – replace 'death cause' with 'a cause of death'.

REPLY: done
* * *
60) Section 3.3.4, P22, Line 25 to 27 – Could the difference in population and mortality rates between the two studies lead to some of the differences in Figure 14?

REPLY: This is unlikely for Figure 14 (now Figure 16) as it shows concentration changes, not mortalities. If the referee intends to refer to Fig. 15 (now 17), we mention in the text that Silva et al. use indeed different population and base mortality projections. In particular – as mentioned - the projection for respiratory base mortality rates (which is relevant only for the O3 health impact and not for PM2.5) for 2050 in Silva et al. is very different from the values used in FASST (where they are constant compared to 2030 base mortality rates). The discrete dots in the O3 mortality graph are a simple attempt to demonstrate the impact in FASST of using these different mortality rates.
* * *
61) Section 3.3.4, P23, Line 9 to 11 – How can FASST account for inter-model variability in its results? I think that this is mentioned as a future development so needs to be linked to that here.

REPLY:   We intend to say that the difference between FASST and the ACCMIP model ensemble for what concerns O3, is probably not due to a poor performance of FASST (which is a fairly good approximation of TM5) but rather a consequence of generally occurring differences between models.

CHANGES TO MANUSCRIPT: *modified the phrase as follows: (P28 L7)*

The ozone exposure metric 6mDMA1 falls within the range of the ACCMIP model ensemble for 2030 - 2050, but the slope between 2030 and 2050 is lower than for the ACCMIP ensemble mean, i.e. FASST shows a much lower response sensitivity for $O_3$ to changing emissions between 2030 and 2050 than the ACCMIP models (-1ppb from 2030 to 2050 in FASST, versus -3ppb for the ACCMIP mean). Given our previous observation that FASST reproduces TM5 relatively well, this indicates that inter-model variability is a stronger factor in the model uncertainty than the reduced-form approach.
* * *
62) Section 3.3.4, P23, Line 17 – replace 'While also' with 'Whilst calculated'

REPLY: It seems the use of "while" or "whilst" is interchangeable in English language. As non-native English speaker it feels more comfortable to use "while".
* * *
63) Section 3.3.4, P23, Line 18 to 21 – Why does the different baseline mortality and population statistics have such a big impact on O3 mortality rates but not PM2.5?

REPLY: The reason is that respiratory mortality is not considered a cause of death from PM2.5; the GBD methodology includes COPD, LC, IHD and Stroke for $PM_{2.5}$ and respiratory disease for O3.

CHANGES TO MANUSCRIPT: *added the following phrase: (P28 L25)*

Respiratory mortality is not considered as a cause of death for PM2.5, which explains why a similar disagreement is not observed in the $PM_{2.5}$ mortality trend in Fig. 17b.
* * *
64) Section 3.3.4, P23, Line 27 to 31 – Could a little more discussion on regional mortality burdens be put into the main text. Interesting differences between regions and for RCP2.6 vs RCP8.5.

REPLY: Although the scope of this paper is not to make a scenario analysis or assess trends and impacts across regions, but rather to validate the FASST model, we agree that some more discussion is useful.

CHANGES TO MANUSCRIPT: *paragraph has been rewritten as follows: (P28 L28)*

A regional break-down of mortality burden from $PM_{2.5}$ in 2030 and 2050, relative to exposure to year 2000 concentrations, for major world regions and for the globe is shown in Figures S9.1 and S9.2 of the SI. Compared to Fig. 17 which shows the global mortality trends as a combined effect of changing population, mortality rates and pollution level, here the effect of changing population and baseline mortality is eliminated by exposing the evaluated year's population to pollutant levels of the relevant year and to RCP year 2000 levels respectively, and calculating the change between the two resulting mortality numbers. FASST reproduces the over-all observed trends across the regions: we see substantial reductions in North America and Europe in 2030, while in East Asia significant improvements in air quality impacts are realized after 2030. For the India region, all scenarios project a worsening of the situation. The global trend is dominated by the changes in East Asia. The observed differences between FASST and ACCMIP ensemble are not insignificant and partly due to different mortality and population statistics in particular for the year 2050, still they are consistent with the findings in the previous section: FASST tends to overestimate absolute $PM_{2.5}$ concentrations for emission scenarios different from RCP2000, and consequently tends to under-predict the benefit of emission reductions, while over-predicting the impact of increasing emissions.
* * *
65) Section 4, P24, Line 17 to 18 – Make statement in this sentence less strong by inserting 'tend to' between 'metrics' and 'remain'.

REPLY: done
* * *
66) Section 4, P24, Lines 21 to 23 – I think the first two sentences could be re-written to simply specify that because the emissions and meteorology are fixed the source receptor matrices remain fixed. Also I think the work 'arbitrary' should be removed.

REPLY: we have rephrased the sentences

CHANGES TO MANUSCRIPT: *(P30 L22)*

Another issue for caution relates to the FASST analysis of emission scenarios with spatial distribution that differs from the FASST reference scenario (RCP year 2000). The definition of the source regions when establishing the SR matrices implicitly freezes the spatial distribution of pollutant emissions within each region, and therefore the reduced-form model cannot deal with intra-regional spatial shifts in emissions.

67) Section 4, P24, Lines 25 – remove repetitive statement of 'compared to the base simulation year 2000'.

REPLY:  done

68) Section 4, P24, Lines 27 – remove 'be

REPLY:  done

69) Section 4, P24, Lines 30 to 31 – reword sentence to 'It can be expected that errors will be larger for the newer generation scenarios with dynamic allocation of emission across countries and macro-regions'

REPLY: done

70) Section 4, P25, Lines 5 to 7 – Sectors are mentioned that can't be assessed but little has been mentioned about shipping and aviation which can be assessed and are specifically included as a source region in FASST. I think it is worth mentioning these source regions in this section

REPLY:   Thank you for bringing this up – indeed worth mentioning.

CHANGES TO MANUSCRIPT: *added in discussion P31 L7:*

This limitation however does not apply to international shipping and aviation for which specific SR matrices have been established.

71) Section 5, P25, Line 32 – removal of '....' at end of page

REPLY:  done

72) Section 5, P26, Line 6 – subscripts for O3 and $PM_{2.5}$ required

REPLY: done

73) Section 5, P26, Line 19 – Slightly more detail could be provided on how the HTAP2 modelling exercise will inform/improve TM5-FASST, especially as TM5 was not a model that participated in HTAP2.

REPLY: The FASST architecture makes it possible to include new or additional SR matrices, even when they have been obtained from different models and with different regional definitions. SR simulations are now available from various models participating in HTAP2, but the 'required' and 'desired' simulations have not been fully completed by all participating models, and gapfilling method has been proposed (Turnock et al., 2018). Therefore a tool like FASST which

could bring this knowledge in a common structure, synthesizing the available data in an ensemble approach and make it accessible and applicable for interested users, would create a great added value. In the context of the UNECE/CLRTAP TF HTAP such a tool is already under development.

CHANGES TO MANUSCRIPT: *added the paragraph P34 L22*

The FASST architecture allows for an implementation of new or additional SR matrices, for instance new HTAP2 model ensemble mean matrices, each one accompanied by an ensemble standard deviation matrix to include the model variability in the results. Efforts are now underway to create a new web-based and user-friendly HTAP-FASST version, operating under the same principles as TM5-FASST, but based on an up-to-date reference simulation and underlying meteorology, thus creating a link between the knowledge generated by the HTAP scientific community and interested policy-oriented users. Indeed, similar to how the development of TM5-FASST was built upon extending the HTAP1 experiments in a single model context, the regional definitions and sector definitions used in HTAP2 (Galmarini et al., 2017; Koffi et al., 2016) were largely synchronized with the TM5-FASST set-up, increasing the community's capacity for multi-model assessments of hemispheric pollution. It is intended that the lessons-learned are informing the HTAP2 exercise

74) Figure 14 – I find that the grey lines mask out the black lines in some instances and I think the Figure would look better if the grey lines could be made less bold or more transparent. Also I am not sure why there is a different number of grey lines on each part of the Figure. Did a different number of models submit results for each experiment?

REPLY: Indeed, in ACCMIP not all models participated in each experiment, hence the different numbers. We have modified the figure to make the black lines more visible, and added information to the legenda.

CHANGES TO MANUSCRIPT: *modified Fig. 14 (now 16) into*

[Figure]

Figure 16: Global population-weighted differences (scenario year minus year 2000) (a) in annual mean PM$_{2.5}$ concentrations and (b) in O$_3$ exposure metric 6mDMA1 for 3 RCP scenarios in each future year, from the ACCMIP model ensemble (Silva et al., 2016) (black symbols and lines) and TM5-FASST_v0 (red symbols and lines). FASST URB_INCR: including the urban increment correction. Grey symbols: results from individual ACCMIP models. Grey lines connect results from a single model. Not all models have provided data for all scenarios. ACCMIP error bars represent the range (min, max) across the ACCMIP ensemble.

75) Table S3 – Certain lines in the table seem to be missing any information. e.g. P5 Germany, P4 USA, P5 Japan.

REPLY: That's a correct observation, in fact in those cases the experiments were not performed.

CHANGES TO MANUSCRIPT: *we removed the irrelevant lines, added a prime to the Pi' to distinguish from the -20% perturbations and added a line for the additional P1' simulations that were performed as well.*

76) Figure S3.3 – Why has the sign been reversed? For a 20% reduction in CH4 you would expect a decrease in O3 concentrations but the figure shows positive changes. This seems confusing

REPLY: Apologies for the confusion. The SR response field were stored as a positive change to a positive perturbation (although the perturbation runs were performed as negative perturbations resulting in a negative response)..

CHANGES TO MANUSCRIPT*: The caption has now been modified to:*

Figure S3.3 Decrease in annual mean surface $O_3$ for a 20% decrease in year 2000 CH4 concentration, i.e. 1760 to 1408 ppb (TF-HTAP1 SR1-SR2 scenarios)
* * *
77) Section S4.1, Equation 4.4 – I am not sure I can follow how the INCR formulation was derived and why it includes the (fup)2 terms.

REPLY: we added one intermediate step in the calculation that explains how the quadratic terms in fup are obtained.

CHANGES TO MANUSCRIPT*: added*

The population-weighted concentration is calculated as

$$C_{BC,TM5}^{pop} = f_{up}C_{BC,URB} + (1 - f_{up})C_{BC,RUR} \qquad [4.3]$$
* * *
78) Figure S5 – hard to decipher the different lines on the graph. Cannot see red lines most of the time. Please make clearer

REPLY: we have decreased the size of the dots and increased the line width so it is better visible

CHANGES TO MANUSCRIPT: *figures modified in the folowing way:*

[revised manuscript text omitted]

Fiore, A. M., Dentener, F. J., Wild, O., Cuvelier, C., Schultz, M. G., Hess, P., Textor, C., Schulz, M., Doherty, R. M., Horowitz, L. W., MacKenzie, I. A., Sanderson, M. G., Shindell, D. T., Stevenson, D. S., Szopa, S., Van Dingenen, R., Zeng, G., Atherton, C., Bergmann, D., Bey, I., Carmichael, G., Collins, W. J., Duncan, B. N., Faluvegi, G., Folberth, G., Gauss, M., Gong, S., Hauglustaine, D., Holloway, T., Isaksen, I. S. A., Jacob, D. J., Jonson, J. E., Kaminski, J. W., Keating, T. J., Lupu, A., Marmer, E., Montanaro, V., Park, R. J., Pitari, G., Pringle, K. J., Pyle, J. A., Schroeder, S., Vivanco, M. G., Wind, P., Wojcik, G., Wu, S. and Zuber, A.: Multimodel estimates of intercontinental source-receptor relationships for ozone pollution, J. Geophys. Res. Atmospheres, 114(D4), doi:10.1029/2008JD010816, 2009.

Forouzanfar, M. H., Afshin, A., Alexander, L. T., Biryukov, S., Brauer, M., Cercy, K., Charlson, F. J., Cohen, A. J., Dandona, L., Estep, K., Ferrari, A. J., Frostad, J. J., Fullman, N., Godwin, W. W., Griswold, M., Hay, S. I., Kyu, H. H., Larson, H. J., Lim, S. S., Liu, P. Y., Lopez, A. D., Lozano, R., Marczak, L., Mokdad, A. H., Moradi-Lakeh, M., Naghavi, M., Reitsma, M. B., Roth, G. A., Sur, P. J., Vos, T., Wagner, J. A., Wang, H., Zhao, Y., Zhou, M., Barber, R. M., Bell, B., Blore, J. D., Casey, D. C., Coates, M. M., Cooperrider, K., Cornaby, L., Dicker, D., Erskine, H. E., Fleming, T., Foreman, K., Gakidou, E., Haagsma, J. A., Johnson, C. O., Kemmer, L., Ku, T., Leung, J., Masiye, F., Millear, A., Mirarefin, M., Misganaw, A., Mullany, E., Mumford, J. E., Ng, M., Olsen, H., Rao, P., Reinig, N., Roman, Y., Sandar, L., Santomauro, D. F., Slepak, E. L., Sorensen, R. J. D., Thomas, B. A., Vollset, S. E., Whiteford, H. A., Zipkin, B., Murray, C. J. L., Mock, C. N., Anderson, B. O., Futran, N. D., Anderson, H. R., Bhutta, Z. A., Nisar, M. I., Akseer, N., Krueger, H., Gotay, C. C., Kissoon, N., Kopec, J. A., Pourmalek, F., Burnett, R., Abajobir, A. A., Knibbs, L. D., Veerman, J. L., Lalloo, R., Scott, J. G., Alam, N. K. M., Gouda, H. N., Guo, Y., McGrath, J. J., Charlson, F. J., Erskine, H. E., Jeemon, P., Dandona, R., Goenka, S., Kumar, G. A., et al.: Global, regional, and national comparative risk assessment of 79 behavioural, environmental and occupational, and metabolic risks or clusters of risks, 1990–2015: a systematic analysis for the Global Burden of Disease Study 2015, The Lancet, 388(10053), 1659–1724, doi:10.1016/S0140-6736(16)31679-8, 2016.

Galmarini, S., Koffi, B., Solazzo, E., Keating, T., Hogrefe, C., Schulz, M., Benedictow, A., Jurgen, G., Janssens-Maenhout, G., Carmichael, G., Fu, J. and Dentener, F.: Technical note: Coordination and harmonization of the multi-scale, multi-model activities HTAP2, AQMEII3, and MICS-Asia3: Simulations, emission inventories, boundary conditions, and model output formats, Atmospheric Chem. Phys., 17(2), 1543–1555, doi:10.5194/acp-17-1543-2017, 2017.

IIASA and FAO: Global Agro-Ecological Zones V3.0, [online] Available from: http://www.gaez.iiasa.ac.at/ (Accessed 11 November 2016), 2012.

Jerrett, M., Burnett, R. T., Arden, P. I., Ito, K., Thurston, G., Krewski, D., Shi, Y., Calle, E. and Thun, M.: Long-term ozone exposure and mortality, N. Engl. J. Med., 360(11), 1085–1095, doi:10.1056/NEJMoa0803894, 2009.

Kanakidou, M., Seinfeld, J. H., Pandis, S. N., Barnes, I., Dentener, F. J., Facchini, M. C., Dingenen, R. V., Ervens, B., Nenes, A., Nielsen, C. J., Swietlicki, E., Putaud, J. P., Balkanski, Y., Fuzzi, S., Horth, J., Moortgat, G. K., Winterhalter, R., Myhre, C. E. L., Tsigaridis, K., Vignati, E., Stephanou, E. G. and Wilson, J.: Organic aerosol and global climate modelling: a review, Atmospheric Chem. Phys., 5(4), 1053–1123, doi:10.5194/acp-5-1053-2005, 2005.

Kirschke, S., Bousquet, P., Ciais, P., Saunois, M., Canadell, J. G., Dlugokencky, E. J., Bergamaschi, P., Bergmann, D., Blake, D. R., Bruhwiler, L., Cameron-Smith, P., Castaldi, S., Chevallier, F.,

Feng, L., Fraser, A., Heimann, M., Hodson, E. L., Houweling, S., Josse, B., Fraser, P. J., Krummel, P. B., Lamarque, J.-F., Langenfelds, R. L., Le Quéré, C., Naik, V., O'Doherty, S., Palmer, P. I., Pison, I., Plummer, D., Poulter, B., Prinn, R. G., Rigby, M., Ringeval, B., Santini, M., Schmidt, M., Shindell, D. T., Simpson, I. J., Spahni, R., Steele, L. P., Strode, S. A., Sudo, K., Szopa, S., van der Werf, G. R., Voulgarakis, A., van Weele, M., Weiss, R. F., Williams, J. E. and Zeng, G.: Three decades of global methane sources and sinks, Nat. Geosci., 6(10), 813–823, doi:10.1038/ngeo1955, 2013.

[revised manuscript text omitted]

---

## Author Comment (AC2) · 6 Sep 2018

We thank reviewer 2 for the insightful comments, and for pointing to inconsistencies. We apologize for needing more time than anticipated to address all comments, but we believe that we have been able to address most issues, and that we have significantly strengthened the manuscript.

Before addressing the comments we would like to mention that we have modified the abbreviation of the $O_3$ health exposure metric (6 monthly daily maximum 1-h concentration) from M6M to 6mDMA1 (and accordingly M3M to 3mDMA1) as the latter seems to be commonly used in other works.

In the following we have placed the numbered reviewer comments in boxes. Our reply to the reviewer is in blue font, the changes to the manuscript in red font.

We also attach a revised version of manuscript and supplement with tracked changes compared to the first version.

REVIEWER 2 comments:

The manuscript by Van Dingenen et al. presents and evaluates TM5-FASST, a reduced form air quality assessment tool. The manuscript is long, but does a thorough job of both presenting the computational methods used for formulating the FASST tool and the types of impacts that it calculates (from health impacts to climate) as well as evaluating the tool against simulations from the full TM5 model as well as results in the literature. I have some additional questions in a few areas, described below, but in general was satisfied / impressed with the evaluation and performance. The writing could use a bit more editing for grammar and some of the figures need clarification on units, axis, etc. Addressing these will amount to moderate revisions and some additional evaluations.

> 1) However my only main concern about would be if this article should be moved to GMD instead of ACP, as the emphasis really is on the tool development and evaluation; there is not any content on application of the tool to new science or policy questions. It may not thus fit the scope of ACP.

REPLY:
We agree that this paper would have been also suited for GMD. However, due to high relevancy of this publication for the work of the TF HTAP, we decided that thematically the paper also fitted very well in this ACP special issue. Unfortunately, for this special issue, it was decided not to have a joint special issue between GMD and ACP (or other Copernicus journals), which would have been the perfect solution. We are confident that the interested reader will also find this publication in ACP.

> 2) 2.1-8: Why are reduced form or source-receptor models needed in the first place? I think there's a significant point to be made here about the complexity of air quality modeling vs the level of sophistication and computational intensity that can be acceptable to the decision-making community. But the article as presently writing misses this point, so the justification for the tool isn't readily apparent

REPLY:
Thank you for making this point, it is indeed important to introduce these issues to a general readership. This comment is closely related to the next one, so we address both in the introduction which has been expanded.

CHANGES TO MANUSCRIPT: *Changed first part of the introduction to: (P2 L14)*

**1. Introduction**

A host of policies influence the emissions to air. In principle any policy that influences the economy and use of resources will also impact emissions into the atmosphere. Specific air pollution policies aim to mitigate the negative environmental impacts of anthropogenic activities, some of which may be affected by other policies, like climate mitigation actions, transport modal shifts or agricultural policies. Further, air quality policies may impact outside their typical environmental target domains (human and ecosystem health, vegetation and building damage,...) for instance through the role played by short-lived pollutants in the Earth's radiation balance (Myhre et al., 2011; Shindell et al., 2009). Insight into the impacts of policies in a multi-disciplinary framework through a holistic approach could contribute to a more efficient and cost-effective implementation of control measures (e.g. Amann et al., 2011; Maione et al., 2016; Shindell et al., 2012).

Several global chemical transport models are available for the evaluation of air pollutants levels from emissions, sometimes in combination with off-line computed climate relevant metrics such as optical depth or instantaneous radiative forcing (e.g. Lamarque et al., 2013; Stevenson et al., 2013). These models provide detailed output, but are demanding in terms of computational and human resources for preparing input, running the model, and analyzing output. Further they often lack flexibility to evaluate ad-hoc a series of scenarios, or perform swift what-if analysis of policy options. Therefore there is a need for computationally-efficient methods and tools that provide an integrated environmental assessment of air quality and climate policies, which have a global dimension with sufficient regional detail, and evaluate different impact categories in an internally consistent way. Reduced-form source-receptor models are a useful concept in this context. They are typically constructed from pre-computed emission-concentration transfer matrices between pollutant source regions and receptor regions. These matrices emulate underlying meteorological and chemical atmospheric processes for a pre-defined set of meteorological and emission data, and have the advantage that concentration responses to emission changes are obtained by a simple matrix multiplication, avoiding expensive numerical computations. Reduced-form source-receptor models (SRM) are increasingly being used, not only to compute atmospheric concentrations (and related impacts) from changes in emissions but they have also proven to be very useful in cost optimization and cost-benefit analysis because of their low computational cost (Amann et al., 2011). Further, because of the detailed budget information embedded in the source-receptor matrices, they are applied for apportionment studies, as a complementary approach to other techniques such as adjoint models (Zhang et al., 2015) and chemical tagging (e.g. Grewe et al., 2012).

Although the computational efficiency of SRMs comes at a cost of accuracy, regional detail and flexibility in spatial arrangement of emissions, they have been successfully applied in regional studies (Foley et al., 2014; Li et al., 2014; Liu et al., 2017; Porter et al., 2017) and have demonstrated their key role in policy development (Amann et al., 2011).
* * *
3) Intro: Overall the introduction is rather brief. There are other reduced form models on regional scales that are used for different purposes (in the US and Asia, in particular). There are also theoretical advantages (quick) and disadvantages (approximations of linearity; enforced aggregation at pre-defined scales; outdated emissions inventories or old atmospheric conditions) of reduced form models. A lot more thought could be put into discussion and introducing these issues. This is ACP, not GMD, so more than just a model description is expected
* * *
REPLY: see previous comment

4) 3.18: This sentence is a bit too vague to be useful. The authors should mention what type of model updates have been made (emissions? aerosols? etc) and why they are deemed to not be relevant for this current work.

REPLY: We have added some more text to this paragraph to explain the major differences. The choice of emissions is not relevant in this context as emission datasets are external to the model framework, and in general chosen by the user depending on scientific issue.

CHANGES TO MANUSCRIPT: *Added to Section 2.1 (P4 L24)*

TM5 results used in the present study allow comparison with a range of other global model results in HTAP1, but ignore subsequent updates and improvements in TM5 as for instance described in Huijnen et al. (2010), which we consider not critical for this study. The most recent TM5 model does no longer consider zoom regions, but recoded the model into a Massive Parallel framework, enabling efficient execution on modern computers. While global horizontal resolution (1°x1°) is similar to the resolution of the most refined zoom region in TM5, vertical resolution was increased. Further, the model also uses vertical mass fluxes from the parent ECMWF meteorological model, not available at the time of development of TM5-cy2-ipcc, which could lead to somewhat different mixing characteristics. The gas phase chemical module has been updated to a modified version of CMB5.
.

5) This does raise the question of uncertainties introduced in this tool owing to use of a single year (was 2001 an average year, in terms of temp, precipitation, etc.?) to approximate a reasonable climatology, as well as this use of a year that is significantly older than most present applications, considering decadal-scale climate change.

REPLY: This is an issue raised by both reviewers. We agree with the reviewer that the year 2001 meteorology is somewhat outdated. The perturbation runs for constructing the SR library of FASST were performed with the TM5 model set-up defined in the first phase of HTAP1 (during the period 2008 – 2011)  and because of the computational costs, an update with more recent meteorology was not possible (TM5 is not taking part in HTAP2 where meteorological year 2010 has been used).  A systematic check of the representativeness of this particular year for each of the FASST regions is beyond the scope of this study, in the first place because FASST is considered to be a screening tool focussing on impacts of emission changes.  However we have substantially extended the discussion on the use of a single meteorological year.

CHANGES TO MANUSCRIPT:   *Added to Section 2.1 P5 L10*

Meteorological fields are obtained from the ECMWF operational forecast representative for the year 2001. The implications of using a single meteorological year will be discussed in section 4.2.

*Discussion section 4.2  added (P31 L25)*

**4.2 Inter-annual meteorological variability**

A justified critique on the methodology applied to construct the FASST SRs relates to the use of a single and fixed meteorological year 2001, implying possible unspecified biases in pollutant concentrations and source-receptor matrices compared to using a 'typical meteorological/climatological year'. We followed the choice of the meteorological year 2001 made for the HTAP1 exercise. As the North-Atlantic Oscillation (NAO) is an important mode of the inter-annual variability in pollutant concentrations and long range transport (Christoudias

et al., 2012; Li et al., 2002; Pausata et al., 2013; Pope et al., 2018), the HTAP1 expectation was that this year was not an exceptional year for long-rang pollutant transport - e.g. for the North-Atlantic region, as indicated by a North Atlantic Oscillation (NAO) index close to zero for that year (https://www.ncdc.noaa.gov/teleconnections/nao/). The HTAP1 report (Dentener et al., 2010) also suggested that "Inter-annual differences in SR relationships for surface $O_3$ due to year-to-year meteorological variations are small when evaluated over continental-scale regions. However, these differences may be greater when considering smaller receptor regions or when variations in natural emissions are accounted for". The role of spatial and temporal meteorological variability can thus be reduced by aggregating resulting pollutant levels and impacts as regional and annual averages or aggregates, the approach taken in TM5-FASST. The impact of the choice of this specific year on the TM5-FASST model uncertainty or possible biases in base concentrations and SR coefficients is not easily quantified. For what concerns the pollutant base concentrations, some insights in the possible relevance of meteorological variability can found in the literature. For example, Anderson et al., (2007) showed that in Europe, the meteorological component in regional inter-annual variability of pollutant concentrations ranges between 3% and 11% for airborne pollutants ($O_3$, $PM_{2.5}$), and up to 20% for wet deposition. On a global scale, Liu et al. (2007) demonstrated that the inter-annual variability in PM concentrations, related to inter-annual meteorological variability can even be up to a factor of 3 in the tropics (e.g. over Indonesia) and in the storm track regions. A sample analysis (documented in section S2.2 of the SI) of the RCP year 2000 emission scenario with TM5 at 6°x4° resolution of 5 consecutive meteorological years 2001 to 2005 indicates a year-to-year variability on regional $PM_{2.5}$ within 10% (relative standard deviation) and within 3% for annual mean $O_3$. We find a similar variability on the magnitudes of 20% emission perturbation responses within the source region for 6 selected regions (India, China, Europe, Germany, USA and Japan). The relative share of source regions to the pollutant levels within a given receptor region shows a lower inter-annual variability (typically between 2 and 6% for $PM_{2.5}$) than the absolute contributions.
* * *
6) 4.29: There is extensive research on the chemical oxidation of elemental carbon and the role this plays on the lifetime of this species in the atmosphere. Comment on why this is not included.
* * *
REPLY: The referee is right, also primary pollutants can undergo chemical conversion – however we feel this comment relates rather to 6.15 where we state that in TM5 (and FASST) the lifetime of BC and POM is not changing. The statement in 4.29 was intended to point out the difference between primary and secondary pollutants where in the latter case a completely new chemical compound is formed from precursors via chemical reactions, while for primary pollutants, dispersion and deposition are the primary process affecting their atmospheric concentration. Since the development of TM5, in literature two approaches have been developed towards parameterizing 'ageing' of elemental carbon. Ageing through condensation of hydrophobic species such as SO4 (and in the real world also other soluble components) is considered in e.g. the HAM aerosol physics model (Stier et al., 2005). The second approach considers oxidation of carbonaceous aerosol by $O_3$ following Tsigaridis and Kanakidou (2003). More recent work (e.g. Huang et al., 2012) analyses the joint impact of the two approaches, explicitly including the chemical-physical ageing processes. In general including the explicit processes tends to lengthen the atmospheric residence time of EC/BC compared to the earlier simple parameterisation in CTMs. The reason of not including these processes at the time of the release of TM5-JRC-Cy2-IPCC was that at that time none of the approaches was robustly anchored in improved performance at multiple observational sites, while at the same time the uncertainties in the wet removal parameterization were (and still are) also highly uncertain.

CHANGES TO MANUSCRIPT*: We feel this comment addresses original 6.15 rather than 4.29*

*Original 6.15*

BC and POM emissions are assumed not to interact with other pollutants and their atmospheric lifetime are assumed not to 15 be affected by mixing with other soluble species like sulfate, nitrate or ammonium salts

*modified to (P8 L22):*
BC and POM are assumed not to interact with other pollutants and their atmospheric lifetimes are prescribed and assumed neither to be affected by mixing with other soluble species like sulfate, nitrate or ammonium salts, nor to undergo oxidation by $O_3$. Recent work (e.g. Huang et al., 2012) indicates that a parametrized approach, as applied in TM5, tends to underestimate BC and POM atmospheric lifetimes, leading to a low concentration bias. When explicitly modelled, including the combined impact of both mechanisms, Huang et al., 2012 find that the global atmospheric residence times of BC and POM are lengthened by 9% and 3% respectively.

7) 5.1: I'm not sure what is a "parameter" in this context – please explain.

REPLY: Thank you for pointing out, this is a typo that has been corrected

CHANGES TO MANUSCRIPT: replaced 'parameters' by 'pollutants'

8) 5.1 - 14: It seems like some discussion of the fact that this functional relationship is only approximate is warranted. Instead, it is presented here as if the actual functional relationship is known, where in fact just a local linear approximation is used. This must have some limitations. For example, what is the basis for the statement later on this page that - 20% perturbation is small enough to evaluate sensitivities and large enough for extrapolation? I recognize that -20% is a commonly used modeling experiment, but it is also commonly known that this approach has limitations for source attribution that are well documented in the literature (compared to tagging, 2nd order methods, or other).

REPLY: The referee makes a good point here, obviously the linear approach is approximate and has both advantages and limitations. We have already addressed most of this discussion in the introduction. We also modified the relevant phrase in the text.

CHANGES TO MANUSCRIPT:

*Replaced:*
In the current version v0 of TM5-FASST the function is a linear relation expressing the change in pollutant concentration in the receptor region upon a change in precursor emissions in the source region...

*By (P6 L27):*
In the current version v0 of TM5-FASST the emission-concentration relationship is locally approximated by a linear function expressing the change in pollutant concentration in the receptor region upon a change in precursor emissions in the source region...

9) The notation in equations (1) and (2) is not correct. In equation 1, there is an inconsistency between the description of the notation for the concentrations vs emissions species (i and j) and what is written in the equation. Assuming the equation is correct, the text should refer to change in concentration of component j (not i) owing to emitted precursor i (not j).

REPLY: Indeed, thanks for spotting.

CHANGES TO MANUSCRIPT:
*Replaced:*

For each receptor point y (i.e. each model vertical level 1°x1° grid cell), the change in concentration of component i in receptor y resulting from a -20% perturbation of emitted precursor j in source region x, …

*By (P7 L29):*
For each receptor point y (i.e. each model vertical level 1°x1° grid cell), the change in concentration of component j in receptor y resulting from a -20% perturbation of emitted precursor i in source region x, …

10) In equation (2), the notation on the summations is not complete nor correct. The first sum should be from x = 1 (below the sum) to n_x (written above the sum), and the second should be for i = 1 (below the sum) to n_i(j) (above the sum). It's also not clear why y would be bold in this equation. As explained in the text, the number of precursor pollutants (n_i) depends on the pollutant response in consideration, hence n_i is n_i(j). So the pollutant responses are dry aerosol concentrations? At what T,P,RH?

REPLY:
Indexing has been corrected and bold face removed.
The stored SR matrices for each component are indeed the dry mass, as obtained from the TM5 model lower layer (or as column density for radiative properties), using the meteorological data for year 2001.
For comparison with measurements and for health impact assessment FASST provides an estimate of $PM_{2.5}$ residual H2O at 35% RH and 25°C using mass growth factors for ammonium salts of 1.27 (Tang, 1996) and sea-salt of 1.15 (Ming and Russell, 2001). This allows for a calculation of $PM_{2.5}$ mass simulating the protocol for determination of gravimetric $PM_{2.5}$ mass in monitoring networks, and these are also the values on which epidemiological studies are based. Radiative forcing obviously takes into account atmospheric RH conditions.

CHANGES TO MANUSCRIPT:
*Added below Eq. (1) (P8 L1):*
In the present version TM5-FASST_v0, the SR coefficients for pollutant concentrations are stored as annual mean responses to annual emission changes. Individual $PM_{2.5}$ components SRs are stored as dry mass ($\mu g\ m^{-3}$). $PM_{2.5}$ residual water at 35% is optionally calculated a posteriori for sensitivity studies, assuming mass growth factors for ammonium salts of 1.27 (Tang, 1996) and for sea-salt of 1.15 (Ming and Russell, 2001). The presence of residual water in $PM_{2.5}$ is not irrelevant: epidemiological studies establishing $PM_{2.5}$ exposure-response functions are commonly based on monitoring data of gravimetrically determined $PM_{2.5}$, for which measurement protocols foresee filter conditioning at 30 – 50% RH. As many health impact modelling studies consider dry $PM_{2.5}$ mass or do not provide information on the inclusion of residual water we use dry $PM_{2.5}$ for health impact assessment in this study for consistency, unless mentioned differently.

*Correcting indexing (P8 L11):*
The total concentration of component (or metric) *j* in receptor region *y*, resulting from arbitrary emissions of *all $n_i$ precursors i* at *all $n_x$ source regions x,* is obtained as a perturbation on the base-simulation concentration, by summing up all the respective SR coefficients scaled with the actual emission perturbation:

$$C_j(y) = C_{j,base}(y) + \sum_{k=1}^{n_x} \sum_{i=1}^{n_i} A_{ij}[x_k, y] \cdot \left[ E_i(x_k) - E_{i,base}(x_k) \right] \qquad (2)$$

11) 6.21: This equation needs to be corrected following suggestions for equation (2).

REPLY: Correct, done.

12) 6.23: It is oxymoronic to refer to secondary biogenic POM. This would just be secondary biogenic OM.

REPLY: Apologies for the confusion, but in this case POM actually stands for particulate organic matter.

CHANGES TO MANUSCRIPT: *changed P4 L16 to* "Biogenic secondary organic aerosol (BSOA)"

13) 6.24: Can the authors comment on how neglect of anthropogenic SOA might be biasing the results of this tool?

REPLY:
This is a difficult question, which may be worthy of an entire review. The main reason for ignoring anthropogenic SOA at the time of development of TM5- cy2-ipcc was that in the version of the CMB4 chemical scheme implemented in the model, Benzene and toluene chemistry was not included, as it was considered of local importance. In addition reliable global inventories were not available. Having said this, the importance of anthropogenic SOA will strongly depend on local emission strength and atmospheric chemistry conditions. For instance a recent conducted in China (Hu et al., 2017) suggest that in summer biogenic SOA is larger in summer (75 %) than in winter (25 %) 5  and over 35 $\mu g/m^3$ in 4 Chinese cities.
A global modelling study by (Farina et al., 2010) based on the volatility approach suggests that SOA formation from monoterpenes, sesquiterpenes, isoprene, and anthropogenic precursors is estimated as 17.2, 3.9, 6.5, and 1.6 Tg $yr^{-1}$, respectively. While in that study global levels of SOA were low (annual average 0.02 ug/m3)- in particular in Europe and China levels up to 1 ug/m3 were calculated, where levels of primary organic aerosol were reaching 20 ug/m3. Although this back-off the envelop assessment suggest that for larger regions the impact is less than 5-10 %, in urban regions with high anthropogenic VOC emissions the impact may be larger.

CHANGES TO MANUSCRIPT:
*We added following phrase to the discussion section 4.1 (P31 L17):*

The omission of secondary organic PM in TM5 is estimated to introduce a low bias in the base concentration of the order of 0.1 $\mu g\ m^{-3}$ as global mean however with regional levels in Central Europe and China up to 1 $\mu g\ m^{-3}$ in areas where levels of primary organic matter are reaching 20 $\mu g\ m^{-3}$ (Farina et al., 2010) indicating a relatively low contribution of SOA to total $PM_{2.5}$

14) 6.25: Just because the impacts are annual in nature doesn't mean the emissions contributions to the impacts are seasonally consistent. Surely the impact of $NO_x$ on ammonium nitrate and $O_3$ is quite different in different seasons; it's not clear why one would have access to this information but not use it.

REPLY: It is certainly true that there are seasonal differences in emission-concentration sensitivities. However, when relevant, these seasonal trends are implicitly included in the exposure metrics and impacts. Several metrics are in fact based on detailed temporal ozone trends, e.g. considering only the daily maximal hourly value, or hourly values exceeding a 40 ppb threshold during the crop growing season. These responses – seasonal in nature – are stored to be scaled with annual emissions. Health impacts from $PM_{2.5}$ are based on annual averaged values and are not evaluated on a seasonal basis.  Hence, although there may be scientific (process understanding) interest in elaborating seasonal trends, from a health/crop/climate impact assessment perspective, there is not much added value storing temporal trends in the source-receptor matrices which would come at a high computational cost (multiplying the number of SR matrices with 12).

15) 7.7-21: I got a bit lost in this discussion of the way CH4 concentrations responses are treated. It would be good if this section could be expanded and formalized a bit better, using equations where useful, such that the approach could be evaluated and replicated.

REPLY: This was indeed not explained in an optimal way. As there are two instances in the paper where CH4 responses are treated (O$_3$ response from CH4 emissions, and indirect forcing from short-lived precursors on CH4 and background ozone – see next comment) we have moved and expanded the description of the methodology, which is based on our interpretation of published work, in the SI (S3)

CHANGES TO MANUSCRIPT: added *section S3 in the SI:*

**S3.1 CH$_4$ – O$_3$ source-receptor relations from HTAP1 perturbation experiments:**

CH$_4$ emissions lead to a change in CH4 concentrations with a perturbation response time of about 12 years. In order to avoid expensive transient computations, HTAP1 simulations SR1 and SR2 with prescribed fixed CH4 concentrations (1760 ppb and 1408 ppb, see Dentener et al., 2010) were used to establish CH4 – O$_3$ response sensitivities.  Previous transient modeling studies have shown that a change in steady-state CH$_4$ abundance can be traced back to a sustained change in emissions, but the relation is not linear because an increase in CH$_4$ emissions removes an additional fraction of atmospheric OH (the major sink for CH$_4$) and prolongs the lifetime of CH$_4$ (Fiore et al., 2002, 2008; Prather et al., 2001).
In a steady-state situation, the CH4 concentration is the result of balanced sources and sinks. In the HTAP1 experiments, keeping all other emissions constant, the change in the amount of CH4 loss (mainly by OH oxidation with a lifetime of ca. 9 years, neglecting loss to soils and stratosphere with lifetimes of ca.160 and 120 years respectively (Prather et al., 2001) ) under the prescribed change in CH4 abundance should therefore be balanced by an equal and opposite source which we consider as an "effective  emission". The amount of CH4 oxidized by OH in one year being diagnosed by the model, the resulting difference between the reference and perturbation experiment of -77 Tg sets the balancing "effective" emission rate to 77Tg/yr, which is then used to normalize the resulting O$_3$ and O$_3$ metrics response to a CH4 emission change.
The same perturbation experiments also allow us to establish the CH4 self-feedback factor F describing the relation between a change in emission and the change in resulting steady-state concentration:

$$\frac{C_2}{C_1} = \left(\frac{E_2}{E_1}\right)^F \tag{S3.1}$$

With CH4 concentrations prescribed, CH4 emissions were not included in the SR1 and SR2 experiments. The feedback factor F is derived from model-diagnosed respective CH4 burdens (B) and total lifetimes (LT) as follows (Fiore et al., 2009; Wild and Prather, 2000):

$$F=1/(1-s) \tag{S3.2}$$

$$s = \partial \ln(LT) / \partial \ln(B) \tag{S3.3}$$

TM5 returns s =  0.33 which can be compared to a range of values between 0.25-and 0.31 in IPCC-TAR (Prather et al., 2001, Table 4.2) , resulting in a TM5-inherent calculated feedback factor F=1.5. This factor can be used to estimate the corresponding SR2-SR1 change in CH4 emission in a second way. From Eq. S3.1 we find that a 20% decrease in CH4 abundance corresponds to a 14% decrease in total CH4 emissions. Kirschke et al. (2013) estimate total CH4 emissions in the 2000s in the range 550 – 680 Tg yr-1 from  which we obtain an estimated

emission change between the HTAP SR1 and SR2 experiments in the range 77 – 95 Tg yr-1, in line with our steady-state loss-balancing approach.
* * *
16) It also wasn't clear to me – is $NO_x$ allowed to impact CH4, particularly for the purposes of climate impacts?
* * *
REPLY: It is, as are all short-lived ozone precursors. We have added an extensive description in the SI on how the emission – forcing contributions in terms of (1) direct $O_3$ (2) indirect CH4 and (3) CH4-induced long-term $O_3$

CHANGES TO MANUSCRIPT: *added section S6.2 to the SI*

**S6.2 Secondary forcing feedbacks of $O_3$ precursors on CH4 and background $O_3$**
Emissions of short-lived species ($NO_x$, NMVOC, CO, $SO_2$) influence the atmospheric OH burden and therefore the CH4 atmospheric lifetime, which in turn contributes to long-term change in CH4 and background ozone. Hence, the total forcing contribution from $O_3$ precursors consists of a short-term direct contribution from immediate $O_3$ formation (S-$O_3$), and secondary contributions from CH4 (I-CH4) and a long-term feedback from this CH4 on background $O_3$ (M-$O_3$). We apply the formulation by (Fiore et al., 2009; Prather et al., 2001; West et al., 2007) to calculate the secondary change in steady-state CH4 from SLS emissions, using the TM5 perturbation experiments for FASST (see section S3). TM5 diagnoses the CH4 loss by oxidation for reference and perturbation run (where the emissions of SLS are decreased with -20%), from which we calculate the CH4 oxidation lifetime ratio between reference and perturbation:

$$\frac{LT_P}{LT_{Ref}} = \frac{CH4\_ox_P}{CH4\_ox_{Ref}} \qquad\qquad [S6.5]$$

Where LT is the CH4 lifetime against loss by OH oxidation, and $CH4\_ox$ = the amount (Tg) of CH4 oxidized.
The new steady-state methane concentration M due to the changing lifetime from perturbation experiment P, induced by $O_3$ precursor emissions follows from (Fiore et al., 2008, 2009; Wild and Prather, 2000):
$M = M_0 \times \left(\frac{LT_P}{LT_{ref}}\right)^F$ where $M_0 = 1760$ ppb, the reference CH4 concentration and F = 1.5, determined from the HTAP1 CH4 perturbation experiments, as described in section S3.

The change in CH4 forcing (I-CH4) associated with the change to the new steady-state concentration is obtained from IPCC AR5 equations:

$\Delta F = \alpha\left(\sqrt{M} - \sqrt{M_0}\right) - \left(f(M, N_0) - f(M_0, N_0)\right)$    [S6.6]
$f(M, N) = 0.47ln[1 + 2.01 \times 10^{-5}(MN)^{0.75} + 5.31 \times 10^{-15}M(MN)^{1.52}]$    [S6.7]
Where M, M0 = CH4 concentration in ppb, N0 = N2O (=320 ppb)

The associated long-term $O_3$ forcing (M-$O_3$) per Tg precursor emitted is obtained by scaling linearly the change in $O_3$ forcing obtained in the HTAP1 CH4 perturbation simulation (SR2–SR1), with the change in CH4 obtained above, and normalizing by the precursor emission change (Fiore et al., 2009)

$$\Delta F = \frac{\Delta F_{O3}[SR2 - SR1]}{M_{SR2} - M_{SR1}}(M - M_0) \quad [S6.8]$$

The response of CH4 and $O_3$ forcing to CO emission changes (for which no regional TM5-FASST perturbation model simulations were performed) was taken from TM5-CTM simulations performed for the HTAP1 assessment (Dentener et al., 2010) using the average forcing

efficiency for North America, Europe, South-Asia and East-Asia. For regions not covered by the HTAP1 regions, the HTAP1 rest-of-the-world forcing efficiency was used.
The resulting region-to-globe emission-based forcing efficiencies are given in Tables S6.2 to S6.5 for aerosols, CO, CH4 and other $O_3$ precursors respectively.
* * *
17) Many of the studies in the table are a bit out-of-date, as they would be around atmospheric conditions / emissions levels that are rather old, or in comparison to datasets that have greatly matured (for example comparison to satellite-based NO2 retrievals, which are now much more accurate and consistent across retrievals than in the study of van Noije 2006.
* * *
REPLY: This is indeed the case. However, the TM5 version used in this study was developed and evaluated in the studies shown in the table. Since then no new developments and evaluation studies have been performed on the version used in this work. As in this study we are focusing on an evaluation of TM5-FASST, using TM5 as a reference, it is beyond the scope of this study to re-evaluate TM5 with new data sets, which would be worth one or more new papers on its own.

CHANGES TO MANUSCRIPT: *added following phrase to section S2.1 in the SI:*

We are aware of recent more accurate observational data have become available for the validation of the model since the validation studies listed in Table S2.1, in particular from satellite-based retrievals. However here we focus on the validation of FASST, using TM5 as a reference, and it is beyond the scope of this study to re-evaluate the TM5 model itself.
* * *
18) 8.3-4: Statements like this could be supported by reference many articles on the topic, including evaluation of how much this matters for different species at different scales.
* * *
REPLY:
We agree with this point. We address this now in a dedicated section in the discussion where we refer to exemplary studies that have specifically addressed the issue of grid resolution on exposure. They indicate in general that $O_3$ tends to be overestimated and (primary) $PM_{2.5}$ tends to be underestimated compared to higher resolution models. Further we have included in the SI a quick analysis of the TM5 base simulation at resolution 6°x4°, 3°x2° and 1°x1° to illustrate the impact of resolution on concentrations and emission-concentration response sensitivities.

CHANGES TO MANUSCRIPT:

*Modified last part of section 2.1 (P5 L28):*

The model grid resolution influences the predicted pollutant concentrations as well as the estimated population exposure, especially near urban areas where strong gradients occur in population density and pollutant levels, which cannot be resolved by the 1°x1° resolution. In section 2.4 we describe a methodology to improve population $PM_{2.5}$ exposure estimates by applying sub-grid concentration adjustments based on high-resolution ancillary data. The bias introduced by model resolution affects as well computed SR matrices, e.g. off-setting the share of 'local' versus 'imported' pollution in a given receptor region. We will discuss this aspect more in detail in section 4.3.

*Added section 4.3 to the discussion section (P32 L20):*

[revised manuscript text omitted]

sources than the residential and transport sector are assumed to occur more remotely from urban areas.
* * *
20) Section 2.4 and SIS4 are useful in understanding the urban increment, and some evaluation of improvement in performance compared to satellite-derived PM$_{2.5}$ is included. However, the evidence is a bit indirect. I'd like to see a comparison of native 1x1 and urban downscaled BC concentration to in situ measurements from urban monitoring sites, such as are available in the US.
* * *
REPLY: The referee is correct in stating that the evidence is indirect and that improvements are possible. However we feel that an intercomparison with BC from monitoring stations is not the most appropriate way, because
- there are large uncertainties with BC mass measurements
- BC in TM5 really represent Elemental Carbon (excluding observation that are based on optical measurements),
- not in the least TM5, like many other models, has a low-bias towards observations.

Further, the urban-incremented FASST mean 1°x1° concentration is not directly comparable to point measurements of monitoring stations in particular when placed in urban locations. To address the reviewer's comment, we have instead elaborated the recent data set of van Donkelaar et al. (2016) which integrates a PM$_{2.5}$ satellite product for anthropogenic PM$_{2.5}$ with data from monitoring stations. The data set is available at a 0.1°x0.1° resolution, allowing for an aggregation at population-weighted 1°x1° grid mean that can directly be compared to FASST native as well as urban-incremented concentrations at grid cell or regional level.

CHANGES TO MANUSCRIPT:
*We have significantly extended section S4 of the SI with additional text, figures and tables. We include here the new text of the section and refer to the revised SI for the figures.*

**S4.2 Comparison of TM5-FASST urban incremented PM$_{2.5}$ with observations**

We use the year 2010 0.1°x0.1° resolution global satellite product from the Dalhousie University Atmospheric Composition Analysis group (available at http://fizz.phys.dal.ca/~atmos/martin/?page_id=140), which includes ground-based observations via a Geographically Weighted Regression, while mineral dust and seasalt have been removed, as described in van Donkelaar et al., (2016).
The high-resolution satellite data (SAT) contain the sub-grid population and concentration gradients that we try to simulate with parametrization described above. Creating a SAT population-weighted average at 1°x1° resolution makes it possible to evaluate the TM5-FASST native and urban-incremented 1°x1° output. We convert the 0.1°x0.1° SAT resolution to the 2.5'x2.5' resolution of the CIESIN (year 2000) population dataset i.e. 24 sub-grid cells for each 1°x1° cell, to be overlaid with the satellite dataset. FASST PM$_{2.5}$ 1°x1° grid maps are calculated from the HTAP2 year 2010 emission inventory, includig the GFED v3 biomass burning emission inventor (REF). To remain consistent with the SAT product, residual water at 35% has been included. Fig. S4.1 shows global gridmaps of FASST and SAT PM$_{2.5}$ with dust and sea salt removed), and with the sub-grid increment included in the FASST result.
We evaluate both FASST native and urban incremented 1°x1°grid cell concentrations, using the parameterization described in the previous section. We calculate the following 1°x1° grid mean concentrations from the 2.5'x2.5' SAT PM$_{2.5}$ and population sub-grid cells

$$SAT_{AREA} = \frac{1}{24} \sum_{i=1}^{24} PM_{2.5,i}$$

$$SAT_{POP} = \frac{\sum_{i=1}^{24} PM_{2.5,i} \cdot POP_i}{\sum_{i=1}^{24} POP_i}$$

$SAT_{AREA}$ is the equivalent of the native FASST 1°x1° grid cell concentration, while $SAT_{POP}$ represents the population-weighted mean 1°x1° concentration considering sub-grid gradients, to be compared with the FASST urban-incremented value, hereafter referred to as incremented concentrations. Regional and global mean population exposure to $PM_{2.5}$ (Table S4.3) is calculated using population-weighing on the 1°x1° grid cells, for both native (area-mean) and incremented concentrations.

Table S4.3 and Fig. S4.2 show that for all regions, except for MEA (Mediterranean + Middle East), we find an over-all good agreement in regional mean $PM_{2.5}$ exposure between FASST and SAT, both for the native and incremented values. Figure S4.3 shows the absolute regional-mean increment in $PM_{2.5}$ exposure. We find that applying the FASST sub-grid parameterization increases global mean exposure with 1.4 µg m$^{-3}$ (FASST), versus an increase of 1.1 from SAT, corresponding to a global population-weighted mean 5% increase for both methods. The FASST urban increment parameterization generates a regional-mean increase in $PM_{2.5}$ exposure from 0.6 µg/m$^3$ (Latin America) to 3.4 µg/m$^3$ (Russia and former Soviet Union states). In Europe and North-America the regional increase is around 1µg/m$^3$. Except for East-Asia and Latin America, the regional FASST increment exceeds the SAT value. SAT regional increments range between 0.3 µg/m$^3$ for Russia and former Soviet Union states and 1.8 µg/m$^3$ in East-Asia. Although we don't find a direct correlation between the SAT and FASST computed increments, it is encouraging that without applying any fitting procedure, and using two completely different approaches, increments from FASST and SAT are in the same order of magnitude.

Figs. S4.4 (Europe and North-America), S4.5 (China and India) and S4.6 (Africa and Latin America) show a detailed grid-to-grid comparison for selected key regions between native and incremented FASST on the one hand and $SAT_{POP}$ on the other. In general, individual grid cells are reproduced within a factor of two. The FASST increment parameterization slightly improves the correspondence with $SAT_{POP}$ compared to the native data except for China where the native FASST concentrations already exceed $SAT_{POP}$. Although an agreement at grid cell level is not the ambition of FASST, these results indicate that our crude approach is roughly performing, but that a more sophisticated approach in the urban increment may be warranted.

Finally, seen the large uncertainties on absolute $PM_{2.5}$ concentrations, one may wonder if the implementation of an urban increment parameterization is worth the effort. A FASST RCP2000 analysis of global mortalities with and without the generic urban increment factors (given in Table S4.2) shows that the global 5% increase in $PM_{2.5}$ exposure due to the urban increment accounts for an increase in total mortality numbers with 14% when dry $PM_{2.5}$ is considered, and with 11% when $PM_{2.5}$ is humidified at 35% RH. The difference is due to the threshold in the exposure-response functions (see section S5 in this SI). In areas where the native grid concentration is just below threshold, a small increase in $PM_{2.5}$ will have a strong response in mortalities while areas with native 1°x1° concentrations above the threshold will respond more proportional to the subgrid increment. Including hygroscopic growth at 35% from the onset reduces the cases where native resolution $PM_{2.5}$ remains below the threshold which explains the lower impact of the subgrid increment factor.

21) 11.25-27: This justification would be improved if the authors were a bit more quantitative. Also, if the lowest model level $O_3$ compares favorably to the surface $O_3$ measurements, this begs the question then of why the modeled $O_3$ thought the lower atmosphere in TM5 is biased low (as surface-level concentrations would be lower than 30 m concentrations).

REPLY: As can be seen in the paper by Van Dingenen et al. (2009), which uses the same TM5 model versions, but slightly different emissions (see their figures 6 and 7 copied below, but not used in the current manuscript) during the summer months (i.e. crop growing season), the daytime a vertical gradient beween 30 m (model centre) and 10 m (standard height of

observations) is nearly absent – presumably due to higher atmospheric instability. The TM5 value at 30m is generally reproducing well the observations, and when it does not, the vertical gradient in TM5 is not the dominant factor causing discrepancies.

[Figure]

CHANGES TO MANUSCRIPT:

*Changed phrase*
However comparing TM5 simulated gridbox-centre ozone metrics with observations from 99 monitoring stations 25 worldwide, Van Dingenen et al. (2009) find that, averaged over the horizontal resolution of the grid cells, the TM5 simulated 30m monthly $O_3$ and $O_3$ metrics represent the observed values within their variability range.

*To (P13 L10):*
However comparing TM5 simulated gridbox-centre ozone metrics with observations from 99 monitoring stations worldwide, Van Dingenen et al. (2009) find that, when averaged at the regional scale, TM5 simulated crop metrics obtained from the grid box centre are reproducing the observations within their standard deviations, and that the monthly 10m TM5 metric values do not significantly improve the bias between model value and observations. Therefore we use the standard model output at 30m.

22) 12.21: The authors evaluate their approach to calculating direct aerosol radiative forcing by providing plots of the species specific forcings in Fig S6.1 and noting they are "reliable results." However, I don't have any sense of what makes these results reliable. What features of the distributions shown in these plots are those that we would expect, easily explain, or could compare to observations or other modeling studies? The BC RF in the eastern part of Antarctica exhibits a strange horizontal strip that I'm not sure about. Also, the figure legend on S6.1 is redundant and the units on both of the color bars are incorrect.

REPLY:
The reviewer correctly questions our use of the word reliable, which overstates our confidence in the uncertainty associated with the whole computational chain from emissions to concentrations and aerosol columns, and the scaling with normalized radiative forcing patterns. All these steps come with intrinsic uncertainties, and the radiative forcing uncertainty at grid basis is inevitably associated with relatively large uncertainties. The statement on 'reliability' is based on comparison with other globally aggregated results with an independent study performed by Unger et al, which gives remarkably similar source specific RF results. The 'strip' at the South Pole is likely due a numerical issue related the polar singularity and the necessary grid-size inflation in TM5 to deal with the singularity in a lon-lat projection.

CHANGES TO MANUSCRIPT: *Modified the phrase to (P14 L19):*

Our evaluation of pre-industrial to present radiative forcing in the validation section demonstrates that, in the context of the reduced-form FASST approach, the applied method however provides useful results. Figure S6.1 (a, b, c) in the SI shows the resulting global radiative forcing fields for sulfate, POM and BC. The regional emission-normalized forcing SRs for aerosol precursors (in W m$^{-2}$ Tg$^{-1}$) are given in Table S6.2 of the SI.

*Figure S6.1 has been modified to display the proper legend next to each graph, and units have been corrected to W mg$^{-1}$.*

23) Section 2.7.2: The tool does not include the substantial non-direct cloud interactions for BC, nor the impact of BC on snow/ ice albedo. These factors contribute significantly to the targeting of BC-rich sources for SLCP mitigation. Comment on how omission would affect TM5-FASST results.

REPLY:
This is a correct observation, and indeed worth mentioning. Surface albedo effects (snow and sea-ice) is estimated to contribute with (+0.04 to +0.33) W/m$^2$, cloud interaction with (-0.47 to +1.0) W/m$^2$ on a total estimated forcing of (0.17 to 2.1) W/m$^2$ (Bond et al., 2013) where FASST estimates a total anthropogenic BC forcing of +0.15 W/m$^2$ hence all these contributions are significant. As mentioned in the conclusions, future developments could indeed include these effects, in particular changes in the surface albedo, seen the fact that BC deposition is computed by FASST. Nevertheless, we not that TM5, like other global models, has large uncertainties associated with the calculation of BC depositions.

CHANGES TO MANUSCRIPT:
*In section 2.7.1 (P14 L15):*
Neglecting the aerosol mixing state and using column-integrated mass rather than vertical profiles introduces additional uncertainties in the resulting forcing efficiencies. Accounting for internal mixing may increase the BC absorption by 50 to 200% (Bond et al., 2013), while including the vertical profile would weaken BC forcing and increase SO$_4$ forcing (Stjern et al. 2016). Further, the BC forcing contribution through the impact on snow and ice is not included, nor are semi- and indirect effects of BC on clouds. Our evaluation of pre-industrial to present

radiative forcing in the validation section demonstrates that, in the context of the reduced-form FASST approach, the applied method however provides useful results.

*In section 3.3.1(P24 L25):*
However, comparing to another widely used literature source (Bond et al., 2013), the TM5-FASST_v0 BC forcing estimate still falls within the 90% CI (0.08, 1.27) W/m$^2$ direct radiative forcing given for the year 2005, with a comparable global BC emission rate. Our low-end BC forcing estimate can be partly explained by the simplified treatment as externally mixed aerosol, without accounting for the enhancement of the mass absorption cross-section when BC particles become mixed or coated with scattering components. Not-included snow albedo and indirect cloud effects would contribute with +0.13 (+0.04 to +0.33) W/m$^2$ and +0.23 (-0.47 to +1.0) W/m$^2$ respectively (Bond et al., 2013).
* * *
24) Fig 2: What is the mechanism by which the perturbation in NO$_x$ emissions causes a reduction in SO4 in IND (as opposed to an increase in all other regions)?
* * *
REPLY: This is indeed an interesting observation, which could be linked to the oxidative capacity of the atmosphere in that region, and/or to the thermodynamic properties of the ammonium-sulfate-nitrate system and the specific meteorological conditions in that area. Indeed, India has the particular feature that sulfate is dominating the inorganic aerosol fraction, and NH3 may be in excess. Answering this question would require a deeper analysis of TM5 budget data and the particular thermodynamic aerosol regimes for this case, where we notice that especially above India there are no reliable observations that could shed light on model discrepancies . Therefore we think that further analysis of this interesting model result is beyond the scope of this work where we focus on documenting and validating the linearity approach of FASST. However, in the text we point to this results for further multi-model analysis.

CHANGES TO MANUSCRIPT: *the paragraph has been expanded as follows: (P19 L4)*

For India we further observe a relative strong nitrate response to NO$_x$ emissions, with NO$_3$⁻ increasing by a factor of 3 for a doubling of NO$_x$ emissions, although the responses shown in Fig. 2 indicate that absolute changes (in µg m$^{-3}$) in NO$_3$ are relatively low and that secondary PM$_{2.5}$ in this region is dominated by SO$_4$. We are not aware of reliable observations or other published NO$_x$-aerosol sensitivity studies from that region that could corroborate this calculated sensitivity. Because such a feature may strongly affect projected future PM$_{2.5}$ levels and associated impacts, we recommend regional multi-model studies devote attention this feature.
* * *
25) 18.1-2: It seems NO$_x$ levels in the US and Europe are much lower now, and I'm not sure these titrations still exist; they are at least less persistent in the summer. See for example recent article by Jin, Fiore, et al., JGR, 2017.
* * *
REPLY: Thank you for pointing to this interesting paper. Indeed NO$_x$ emissions have been decreasing in the last two decades and indeed, the FASST SR relations were established for year 2000 conditions favouring a NO$_x$-saturation regime over W-Europe and NE-US. The fixed O$_3$ emission-response slopes are a major caveat for the evaluation of future scenarios, however, as already pointed out in the paper, while annual O$_3$ displays the typical reverse NO$_x$-O$_3$ response because of the winter-time titration, the slope reverses to positive in most cases when considering seasonal metrics centred on summer (Figure 6 in our paper). This being said, further reduction in NO$_x$ and NMVOC is likely to change the O$_3$ (metric) response sensitivity, and indeed the fixed and linear SRs are a limitation of the tool. A possible, but non-trivial implementation, way to address this trend is to introduce higher order terms in the SRs and/or

to update the year 2000 SRs with more recent ones obtained in the frame of HTAP2 (e.g. based on Turnock et al., 2018).

CHANGES TO MANUSCRIPT:
*This issue is now introduced in the discussion section 4.1 (P30 L4)*

The reliability of the model output in terms of impacts depends critically on the validity of the linearity assumption for the relevant exposure metrics (in particular secondary components), which becomes an issue when evaluating emission scenarios that deviate strongly from the base and -20% perturbation on which the current FASST SRs are based. The evaluation exercise indicated that non-linearity effects in $PM_{2.5}$ and $O_3$ metrics in general lead to a higher bias for stringent emission reductions (towards -80% and beyond) than for strong emission increases compared to the RCP2000 base case, but over-all remain within acceptable limits when considering impacts. Indeed, because of the thresholds included in exposure-response functions, the higher uncertainty on low (below-threshold) pollutant levels from strong emission reductions has a low weight in the quantification of most impacts. In future developments the available extended-range (-80%, +100%) emission perturbation simulations could form the basis of a more sophisticated parameterization including a bias correction based on second order terms following the approach by Wild et al. (2012) both for $O_3$ and secondary $PM_{2.5.}$ The break-down of the linearity at low emission strengths is relevant for $O_3$ and $O_3$ exposure metrics as the implementation of control measures in Europe and the US has already substantially lowered $NO_x$ levels over the past decade, gradually modifying the prevailing $O_3$ formation regime from $NO_x$-saturated (titration regime) to $NO_x$-limited (Jin et al., 2017).

26) 18.6: The statement that the sensitivities of impact-relevant $O_3$ metrics (M6M and M12) are more linear than for annual average pop-weighted $O_3$ is not supported by the results shown in Fig 6. The responses to $NO_x$ emission changes seem to be more nonlinear for M6M or M12 in some cases, such as GBR as well as others. The text should be revised accordingly.

REPLY: This is indeed wrongly formulated; this should rather refer to the changing sign of the slope.

CHANGES TO MANUSCRIPT: *changed phrase*
However, the impact-relevant $O_3$ metrics, both health and crop related, are based on summertime and daytime values and are expected to behave more linearly (Wu et al., 2009).

*To (P20 L27):*
On the other hand, the impact-relevant $O_3$ metrics, both health and crop related, are based on summertime and daytime values and are expected to be less affected by titration and consequently to maintain a positive emission-response slope (Wu et al., 2009).

27) 18.20: AOT40 would focus on high $O_3$ values. It's not clear to my, chemically speaking, why this would be expected to response more nonlinearly than other metrics. Presumably larger $O_3$ values are occurring more in the summer. Earlier it was claimed that summer sensitivities would be more linear ... so I'm a bit confused here.

REPLY:
AOT40 is a threshold-based metric accumulating only values above 40ppb, and this built-in step function makes it difficult to approximate it with a linear function over a large perturbation range. For instance, in regions where ozone levels are just above 40ppb, a small decrease in $O_3$ can cause a big decrease in AOT40, while a similar small increase would cause a smaller AOT40 response (in absolute terms). Similarly, a SR sensitivity established from a perturbation at high $O_3$ levels will behave rather linearly in the high $O_3$ range, but cannot be extrapolated to very

strong reductions where it will lead to an overestimation of AOT40. Therefore the SR sensitivity based on the 20% decrease is less likely to be generally applicable over a large perturbation range.

This is less the case of metrics like M12 which are based on 3-monthly means of daytime ozone and behave more linearly with respect to emission perturbations.

28) 18.31: Why is that the case, chemically speaking?

REPLY: In this case – without investigating the underlying chemical mechanism - the larger deviation is a consequence of the slight convex shape of the $O_3$ response to $NO_x$ for these countries, combined with the extrapolation of the -20% slope to larger perturbations.

CHANGES TO MANUSCRIPT:
*With the major revision of sections 3.1.2 this particular phrase and the figures it was referring to have been removed. However we do mention (P19 L10):*

Because the TM5-FASST linearization is based on the extrapolation of the -20% perturbation slope, concave-shaped trends in Fig. 3 indicate a tendency of TM5-FASST to over-predict secondary $PM_{2.5}$ at large negative or positive emission perturbations, and opposite for convex-shaped trends.

29) 20.4-14: I find it interesting that the change in $PM_{2.5}$ is predicted by FASST better than absolute concentrations (which I would expect) but that the change in $O_3$ metrics is predicted more poorly than absolute concentrations (would not expect). Do the authors have any thoughts about the reasons behind the latter?

REPLY:
Ozone behaves in general less linearly than $PM_{2.5}$, i.e. the perturbation term in Eq. 2 is more robust for $PM_{2.5}$ than for $O_3$. For strong perturbations, either side of the reference case, total $PM_{2.5}$ is overpredicted, hence making the difference between the high and low emission case cancel out some of the bias compared to absolute total $PM_{2.5}$.

For $O_3$, one must consider that the relative contribution of the "base" term in Eq. 2 is relatively high, even for strong anthropogenic perturbations because it contains the natural background. Roughly speaking, setting all anthropogenic emissions to 0 would still leave about 30 ppb of 6mDMA1. Therefore, this fixed contribution in the total reduces the weight of the relative error of the perturbation term, but when making the delta it does not contribute anymore.

Note that we have introduced statistical metrics Normalized Mean Bias (NMB) and Mean Bias (MB) to evaluate the agreement between FASST and TM5 with
Normalized Mean Bias = $\overline{(FASST - TM5)}/\overline{TM5}$ and
Mean Bias = $\overline{(FASST - TM5)}$

CHANGES TO MANUSCRIPT:
*We included the following phrase in the re-written discussion in section 3.2 (P23 L13):*

Contrary to $PM_{2.5}$, the NMB for the delta 6mDMA1 between two scenarios is higher than the NMB on absolute concentrations, with a low bias for the delta metric of -38% and -45% for Europe and North-America respectively, and a high bias of 35 to 46% in Asia. However, the MB on the delta is of the same order or lower than the absolute concentrations (Table 9). This is a consequence of the fixed background ozone in the absolute concentration reducing the weight of the anthropogenic fraction in the relative error.

30) 20.15-22: That's a reasonable comparison. I also wonder though what is the total number of estimated premature deaths associated with $PM_{2.5}$ and $O_3$, and how these numbers compare to those in the literature (from e.g. GBD), for present day conditions. This would help evaluate the accuracy of the absolute estimates in addition to estimates of changes.

REPLY: We have now included a table with some values from literature (both or $PM_{2.5}$ and $O_3$) in section 3.3 which is dedicated to a comparison with other published work, also illustrating the various assumptions that are involved making a direct comparison quite difficult.

CHANGES TO MANUSCRIPT: *Included a new section under section 3.3.4 Health impacts (P26 L24):*

**Present-day health impacts**
Table 14 gives an overview of recent global $PM_{2.5}$ health impact studies, together with FASST estimates for the year 2000 (RCP) and year 2010 (HTAP2 scenario). The studies differ in emission inventories and year evaluated, in applied methodologies to estimate $PM_{2.5}$ exposure, in model resolution, as well as in the choice of the exposure response functions, the value of the minimum exposure threshold, and mortality statistics. Studies excluding natural dust from the exposure are mostly applying the log-lin exposure response function and RR from Krewski et al. (2009), and estimate between 1.6 and 2.7 million annual premature mortalities from $PM_{2.5}$ in scenario years 2000 to 2004. FASST returns 2.1 and 2.5 million deaths using the GBD and log-lin exposure functions respectively. Studies including mineral dust are mostly applying the GBD integrated exposure-response functions and a non-zero threshold to avoid unrealistically high relative risk rates at high $PM_{2.5}$ levels in regions frequently exposed to dust. Depending on the choice of the exposure-response function and scenario year, FASST obtains 2.6 to 4.1 million global deaths, comparable with the range 1.7 to 4.2 million from previous studies.
Global ozone mortalities reported in Table 15 have been commonly based on the Jerrett et al. (2009) methodology, implemented in FASST. FASST obtains 197 thousand and 340 thousand deaths for RCP 2000 and HTAP2 2010 scenarios respectively, while the earlier studies find 380 to 470 thousand deaths in 2000, and 140 to 250 thousand in 2010 – 2015. Differences can be attributed to model chemical and meteorological processes, emission inventories, and the use of different sources for respiratory base mortality statistics.
Both for $PM_{2.5}$ and $O_3$, the difference between the different studies falls within the combined RR uncertainty and model variability range.

**Table 14 Overview of previous studies on health impact of PM$_{2.5}$, together with FASST results for 2 different scenarios. Uncertainty ranges are as reported in the respective studies. The uncertainty range on FASST results includes the RR uncertainty only (Fig. S5.1 in the SI)**

| Reference | Year evaluated | Method | threshold | Exposure - response function | Global deaths (millions) |
|---|---|---|---|---|---|
| | | Excluding mineral dust | | | |
| Fang et al., 2013 | 2000 | CTM | no | K2009[a] | 1.6 (1.2 – 1.9) |
| Silva et al., 2013 | 2000 | CTM | no | K2009 | 2.1 (1.3 -3.0) |
| Anenberg et al., 2010 | 2000 | CTM | 5.8µg m$^{-3}$ | K2009 | 2.7 (2.0 -3.4) |
| Evans et al., 2013 | 2004 | SAT | 5.8µg m$^{-3}$ | K2009 | 2.7 (1.9 - 3.5) |
| Lelieveld et al., 2013 | 2005 | CTM | no | K2009 | 2.2 (2.1 - 2.3) |
| **FASST (RCP)** | **2000** | **FASST** | **~7.3µg m$^{-3}$** | **K2009** | **2.5 (1.2 – 3.6)** |
| **FASST (RCP)** | **2000** | **FASST** | **~7.3µg m$^{-3}$** | **B2014[b]** | **2.1 (1.0 – 3.0)** |
| | | Including mineral dust | | | |
| Silva et al., 2016 | 2000 | ACCMIP CTM ensemble | ~7.3µg m$^{-3}$ | B2014 | 1.7 (1.3 – 2.1) |
| Evans et al. 2013 | 2004 | SAT | 5.8µg m$^{-3}$ | K2009 | 4.3 (2.9 – 5.4) |
| Lelieveld et al., 2015 | 2010 | CTM | ~7.3µg m$^{-3}$ | B2014 | 3.2 (1.5 - 4.6) |
| GBD2010 (Lim et al., 2012) | 2010 | Fused (FASST + SAT + ground based) | ~7.3µg m$^{-3}$ | B2014 | 3.2 (2.8 -3.6) |
| GBD2013 (Forouzanfar et al., 2015) | 2013 | Fused (FASST + SAT + ground based) | ~7.3µg m$^{-3}$ | B2014 | 2.9 (2.8 – 3.1) |
| GBD2015 (Cohen et al., 2017) | 2015 | Fused (FASST + SAT + ground based) | ~4.1µg m$^{-3}$ | B2014 | 4.2 (3.7 – 4.8) |
| **FASST (RCP)** | **2000** | **FASST** | **~7.3µg m$^{-3}$** | **K2009** | **3.6 (2.7 -4.5)** |
| **FASST (RCP)** | **2000** | **FASST** | **~7.3µg m$^{-3}$** | **B2014** | **2.6 (1.2 – 3.8)** |
| **FASST (HTAP2)** | **2010** | **FASST** | **~7.3µg m$^{-3}$** | **B2014** | **4.1 (2.0 - 5.9)** |

(a) Krewski et al., 2009
(b) Burnett et al., 2014

**Table 15 Overview of previous studies on long-term health impact of ozone, together with FASST results for 2 different scenarios**

| Ref | year | Method | threshold | Exposure-response function | Global deaths (thousands) |
|---|---|---|---|---|---|
| Anenberg et al., 2010 | 2000 | CTM | 33.3 | J2009 [a] | 470 (182 - 758) |
| Silva et al., 2013 | 2000 | ACCMIP CTM ensemble | 33.3 | J2009 | 380 (117 -750) |
| Lelieveld et al., 2015 | 2010 | CTM | ~37.6 | J2009 | 142 (90 -208) |
| GBD 2010 (Lim et al., 2012) | 2010 | FASST | ~37.6 | J2009 | 152 (52 – 270) |
| GBD 2013 (Forouzanfar et al., 2015) | 2013 | FASST | ~37.6 | J2009 | 217 (161 – 272) |
| GBF 2015 (Cohen et al., 2017) | 2015 | FASST | ~37.6 | J2009 | 254 (97 – 422) |
| **FASST (RCP)** | **2000** | **FASST** | **33.3** | **J2009** | **197 (66 – 315)** |
| **FASST (HTAP2)** | **2010** | **FASST** | **33.3** | **J2009** | **340 (116 – 544)** |

(a) Jerrett et al., 2009
* * *
31) 20.25 - 21.7: I'm I incorrect in thinking that many of the pre-industrial to present IPCC RF's also include an 80% reduction in biomass burning sources? If so, this might further explain why the IPCC values are on the higher side. Also, IPCC estimates and those in Bond include RF of BC on snow, which I don't see as being accounted for in FASST.

REPLY:
We do not have the information on what reductions in biomass burning were assumed in IPCC models, but note that most recent studies point to smaller reductions, subject to large uncertainty. Large scale biomass burning is more prominent for OC emissions, than for BC. For instance, in the RCP2000 emission inventory, BC from large scale forest fires account for 15% of the total BC forcing, hence including BB does not make a large (absolute) difference on the already low BC forcing (from 0.15 to 0.17 $W/m^2$) and cannot account for the low bias. Other missing contributions could indeed be more relevant, like the BC mixing state and residence time, snow and ice albedo impacts and cloud interactions (see also our reply to comment 23).
* * *
32) 21.22: How do they know it's owing to different OH levels and lifetimes rather than to different emissions (line 21.14)?

REPLY: The reviewer makes a correct point. We cannot be certain about this statement. However, the Stevenson ACCMIP study was based on the same emissions database described by Lamarque et al. (2013) as used in this model study, which seems to point to differences in oxidation chemistry and resulting ozone production with respect to CO and NMVOC emissions. We also spotted an error in our data treatment and corrected the data in Table 10 which changes slightly the discussion.

CHANGES TO MANUSCRIPT *(P25 L6)*:
Table 10 compares the contribution of anthropogenic $O_3$ precursors $CH_4$, $NO_x$, NMVOC and CO to the $O_3$ and $CH_4$ radiative forcing with earlier work (Shindell et al., 2005, 2009; Stevenson et al., 2013). Except for $NO_x$ which shows a large scatter across the studies, the FASST computed contributions to global $O_3$ and $CH_4$ forcing  - using the same year 1850 to 2000  emission changes as in Stevenson et al. (2013) - are in good agreement with the model ensemble range in

the latter study. FASST $NO_x$ forcing contributions are a factor 3 lower than in the Stevenson et al. study and more in line with Shindell et al. (2005, 2009) values (based on the period 1750 – 2000), however the latter obtain a NMVOC contribution to $O_3$ forcing which is a factor of 5 to 6 lower than the other estimates. Differences across the studies are likely due to differences in oxidation chemistry and lifetimes across models.
* * *
33) Section 3.3.2: The evaluation of global sector and species specific RF looks good. A key feature of FASST is regional specificity; could they also compare to some studies in the literature that have evaluation the RF of regionally specific emissions by species or sector?
* * *
REPLY: We note that FASST does not contain sector-specific SRs, hence global forcing efficiencies (expressed as $mW/m^2/Tg$) for a single FASST source region are valid for the aggregated contributions of the regional sectors.
The most relevant studies to compare aerosol global forcing responses to regional emissions are the HTAP1 exercise (Yu et al., 2013) and the similar multi-model HTAP2 study (Stjern et al., 2016). For the NH regions considered in these studies, our results correspond well (within 1 stdev) with older Yu et al. study (based on a single model, using similar emission and meteorological year as FASST base simulation), whereas the multi-model ensemble mean of Stjern et al. gives higher forcing efficiencies, although in the latter case the model variability is large, and our results stays within 2 stdv (95% confidence interval).

CHANGES TO MANUSCRIPT:
*We have included an additional subsection under 3.3 (Comparison of TM5-FASST_v0 impact estimates with published studies) (P25 L15):*

**3.2.2 Regional forcing efficiencies by emitted component**
Earlier work in the frame of HTAP1 (Fry et al., 2012; Yu et al., 2013) and HTAP2 (Stjern et al., 2016) evaluated regional forcing efficiencies for larger regions than the ones defined for FASST. For a comparison we aggregate the FASST forcing efficiencies (as listed in section S6.3 of the SI) by making an emission-weighted averages over Europe (EUR), North-America (NAM), South-Asia (SAS), East-Asia (EAS), Mediterranean and Middle East (MEA) and Russia, Belarus and Ukraine (RBU). Tables 11 (PM precursors) and 12 ($NO_x$ and NMVOC) show the earlier studies along with the FASST results. The FASST forcing efficiencies for PM precursors confirm our earlier observation that FASST is particularly biased low for BC, in particular compared to Stjern et al. (2016), but further compares relatively well with the earlier work, in particular with Yu et al. (2013) which was based on a year 2001 baseline, similar to conditions of our base scenario. A similar observation is made for NMVOC for which FASST efficiencies agree well with the study by Fry et al. (2012). The forcing efficiency for ozone precursor $NO_x$ has a high uncertainty. While for East-Asia, North-America and South-Asia the FASST result falls within 1 standard deviation of the HTAP1 model ensemble the FASST $NO_x$ forcing efficiency for Europe shows a larger deviation. Without going into the details of the underlying mechanisms, ozone titration effects, which are better resolved with the higher TM5 model resolution, could be a contributing factor.

*New tables 11 and 12:*

**Table 11. Regional-to-global direct radiative forcing efficiencies for $PM_{2.5}$ precursors (mW/m$^2$/Tg of annual emissions) for the lager source-receptor regions in earlier studies, and from FASST, aggregated to similar regional definitions. Values in brackets represent 1 standard deviation from the respective reported model ensembles.**

|  |  | NAM | EUR | SAS | EAS | RUS | MEA |
|---|---|---|---|---|---|---|---|
| Stjern et al., 2016 | BC | 52 (±21) | 55 (±22) | 94 (±38) | 55 (±16) | 78 (±47) | 202 (±323) |
|  | POM | -8 (±6) | -7 (±4) | -10 (±6) | -5 (±3) | -2 (±5) | -18 (±7) |
|  | SO4 (SO2) | -5 (±2) | -6 (±2) | -8 (±4) | -4 (±1) | -4 (±1) | -10 (±7) |
| Yu et al., 2013 | BC | 27 (±15) | 37 (±19) | 25 (±15) | 28 (±20) |  |  |
|  | POM | -4 (±2) | -4 (±2) | -4 (±2) | -4 (±2) |  |  |
|  | SO4 (SO2) | -4 (±1) | -4 (±1) | -4 (±1) | -3 (±1) |  |  |
| FASST (RCP2000) | BC | 17 | 19 | 19 | 16 | 25 | 43 |
|  | POM | -6 | -4 | -6 | -5 | -4 | -9 |
|  | SO4 (SO2) | -3 | -3 | -4 | -2 | -2 | -7 |

**Table 12. Regional-to-global direct radiative forcing efficiencies for $O_3$ precursors (mW/m²/Tg of annual emissions) for the lager source-receptor regions in earlier work, and from FASST, aggregated to similar regional definitions, including feedbacks on CH4. Values in brackets represent reported 1 standard deviation from the model ensemble in the earlier work.**

|  |  | EAS | EUR | NAM | SAS |
|---|---|---|---|---|---|
| Fry et al., 2010 | $NO_x$ | -0.22 (±0.6) | -1.20 (±0.5) | -0.48 (±0.6) | -1.70 (±2.2) |
|  | NMVOC | 0.42 (±0.2) | 0.46 (±0.2) | 0.42 (±0.2) | 0.72 (±0.2) |
| FASST (RCP200) | $NO_x$ | -0.44 | -0.33 | -0.35 | -1.43 |
|  | NMVOC | 0.60 | 0.57 | 0.61 | 0.74 |
* * *
34) 23.25: Does including this correction for changing mortality rates though lead to worse agreement between ACCMIP and FAST for PM2.5 related deaths (Fig 15)?

REPLY:
It does not because according to the GBD methodology, respiratory mortality is not considered in the PM$_{2.5}$ related causes of death (which are: COPD, LC, IHD and Stroke), it contributes only to the O$_3$ health impact. The ACCMIP projections of PM-relevant base mortalities are much more in line with the ones used in FASST.

CHANGES TO MANUSCRIPT: *added the following phrase (P28 L25):*
Respiratory mortality is not considered as a cause of death for PM2.5, which explains why a similar disagreement is not observed in the PM$_{2.5}$ mortality trend in Fig. 17b.
* * *
35) Section 4: Good discussion. Some caveats about missing accurate treatment of SOA? Or carbonaceous aerosol aging? And possibly being a bit more clear about the limits of the emissions perturbations magnitudes that should be used with this tool (e.g., x2? x5? x10?).

REPLY:
Thank you for the positive feedback and the suggestions. We have extended some of the discussion making a wrap-up of the major caveats of the tool. Regarding the limits of the emission perturbations magnitudes, this depends on many parameters such as the region of

emission, on the emission ratio between various precursors, so it is not possible to set an overall validity range. We believe that the MIT and FLE scenarios explore the domain boundaries in which 'reasonable' emission changes for the next decades (until 2030) and that TM5-FASST behaves sufficiently well to be used as a screening tool to explore scenarios further out in the future.

CHANGES TO MANUSCRIPT

*New discussion section 4.1 is – amongst other caveats - addressing the issues mentioned (P31 L16)*

[revised manuscript text omitted]

---

## Author Comment (AC3) · 6 Sep 2018

The comment was uploaded in the form of a supplement:
https://www.atmos-chem-phys-discuss.net/acp-2018-112/acp-2018-112-AC3-supplement.zip
* * *